# ACTIVATION FUNCTION DESIGN SUSTAINS PLASTICITY IN CONTINUAL LEARNING

**Lute Lillo & Nick Cheney**
Department of Computer Science
University of Vermont
Burlington, VT 05401, USA
`{elillopo,ncheney}@uvm.edu`

## ABSTRACT

In independent, identically distributed (i.i.d.) training regimes, activation functions have been benchmarked extensively, and their differences often shrink once model size and optimization are tuned. In continual learning, however, the picture is different: beyond catastrophic forgetting, models can progressively lose the ability to adapt (referred to as *loss of plasticity*) and the role of the non-linearity in this failure mode remains underexplored. We show that activation choice is a primary, architecture-agnostic lever for mitigating plasticity loss. Building on a property-level analysis of negative-branch shape and saturation behavior, we introduce two drop-in nonlinearities (*Smooth-Leaky* and *Randomized Smooth-Leaky*) and evaluate them in two complementary settings: (i) supervised class-incremental benchmarks and (ii) reinforcement learning with non-stationary MuJoCo environments designed to induce controlled distribution and dynamics shifts. We also provide a simple stress protocol and diagnostics that link the shape of the activation to the adaptation under change. The takeaway is straightforward: thoughtful activation design offers a lightweight, domain-general way to sustain plasticity in continual learning without extra capacity or task-specific tuning.

## 1 INTRODUCTION

Continual learning requires neural networks to acquire new knowledge over time without erasing previously learned information. This poses a fundamental challenge: maintaining a balance between plasticity—the ability to adapt to new data—and stability—the ability to retain prior knowledge. While *catastrophic forgetting* refers to poor performance on previously learned tasks when they are no longer explicitly trained on, loss of plasticity is a distinct phenomenon: networks might retain past capabilities but become increasingly incapable of learning new ones. Despite growing interest, *loss of plasticity* remains less understood and underexplored, particularly in reinforcement learning (RL) settings where the agent's evolving policy changes the distribution of data it encounters, making it difficult to disentangle learning ability from environmental exposure.

Recent work documents symptoms associated with plasticity loss in deep RL, including reduced gradient magnitudes (Abbas et al., 2023), increasing parameter norms (Nikishin et al., 2022), rank-deficient curvature (Lyle et al., 2022; Lewandowski et al., 2023), and declining representation diversity (Kumar et al., 2020; 2023; Dohare et al., 2024). Yet no single factor explains its onset across settings. (Lyle et al., 2024) propose a "Swiss cheese" view: multiple, partly independent mechanisms—e.g., pre-activation distribution shift, uncontrolled parameter growth, and the scale of bootstrapped value targets in temporal-difference learning—can each contribute. Mitigation strategies range from architectural refresh (Continual Backprop's generate-and-test replacement of low-utility units (Dohare et al., 2021)) to regularization that targets plasticity retention (Lyle et al., 2022; Kumar et al., 2023); see (Klein et al., 2024) for a survey.

We argue that a more fundamental knob is hiding in plain sight: the *activation function*. Differences among activations often shrink in i.i.d. training once model size and optimization are tuned, but under continual, non-stationary data they can matter substantially. This motivates a property-level study

of how activation shape, especially negative-side responsiveness and saturation, affects plasticity[1]. Previous work has shown that alternative activations such as CReLU (Shang et al., 2016; Abbas et al., 2023; Lee et al., 2023) or rational functions (Molina et al., 2019; Delfosse et al., 2021b;a) can improve performance under non-stationary conditions. Section 2 closes with a simple IID vs. class-incremental (C-IL) comparison using a shared setup to illustrate this contrast and motivate the case studies that follow.

Our contributions are as follows:

- **Comparison of activations function performance** across supervised continual learning and non-stationary RL benchmarks (Sec. 2, Tab. A1).
- **Analysis of activation function properties** identifying a moderate, non-zero negative-side responsiveness ('Goldilocks zone') and small dead-band width as key predictors of sustained plasticity (Secs. 3, 4).
- **Two drop-in activation functions**—*Smooth-Leaky* and *Randomized Smooth-Leaky*—that preserve a non-zero derivative floor and target the moderate-leak regime with a $C^1$ transition (i.e., having a continuous first derivative) between its linear and non-linear regions efficiently improving continual adaptation. (Secs. 6, 7).

## 2 ACTIVATION FUNCTIONS AND PLASTICITY IN CONTINUAL LEARNING

Activation functions are the first gatekeepers of gradient information. Their slope near zero, negative input behavior, and degree of saturation jointly determine how much learning signal survives backpropagation, a critical factor once data distributions change. We provide a comparative overview of commonly used activation functions, highlighting their potential to either exacerbate or mitigate plasticity loss.

**Rectifiers.** ReLU is efficient but prone to the *dead-unit* problem (Maas et al., 2013): neurons that output 0 receive no gradient and thus become inactive permanently. Continual-learning studies confirm a rising fraction of dormant units over time (dormant neurons phenomenon (Sokar et al., 2023)), shrinking gradient norms and eventually freezing learning (Abbas et al., 2023; Dohare et al., 2024). ReLU networks often show an increase in parameter norms during training (as they drive outputs with ever-larger weights) and a drop in the number of effective directions in weight space that can reduce error (Lewandowski et al., 2023). Leaky-ReLU alleviates this with a constant negative slope, while PReLU (He et al., 2015) and RReLU (Xu et al., 2015) postpone dormancy by making the negative slope learnable or by randomly sampling it during training.

**Saturating sigmoids.** Sigmoid and Tanh map inputs to bounded ranges; when units saturate, derivatives shrink toward zero and gradients can vanish, slowing learning (Glorot & Bengio, 2010). This is relevant for continual learning, which requires sustained adaptation under shift and where loss of plasticity has been repeatedly observed (Dohare et al., 2024; Abbas et al., 2023; Juliani & Ash, 2024). (See Sec. 4 for empirical evidence.)

**Smooth non-monotonic.** Swish (Ramachandran et al., 2017) and GeLU (Hendrycks & Gimpel, 2016) are smooth, weakly non-monotonic rectifiers that preserve small—but non-zero—gradients for inputs near and below zero. This mitigates "dying ReLU" behavior (Maas et al., 2013), so units remain trainable when pre-activation distributions drift under shift. In continual settings, this responsiveness supports ongoing adaptation; empirically, non-monotonic/smooth rectifiers have shown advantages in both supervised and RL domains (Ramachandran et al., 2017; Hendrycks & Gimpel, 2016; Elfwing et al., 2018), and in our experiments (Secs. 3–4) they exhibit lower dead-unit fractions and stronger late-cycle adaptation than zero-floor rectifiers.

**Exponential variants.** ELU/CELU (Clevert et al., 2015; Barron, 2017) reduce bias shift via a negative branch (CELU is $C^1$), while SELU (Klambauer et al., 2017) self-normalizes activations toward zero mean/unit variance. Under continual, non-i.i.d. data, where batch-statistics can drift and hurt retention, such built-in stabilization helps maintain trainable scales across tasks (Ioffe, 2017; Pham et al., 2022).

**Why focus on continual learning?** Under identical models and training budgets on Split-CIFAR-100, activation function rankings compress in i.i.d. joint training but separate sharply in class-incremental (C-IL) settings (Van de Ven et al., 2022) (see Table 1). This motivates probing how

---

[1]Code available at: `https://github.com/lute47lillo/activations_plasticity`

negative-branch behavior affects plasticity under shift. Unless noted, all case studies use the same 4-layer CNN backbone, the Adam optimizer (Kingma, 2014), and training budget (full details in App. B), isolating activation effects from architectural or optimization confusion.

| | *ReLU* | *LReLU* | *RReLU* | *PReLU* | *Swish* | *GeLU* | *CeLU* | *eLU* | *SeLU* | *Tanh* | *Sigmoid* |
|---|---|---|---|---|---|---|---|---|---|---|---|
| **i.i.d.** | 72.11 (0.40) | 72.00 (0.63) | **73.71 (0.24)** | 71.43 (0.52) | 73.16 (0.55) | 72.51 (0.8) | 72.66 (0.42) | 72.64 (0.33) | 72.12 (0.78) | 66.49 (0.59) | 58.78 (0.48) |
| **C-IL** | 24.41 (0.75) | 28.57 (0.64) | **32.95 (0.12)** | 22.71 (0.93) | 24.43 (0.86) | 20.91 (0.64) | 22.79 (0.25) | 27.59 (0.89) | 27.49 (0.18) | 26.44 (0.64) | 25.47 (0.69) |

Table 1: Values for Split-CIFAR-100 are reported as average accuracy (standard deviation) for 5 independent runs with identical architecture, optimizer, and budget. Performance differences across activation functions are modest under i.i.d. joint training but widen under class-incremental learning (C-IL). RReLU attains the top mean in both settings; the C-IL improvement is statistically significant (all $p < 0.05$), whereas i.i.d. differences are not.

## 3 CASE STUDY 1: NEGATIVE-SLOPE 'GOLDILOCKS ZONE'

We now test whether *negative-side responsiveness* drives plasticity under shift. Using the shared setup from Sec. 2 (Table 1), we sweep the negative branch for three families: piece-wise linear (Leaky-ReLU, RReLU), smooth-tailed activations (Swish, GeLU, ELU/CELU/SELU), and adaptive (PReLU at global/layer/neuron scopes). Our goals are: (i) test if there exists a consistently good value of the negative-side slope across activation functions; (ii) test whether smooth tails approximate the effective slope ($\bar{s}$) of optimal linear regimes; and (iii) assess whether adaptively learned slopes discover optimal values without extra guidance.

**'Goldilocks zone' for negative slopes (with shape-matched comparison).** Performance of activations with *constant* negative-branch slope (Leaky-ReLU, RReLU), reliably peaks for a moderate leak $0.6 \lesssim \bar{s} \lesssim 0.9$ and degrades once $\bar{s} \gtrsim 0.9$ (Fig. 1A, Tab. 1), suggesting a 'Goldilocks zone' for negative slopes. To compare smooth-tailed functions on a common scale, we project their negative-branch behavior onto an effective slope axis, $\bar{s} = \mathbb{E}_{x<0}[\varphi'(x)]$, representing the average derivative for negative inputs. When matched by $\bar{s}$, these smooth tails still underperform linear leaks within the 'Goldilocks zone' and exhibit higher dead-unit fractions (Fig. 1B), approaching (but not surpassing) the linear-leak peak only for $\bar{s} > 1$.

**Failure Modes for Slope Magnitude.** We identify two distinct mechanisms defining the zone's boundaries. As $\bar{s} \to 0$, a *dead-unit regime* dominates ($\approx 45\%$ inactive). We implement dead-unit here as effective inactivity, identifying units either stuck in saturation plateaus or contributing negligible magnitude relative to the layer's scale. This inactivity strongly correlates with accuracy loss (Pearson $r = -0.51$, $p = 8.2 \times 10^{-28}$), and weakly but also significant with respect to final scaled gradient norms (Pearson $r = -0.11$, $p = 0.029$) (App. C.1). Conversely, as $\bar{s} \to 1$, performance degrades *despite* minimal dead units. Fig. 1C-D reveals this coincides with *optimization instability*: sharp spikes in principal curvature ($\lambda_{\max}$) and effective rank (App. B.7). Thus, sustaining plasticity requires a trade-off: avoiding gradient starvation (low $\bar{s}$) without inducing landscape stiffness (high $\bar{s}$).

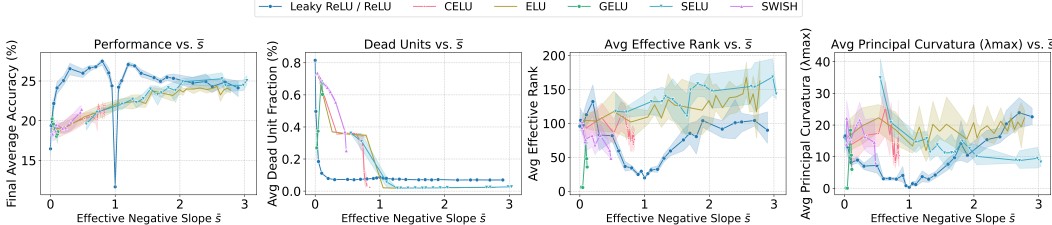

Figure 1: **A:** Final accuracy vs. effective negative slope $\bar{s}$. **B:** Dead-unit fraction vs. $\bar{s}$. Linear-leak families peak for $\bar{s} \in [0.6, 0.9]$. Smooth-tailed activations are plotted on the same $\bar{s}$ axis; they underperform within the 'Goldilocks zone' and only approach the linear-leak peak when $\bar{s} > 1$, reflecting concentrated near-zero responsiveness and vanishing tails. **C:** Effective rank of the gradient Gram matrix. **D:** Dominant $\lambda_{\max}$. Smooth-tailed activations show spikes at large $\bar{s}$, while constant-slope leaks remain comparatively stable.

**Current adaptive, learnable slopes fail and need constraints to stay in-band.** Under non-stationarity, the "right" negative-side responsiveness is not uniform across units or tasks. We therefore asked whether learnable or randomized slopes can find and *maintain* their specific 'Goldilocks zone'. Per-neuron PReLU (PReLU-N) drifts below the band ($\approx 0.3 - 0.6$) over training and attains 30.1% ACC (see Figs.. C.2); layer/global scopes drift even further (Figs. C3). Thus, adaptivity is *relevant*—it offers robustness when a single fixed leak cannot serve all units—but unconstrained adaptation does not reliably remain in the pre-defined 'Goldilocks zone', which explains why learned slopes may sit outside it despite good (but suboptimal) ACC.

# 4 CASE STUDY 2: DESATURATION DYNAMICS UNDER SHOCKS

Case Study 3 showed that negative leak (a non-zero derivative floor[2]) is necessary but not sufficient: after a distributional shift, many pre-activations can be pushed deep into an activation's tail, where gradients are effectively zero. We hypothesize that the time it takes a network to *desaturate* (how quickly gradients reopen after a shock) is a key determinant of adaptation delay in lifelong settings. Therefore, we subject the network to a protocol which isolates how quickly different activation families reopen gradients and regain performance after controlled shocks, complementing the steady-state results from Case Study 3.

Every $C_\ell{=}10$ epochs we apply a one-epoch *scaling shock* by multiplying all pre-activations $z$ by $\gamma \in \{1.5, 0.5, 0.25, 2.0\}$, then revert to $\gamma{=}1$ (App. D.1). Large $\gamma$ (e.g., 2.0) pushes $z$ into negative tails or positive plateaus (saturation); small $\gamma$ (e.g., 0.5) produces the mirrored event. Each activation uses its best negative-slope setting from Case Study 3 (Table B4). A unit is *saturated* at a step if $|\varphi'(x)| < 10^{-3}$. We report: (i) **Peak SF**: the maximum saturated fraction immediately after each shock; (ii) **AUSC**: the *area under the saturation curve* over the recovery window (lower is better); (iii) **Recovery time $\tau_{95}$**: steps needed to regain 95% of pre-shock performance (App. D.4).

**Derivative-floor rule.** Activations with a *strict* non-zero derivative floor (Leaky-ReLU, RReLU, PReLU) achieve the lowest AUSC and near-zero non-recovery rates (<5%) even under the strongest shocks (Fig. 2, middle/right). In contrast, zero-floor types (ReLU, Sigmoid, Tanh) show the largest AUSC and very high non-recovery across all $\gamma$ (Fig. 2, left).

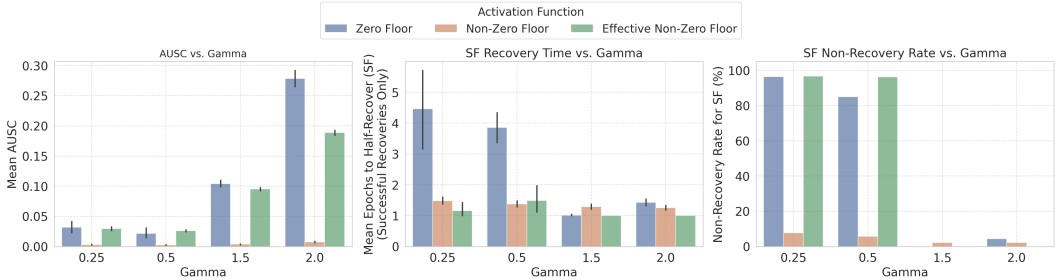

Figure 2: Desaturation under scaling shocks $\gamma$. **Left:** mean AUSC (lower is better). **Middle:** SF recovery time (epochs to halve the saturated fraction after the shock; successful recoveries only). **Right:** SF non-recovery rate (%). Groups: **Zero-floor** = ReLU, Tanh, Sigmoid; **Non-zero-floor** = Leaky-ReLU, RReLU, PReLU; **Effective non-zero-floor** = ELU, CELU, SELU, GELU, Swish. See App. D.2 for details.

**Two-sided penalty.** Activations that saturate on *both* sides (Sigmoid, Tanh) suffer the worst shocks: they show the largest peak saturated fraction and AUSC (Fig. 3, left/right), and fail to desaturate in roughly half of runs (49.8%). One-sided (Kink[3]) *non-zero-floor* rectifiers (Leaky-ReLU, PReLU, RReLU) recover far more reliably (non-recovery $\approx 13.3\%$). One-sided (Smooth) (ELU, CELU, SELU) sit between these extremes: when they recover, they do so quickly (Fig. 3, middle), but failures still occur frequently at strong shocks. Overall, a single hard saturation boundary is less

---

[2]We call an activation *zero-floor* if $\inf_x |\varphi'(x)| = 0$ (e.g., ReLU, and Sigmoid/Tanh whose derivatives approach 0 in the tails). We call it *non-zero-floor* if there exists $\alpha > 0$ such that $\varphi'(x) \geq \alpha$ on the negative branch (e.g., Leaky/PReLU/RReLU). We call it *effective non-zero floor* if the derivative is non-zero on finite negative inputs near the decision boundary even though it decays toward 0 as $x \to -\infty$ (e.g., Swish/GeLU).

[3]By *kink* we mean continuous but not differentiable at $x{=}0$ ($C^0$ but not $C^1$; e.g., ReLU, Leaky-ReLU). By *smooth* we mean at least once differentiable at $x{=}0$ ($C^1+$; e.g., ELU, Swish).

harmful than two; maintaining a non-zero derivative floor on the negative side remains the most protective.

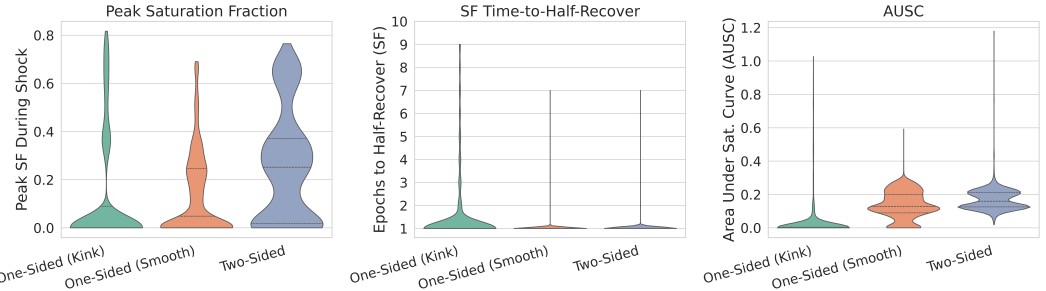

Figure 3: **Sidedness effects under shocks. Left:** Peak saturated fraction during the shock (higher = more units saturated). **Middle:** Saturation Fraction (SF) time-to-half-recover (epochs; successful recoveries only; lower is better). **Right:** AUSC (lower is better). Groups: **One-sided (kink)** = Leaky-ReLU, PReLU, RReLU; **One-sided (smooth)** = ELU, CELU, SELU; **Two-sided (saturating)** = Sigmoid, Tanh. See App. D.3 for details.

**The width of the dead band predicts shock sensitivity.** Beyond the findings of derivative floor and sidedness, we ask how *much* of the input range of an activation produces nearly zero gradients. We define a *Dead-Band Width (DBW)* as the fraction of a typical pre-activation range (e.g., [-100, 100]) where the magnitude of the activation's first derivative, $|\varphi'(x)|$ falls below certain threshold, $\epsilon < 10^{-3}$. Analytically computed $DBW$, $|\varphi'(x)| < 10^{-3}$, strongly tracks desaturation outcomes across activations.

We hypothesize that this analytically derived $DBW$ will positively correlate with experimentally observed adverse saturation dynamics. Intuitively, a wider intrinsic dead-band means pre-activation scaling shocks are more likely to push numerous units into these unresponsive, vanishing-gradient regions and keep them there. This, in turn, diminishes the network's ability to desaturate, which we expect to manifest as a greater overall saturation impact (higher AUSC) and increased non-recovery rates.

Figure 4 strongly supports this hypothesis, revealing that the analytical Dead-Band Width Score is strongly and significantly correlated with adverse saturation outcomes. Activations with a higher $DBW$ experience a greater overall saturation impact (Avg. AUSC, $r = 0.81$, $p = 0.0016$) and are much more likely to fail recovery entirely (Avg. SF Non-Recovery Rate, $r = 0.84$, $p = 0.0013$). $DBW$ does not predict recovery speed. Successful desaturation, when it occurs, is consistently fast (around one epoch) and shows no correlation with the $DBW$. This suggests the $DBW$ score predicts the likelihood and severity of saturation, but not the speed of recovery.

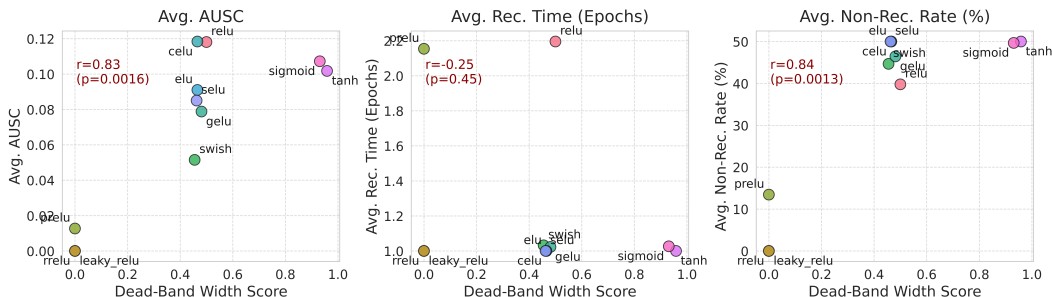

Figure 4: **Correlation of Dead-Band Width Score with Saturation Recovery Metrics (All Gammas Aggregated). (Left):** Average Area Under Saturation Curve (Avg. AUSC) vs. Dead-Band Width Score. A strong positive correlation (Pearson $r = 0.81$, $p = 0.0016$) is observed. **(Middle):** Average Saturation Fraction (SF) Recovery Time (for successful recoveries, measured by epochs) vs. Dead-Band Width Score. No significant correlation is found (Pearson $r = -0.25$, $p = 0.45$). **(Right):** Average SF Non-Recovery Rate (%) vs. Dead-Band Width Score. A strong positive correlation (Pearson $r = 0.84$, $p = 0.0013$) is observed, indicating functions more prone to saturation are more likely to fail SF recovery.

## 5 Implications for Activation-Function Design

Sections 3–4 suggest three rules for plasticity-friendly nonlinearities: (i) maintain a *non-zero derivative floor*, (ii) keep negative-side responsiveness in a moderate, activation-specific 'Goldilocks zone', and (iii) prefer a $C^1$ *(smooth) transition at the origin when (i)–(ii) are held fixed*. Conversely, avoid *two-sided saturation* and wide analytic *dead bands* ($|\varphi'| < 10^{-3}$), which track larger AUSC and non-recovery.

Our measurements reveal two competing objectives: *(A) recovery success* (low non-recovery rate after shocks) and *(B) recovery speed/extent* (low AUSC, short time-to-recover) *conditional on recovery*. Kinked $C^0$ rectifiers with a strict floor minimize non-recovery across $\gamma$ (Fig. 2, middle/right), whereas one-sided $C^1$ shapes often recover faster when they do recover (Fig. 3, middle), yet fail more frequently at the largest shocks (Fig. 2, right). Thus "smooth beats kink" is not universal: **we prioritize (A) recovery success**—irreversible non-recovery dominates downstream performance—**and use (B) as a tie-breaker** among activations that recover. Therefore, we keep the strict floor and negative linear leak of the Leaky-ReLU family aiming for an empirical 'Goldilocks zone', and introduce smoothness only insofar as it *preserves* those two properties.

### 5.1 Smooth-Leaky and Randomized Smooth-Leaky

Guided by Sec. 5—(i) strict non-zero floor, (ii) moderate leak, (iii) prefer $C^1$ over $C^0$ when (i)–(ii) are held fixed—we introduce two drop-in rectifiers that keep capacity unchanged.

The **Smooth-Leaky** activation function (Fig. 5) is designed as a direct, $C^1$, drop-in substitute for Leaky ReLU that preserves the negative-side floor and the positive-side identity while removing the kink with a smooth, curved transition region. It is asymptotically linear ($f(x) \approx \alpha x$ for $x \ll 0$, $f(x) \approx x$ for $x \gg 0$) and controlled by a leak $\alpha$ plus a smooth transition set by $(p, c)$:

$$f(x) = \alpha x + (1 - \alpha)\, x \cdot \sigma\left(\frac{cx}{p}\right) \qquad (1)$$

where $\sigma$ is the sigmoid. Here, $\alpha$ fixes the negative-side floor, and $(p, c)$ set the width/steepness of the transition.

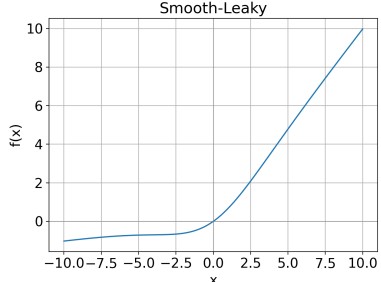

Figure 5: Smooth-Leaky with $\alpha$=0.1, $p$=3.0, $c$=5.0. Randomized Smooth-Leaky draws $\alpha$ from bounds.

To add lightweight exploration around a moderate leak we introduce **Randomized Smooth-Leaky** by replacing the fixed $\alpha$ with a random slope $r$ drawn uniformly from $[l, u]$ on each forward pass; at inference we fix $r$ to its mean $(l+u)/2$:

$$f(x) = r\, x + (1 - r)\, x\, \sigma\left(\frac{cx}{p}\right), \qquad r \sim \mathcal{U}(l, u), \quad r_{\text{test}} = \frac{l+u}{2}. \qquad (2)$$

This randomized variant preserves the strict floor and $C^1$ transition while encouraging robustness to small variations in negative-side responsiveness. Limitations of multi-parameter design and computational budget-fairness are explained in App. B.2.

## 6 Continual Supervised Learning

Following Kumar et al. (2023), we evaluate five supervised continual image-classification benchmarks spanning two shift types: input distribution shift (Permuted MNIST, 5+1 CIFAR, Continual ImageNet) and concept shift (Random Label MNIST, Random Label CIFAR). Training proceeds as a sequence of tasks without task-identity signals where the agent receives mini-batches for a fixed duration per task. **Permuted MNIST** (Goodfellow et al., 2013) applies a fixed random pixel permutation to a shared subset for each task. **Random Label MNIST** (Lyle et al., 2023) and **Random Label CIFAR** assign random labels to a fixed subset to encourage memorization. **CIFAR 5+1** draws and alternates hard (5 classes) and easy (single class) tasks from CIFAR-100. Evaluation focuses on hard tasks to stress plasticity loss mitigation. Finally, **Continual ImageNet** (Dohare et al., 2024; Russakovsky et al., 2015) performs a task-binary classification over two ImageNet classes which do not repeat across tasks, ensuring non-overlapping class exposure and clearer measurement of plasticity over time. An extended explanation of each benchmark problem is found in App. E.

| Activation | Permuted MNIST | Random Label MNIST | Random Label CIFAR | CIFAR 5+1 | Continual ImageNet |
|---|---|---|---|---|---|
| ReLU | 78.85 ± 0.06 | 20.03 ± 2.46 | 25.79 ± 6.18 | 4.76 ± 1.01 | 73.71 ± 0.43 |
| Leaky-ReLU | 84.14 ± 0.01 | 91.53 ± 0.18 | 98.34 ± 0.01 | 48.86 ± 0.70 | 85.28 ± 0.20 |
| Sigmoid | 76.96 ± 0.07 | 79.59 ± 0.75 | 52.24 ± 2.99 | 1.79 ± 0.19 | 63.89 ± 7.38 |
| Tanh | 70.32 ± 0.54 | 63.40 ± 0.12 | 58.56 ± 1.05 | 28.59 ± 2.34 | 70.97 ± 0.44 |
| RReLU | 83.95 ± 0.02 | 93.10 ± 0.02 | 98.02 ± 0.03 | 53.60 ± 1.06 | 84.97 ± 0.17 |
| PReLU | 82.62 ± 0.05 | 92.67 ± 0.23 | 96.86 ± 0.32 | 43.30 ± 0.61 | 82.37 ± 0.11 |
| Swish (SiLU) | 83.41 ± 0.03 | 67.73 ± 0.46 | 87.40 ± 2.42 | 35.31 ± 1.87 | 82.64 ± 0.99 |
| GeLU | 78.97 ± 0.09 | 38.79 ± 0.95 | 42.85 ± 2.12 | 17.60 ± 1.71 | 75.49 ± 0.11 |
| CeLU | 82.93 ± 0.04 | 37.16 ± 0.90 | 29.64 ± 10.44 | 54.23 ± 1.44 | 81.15 ± 0.68 |
| eLU | 80.50 ± 0.09 | 84.23 ± 0.70 | 57.45 ± 20.16 | 47.64 ± 1.44 | 80.10 ± 0.34 |
| SeLU | 80.43 ± 0.16 | 79.95 ± 0.91 | 84.61 ± 2.07 | 49.07 ± 1.25 | 80.98 ± 0.49 |
| CReLU | 82.66 ± 0.04 | 89.47 ± 0.28 | 92.90 ± 0.13 | 20.56 ± 2.28 | 84.85 ± 0.25 |
| Rational | 80.08 ± 0.05 | 92.35 ± 1.97 | 94.82 ± 0.75 | 40.41 ± 4.21 | 80.65 ± 0.38 |
| SwiGLU | 77.69 ± 0.26 | 31.20 ± 2.10 | 83.06 ± 3.51 | 9.57 ± 1.81 | 63.57 ± 2.04 |
| Deep Fourier | 83.69 ± 0.04 | 92.61 ± 0.04 | 96.24 ± 0.51 | **72.29 ± 2.11** | 76.03 ± 0.75 |
| Smooth-Leaky | 84.03 ± 0.02 | 91.69 ± 0.12 | 98.36 ± 0.00 | 49.87 ± 1.67 | 85.38 ± 0.25 |
| Rand. Smooth-Leaky | **84.26 ± 0.02** | **93.33 ± 0.05** | **98.42 ± 0.01** | 57.01 ± 1.59 | **86.23 ± 0.13** |

Table 2: Total Average Online Task Accuracy (%) on Continual Supervised Benchmarks, averaged over 5 independent runs. Values are reported as mean ± standard deviation (SD). Statistical significance between the top two performers in each column was determined using an independent two-sample Welch's t-test ($p < 0.05$). Rand. Smooth-Leaky is statistically significant with respect to the next best-performing activation (Smooth-Leaky). Smooth-Leaky is also significant compared to the next best in Rand. Label CIFAR, CIFAR 5+1 and Continual ImageNet.

Across the five continual benchmarks, we observe a clear pattern, reported in Tab. 2. First, *rectifiers with a learnable or randomized negative branch* dominate: Leaky-ReLU, RReLU, PReLU, Smooth-Leaky, and Rand. Smooth-Leaky consistently outperform ReLU—especially on the harder settings—while smooth rectifiers such as Swish/SiLU are competitive but typically trail the best leaky-family members. Second, we again observe the first reported *'Goldilocks zone'* for the negative branch (cf. Sec. 3): the strongest performers cluster around an initial/effective negative slope in the range $[0.6, 0.9]$ (including the mean of RReLU bounds and neuron-wise $\alpha$ in PReLU). We also evaluated CReLU and Rational activations (Abbas et al., 2023; Delfosse et al., 2021b), Deep Fourier Features (Lewandowski et al., 2024), which satisfies our criteria for a smooth, non-zero gradient floor, maintaining responsiveness via periodic oscillation rather than a fixed linear slope, and SwiGLU (Shazeer, 2020), a gated nonlinearity that that gained recent popularity due to improve Transformer feed-forward layers and widely adopted in PaLM and LLaMA (Touvron et al., 2023; Chowdhery et al., 2023). Constrained rationals were excluded, as previous work shows that they improve RL stability but *reduce plasticity* (Surdej et al., 2025), which is our main focus. In our experiments, CReLU, Rational and Deep Fourier outperform ReLU, but remain below other standard activations and our proposed variants (see Tab. E2 and Fig. E1).

## 7 CONTINUAL REINFORCEMENT LEARNING

Continual learning is particularly critical in reinforcement learning (RL), where non-stationarity arises not only from changes in the environment but also from the agent's evolving policy, which affects the data distribution even in fixed environments. This tight feedback loop between learning and data collection makes RL especially vulnerable to loss of plasticity, where neural networks become progressively less responsive to new experiences. Recent work has shown that deep RL agents suffer from a gradual decline in representational diversity and gradient signal quality as training progresses, leading to suboptimal adaptation in later stages of learning (Dohare et al., 2024; Abbas et al., 2023). Diagnosing and understanding this phenomenon in RL can be more challenging than supervised learning settings due to high variability in algorithmic design (e.g., model-based vs. model-free, use of replay buffers, off-policy dynamics) and the inherent stochasticity of agent-environment interactions. As a result, systematic demonstrations of plasticity loss in RL require carefully controlled protocols and extensive experimentation, a challenge that we approach from the perspective of activation functions and their influence on network adaptability. Activation functions

play a pivotal role in deep reinforcement learning. Although ReLU and Tanh remain the most widely used options (e.g., Mnih et al. (2013; 2015); Hessel et al. (2018)), both exhibit limitations that affect learning dynamics (Nauman et al., 2024). ReLU, as previously mentioned, is susceptible to issues such as the dormant neuron phenomenon (Sokar et al., 2023), loss of plasticity (Lyle et al., 2023; 2022), and overestimation when encountering out-of-distribution inputs (Ball et al., 2023). Tanh, commonly employed to constrain outputs within a fixed range, suffers from saturation at its extremes, leading to vanishing gradients that hinder efficient learning in deep networks (Pascanu et al., 2013).

Therefore, to investigate plasticity loss in a continual RL setting, we train a *single* PPO agent (Schulman et al., 2017) on a fixed and repeating sequence of four MuJoCo locomotion tasks using Gymnasium (Towers et al., 2024): `HalfCheetah-v4` → `Hopper-v4` → `Walker2d-v4` → `Ant-v4` → `HalfCheetah-v4` → `Hopper-v4` → ⋯. The agent cycles through this sequence three times, training for 1M timesteps per environment per cycle (total 12M). Episodes terminate early on invalid configurations following (Dohare et al., 2024) (e.g., falls for `Hopper`/`Walker2d`, unstable heights for `Ant`); `HalfCheetah` runs full-length. The policy and value networks share a two-layer MLP backbone (256 units each) that is updated across all tasks. For each environment we attach a lightweight input adapter (to map its observation space) and a task-specific output head (for its action space); both persist across cycles and continue learning when the environment reappears. We trained using the Adam optimizer. Hyperparameters are in App. B3.

| **Metric** | Swish | PReLU | Sigmoid | Rand. Smooth-Leaky | Smooth-Leaky |
|---|---|---|---|---|---|
| IQM $\pm$ 95% CI[†] | $0.3149 \pm 0.071$ | $0.2716 \pm 0.038$ | $0.3329 \pm 0.059$ | $\mathbf{0.3875 \pm 0.038}$ | $0.3305 \pm 0.037$ |

Table 3: Average Plasticity Score across 5 seeds (higher is better). We only report the top-performing activations. See Table F1 for a full comparison of all activations. [†] Values are reported as Min-Max Normalized IQM mean $\pm$ 95% CI half-width.

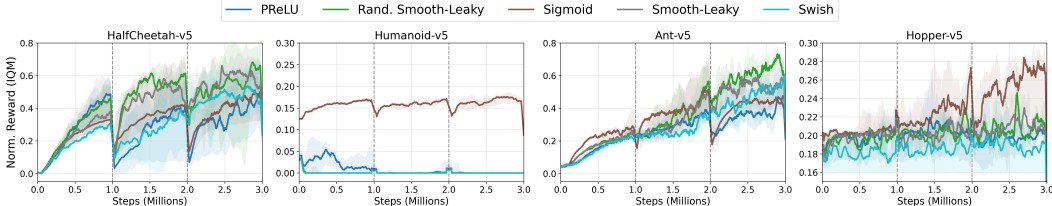

Figure 6: Plasticity Score across 5 seeds (95% bootstrap CIs) showing a complete sequence of 3 cycles across all 4 environments. The table 3 reports the Min-Max Normalized IQM of this score across seeds, only showing the top-performing activations to avoid clutter.

**Plasticity Score (definition).** To rigorously aggregate performance across environments with distinct reward scales, we employ a normalized scoring protocol following (Agarwal et al., 2021). For each run $r$ we compute the mean episodic return of the last cycle's steady-state (final 15% of steps). This value $R_{r,e}$ for environment $e$ is Min-Max normalized: $S_{r,e} = \frac{R_{r,e} - R_e^{\min}}{R_e^{\max} - R_e^{\min}}$. Global bounds $[R_e^{\min}, R_e^{\max}]$ are determined via a robust percentile sweep across all activations. To prevent physics simulation instabilities in failure cases (e.g., `Humanoid-v5`) from skewing the mean with unbounded negative values, we apply a *stability floor*: rewards below a functional failure threshold are clipped to the lower bound ($S_{r,e} = 0$). Finally, we report the *Interquartile Mean* (IQM) to provide a robust measure of expected plasticity. By filtering outliers while accounting for performance magnitude, the resulting **Plasticity Score** $\in [0, 1]$ reflects the agent's efficiency relative to the environment's theoretical maximum.

## 7.1 TRAINABILITY VS. GENERALIZABILITY

Anchoring *plasticity loss* to a single mechanism risks conflating cause and effect. We distinguish *trainability* (learning capacity on current data) from *generalizability* (transfer to unseen variations). Prior work (Berariu et al., 2021) distinguishes *trainability*—reduced ability to lower loss on new data (Dohare et al., 2021; Elsayed & Mahmood, 2024; Lyle et al., 2022)—from *generalizability*—reduced performance on unseen data (Ash & Adams, 2020; Zilly et al., 2021). Mitigating plasticity loss

should improve both, not merely memorize the latest data (Lee et al., 2024); nevertheless, their relationship remains unsettled, and terminology is often mixed (Klein et al., 2024). In brief: loss of trainability (no effective parameter updates) can *cause* loss of functional plasticity, but the reverse need not hold—models may still update yet fail to *transfer* gains.

This failure of plasticity has two faces in RL: (i) *train-side adaptation*—can the agent still improve on the data it now collects?—and (ii) *transfer to perturbed test conditions* (e.g., randomized MuJoCo friction (Dohare et al., 2024)). We report two complementary metrics. The **Plasticity Score** (IQM) summarizes late-cycle trainability. The **Generalization Gap** ($\Delta$GAP) measures how much the train–test gap widens over time (App. F). For each cycle $c$ and environment $e$, $\text{GAP}_{c,e} = R_{c,e}^{\text{train}} - R_{c,e}^{\text{test}}$, where $R$ is the expected return (measured at the end of cycle $c$). Thus $\Delta(\text{GAP}_e) = \text{GAP}_{3,e} - \text{GAP}_{1,e}$; larger $\Delta$ means the train–test gap *widened* over time (worse transfer). We aggregate $\Delta$ per activation using the **Interquartile Mean (IQM)** across environments. Crucially, these metrics must be analyzed together.

A high Plasticity Score is not preferable if it comes with a large, widening gap, just as a small gap is meaningless if absolute rewards are near zero (as seen in physics failures). Our two-metric system captures this tension. While `Rand. Smooth-Leaky` suffers from stability issues in `Humanoid` (zero rewards and thus a zero gap), in stable environments like `Ant` and `Cheetah` where it maximizes the Plasticity Score, it also attains a lower IQM $\Delta(\text{GAP}_e)$ compared to baselines (Tab. 4). This indicates that its trainability does not come at the cost of overfitting; rather, it encourages solutions that transfer well, provided the dynamics remain stable. In contrast, `Sigmoid` achieves a moderate Plasticity Score via safety—its bounded nature prevents divergence in `Humanoid`—yet it exhibits a larger widening of the gap in other tasks.

These metrics are complementary, not redundant. **Plasticity Score** asks "did the agent remain adaptable on the data it collected?", while $\Delta(\text{GAP}_e)$ asks "did that adaptation carry to perturbed tests?". Our activation designs primarily target sustained train-side plasticity (strict floor, moderate leak, $C^1$ transition)—a prerequisite for learning under shift. We report both metrics to surface, not mask, this open question.

| Activation | HalfCheetah | Humanoid | Ant | Hopper | $\Delta_{IQM}$ |
|---|---|---|---|---|---|
| **Swish (SiLU)** | $533.55 \pm 1314.41$ | $0.00 \pm 0.00$ | $780.79 \pm 730.11$ | $13.39 \pm 37.27$ | $273.47$ |
| **PReLU** | $839.60 \pm 596.03$ | $\mathbf{-316.28 \pm 2211.72}$ | $94.17 \pm 660.28$ | $-22.35 \pm 74.09$ | $35.91$ |
| **Sigmoid** | $\mathbf{-288.53 \pm 2576.85}$ | $18.92 \pm 111.38$ | $276.48 \pm 1018.48$ | $152.14 \pm 129.02$ | $85.53$ |
| **Smooth-Leaky** | $847.49 \pm 1611.57$ | $0.00 \pm 0.00$ | $44.09 \pm 1089.34$ | $27.01 \pm 72.58$ | $\mathbf{35.55}$ |
| **Rand. Smooth-Leaky** | $-49.38 \pm 863.55$ | $0.00 \pm 0.00$ | $\mathbf{-336.13 \pm 971.75}$ | $\mathbf{-68.68 \pm 96.31}$ | $\mathbf{59.03}$ |

Table 4: Change in Generalization Gap ($\Delta$GAP). We report the difference in generalization gap between the final and first cycles ($\Delta(\text{GAP}_e) = \text{GAP}_{3,e} - \text{GAP}_{1,e}$). Positive values indicate a *widening* gap (worse transfer) over time. Values are mean $\pm$ 95% CI. The rightmost column reports the **Interquartile Mean (IQM)** of these deltas across environments. *Note:* Runs exhibiting physics simulation failures (reward $< -500$) are treated as having a gap of $0.0$ (indicating no meaningful performance difference to overfit).

## 8 CONCLUSION AND FUTURE WORK

The choice of the activation function greatly impacts performance and plasticity in continual learning; it must be designed, not assumed. We do not claim universal wins, but our results show that first-principles activations mitigate plasticity loss and improve trainability. Building on these findings, future work will test interactions with other standard continual learning approaches such as experience replay and regularization, narrow the RL generalization gap by tracking train-side plasticity and test-time robustness, which will help clarify when improved trainability helps versus when it simply leads to overfitting on recent data, and move from fixed 'Goldilocks zone' slopes to adaptive, per-neuron self-tuning. We plan to derive a stronger theoretical understanding (curvature/desaturation bounds for smooth-leaky families) and scale to larger and more challenging CL domains while studying the interplay of activation design with optimizer and normalization effects, culminating in a principle-guided automated search for new robust activations.

ACKNOWLEDGMENTS

This material is based on work supported by the National Science Foundation under Grant No. 2218063 and 2239691. The authors acknowledge the Vermont Advanced Computing Center (VACC) at the University of Vermont for providing computational resources that have contributed to the research results reported in this paper.

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

# A  CHARACTERIZATION OF ACTIVATION FUNCTION PROPERTIES

| Activation | HDZ | NZG | Sat± | Sat− | $C^1$ | NonM | SelfN | $L/R_{slp}$ | $f''$ |
|---|---|---|---|---|---|---|---|---|---|
| ReLU (Nair & Hinton, 2010) | ✓ | – | – | ✓ | – | – | – | – | – |
| LeakyReLU (Maas et al., 2013) | – | ✓ | – | – | – | – | – | – | – |
| PReLU (He et al., 2015) | – | ✓ | – | – | – | – | – | ✓ | – |
| RReLU (Xu et al., 2015) | – | ✓ | – | – | – | – | – | ✓ | – |
| Sigmoid | – | ✓* | ✓ | ✓ | ✓ | – | – | – | ✓ |
| Tanh | – | ✓* | ✓ | ✓ | ✓ | – | – | – | ✓ |
| Swish (SiLU) (Ramachandran et al., 2017) | – | ✓ | – | – | ✓ | ✓ | – | – | ✓ |
| GeLU (Hendrycks & Gimpel, 2016) | – | ✓ | – | – | ✓ | ✓ | – | – | ✓ |
| ELU (Clevert et al., 2015) | – | ✓ | – | ✓ | ✓† | – | – | – | ✓ |
| CELU (Barron, 2017) | – | ✓ | – | ✓ | ✓ | – | – | – | ✓ |
| SELU (Klambauer et al., 2017) | – | ✓ | – | ✓ | –‡ | – | ✓♣ | – | ✓ |
| CReLU (Shang et al., 2016) | ✓* | ✓ | – | – | – | – | – | – | – |
| Rational□ (Delfosse et al., 2021b) | – | ✓ | – | – | ✓ | ✓ | – | –° | ✓ |
| SwiGLU◇ (Shazeer, 2020) | – | ✓ | – | – | ✓ | ✓ | – | – | ✓ |
| Deep Fourier(Lewandowski et al., 2024) | – | ✓ | – | – | ✓ | ✓ | – | – | ✓ |
| Smooth-Leaky△ | – | ✓ | – | – | ✓ | ✓◇ | – | – | ✓ |
| R-Smooth-Leaky△ | – | ✓ | – | – | ✓ | ✓◇ | – | ✓ | ✓ |

Table A1: Binary property grid (✓ = present, – = absent). **Abbreviations.** HDZ: hard dead zone; NZG: non-zero gradient for $x < 0$; Sat±: two-sided saturation; Sat−: negative-side saturation; $C^1$: first derivative continuous; NonM: non-monotonic segment; SelfN: self-normalizing output; $L/R_{slp}$: learnable or randomized slope; $f''$: non-zero second derivative.

*Gradients are small but non-zero except at extreme inputs (risk of effective inactivity via saturation). † ELU is $C^1$ only for $\alpha = 1$. ‡ SELU has a small derivative jump at $x = 0$ because $\alpha \neq 1$. ♣ SELU's self-normalizing behavior holds under the prescribed $(\lambda, \alpha)$. □ Unconstrained activations are smooth, non-monotonic, and non-saturating; outputs can grow large under training. ° Rational do not have "slopes" like Leaky-ReLU/PReLU; they learn polynomial coefficients. * Each branch retains the ReLU dead zone, but concatenation ensures that for any input at least one branch is active, avoiding global inactivity. ◇ SwiGLU is a gating mechanism (acting on a vector split) rather than a scalar activation $\phi(z)$. While it lacks a static hard dead zone based on input sign, the multiplicative gating creates a *dynamic* dead zone: if the gate branch saturates to zero, gradients for the value branch vanish. △ Proposed in this work. ◇ Potentially non-monotonic depending on the choice of slope parameter (and on the randomization bounds for R-Smooth-Leaky).

# B  EXPERIMENTAL AND ARCHITECTURAL DESIGN

Models for both Case Studies 3 and 4 share the same backbone ($4 \times 32$ conv + 256-unit MLP) and are trained on the 20-task Split-CIFAR100 benchmark; Kaiming initialization uses the correct gain for each starting slope ($\alpha_0 = 0.25$ for every PReLU). Metrics recorded every epoch include final average accuracy (ACC), online forgetting, dead unit fraction, gradient signal-to-noise, $\lambda_{\max}$ for the fc1→fc2 block, and— for PReLU—the full trajectory $\alpha_i(t)$.

For *continual supervised learning* we deliberately use compact networks to accentuate capacity–limited plasticity loss: a model may attain high average online accuracy on a few tasks, yet its ability to adapt degrades over long sequences. We employ two backbones. (i) **MLP:** two hidden layers, each of width 100. (ii) **CNN:** two $5 \times 5$ convolutional layers with 16 output channels each, every conv followed by $2 \times 2$ max pooling, then two fully connected layers of width 100. All architectures end with a linear classifier whose output size is 10 for Permuted MNIST, Random Label MNIST, and Random Label CIFAR; 100 for 5+1 CIFAR; and 2 for Continual ImageNet.

For *continual RL* (Sec. 7), policy and value functions share a multi–head MLP designed for sequential adaptation: a shared backbone with two hidden layers of 256 units is trained across all tasks, while for each environment we instantiate a dedicated input adapter that maps its observation space to the backbone and dedicated output head(s) for its action (and value) space. This modular design reuses core features while accommodating heterogeneous state–action spaces.

### B.1 LIMITATIONS AND IMPLICATIONS OF CHOOSING AN OPTIMIZER.

A valid question is how our results might interact with different optimizers (e.g., SGD, RMSProp) beyond Adam. Our study's primary goal was to isolate the effect of activation functions on plasticity, which required fixing other key variables to create a controlled experiment. We intentionally selected the Adam optimizer for two main reasons:

- **Consistency with Prior Work**: Our continual supervised learning benchmarks are adapted from recent literature(Kumar et al., 2023), which used Adam for their experiments. Anchoring our design to these established baselines allows for more direct comparisons.
- **A "Stress Test" for Plasticity**: More importantly, adaptive optimizers like Adam have been specifically implicated in the literature as a contributing factor to plasticity loss. Prior work has suggested that Adam can struggle to update large-magnitude weights (Dohare et al., 2021) and has been shown to lose its ability to learn on non-stationary tasks (Lyle et al., 2023).

We reasoned that if we wanted to study how activation functions mitigate plasticity loss, we should run our experiments in a setting known to induce it. By demonstrating that our activation-design principles can sustain plasticity even when paired with an optimizer known to exacerbate the problem, we provide a more robust finding. We do not claim this is the optimal optimizer-activation pair for SOTA results, but rather that activation design is a fundamental mechanism for preserving plasticity, even under these challenging and fixed conditions.

### B.2 LIMITATIONS AND IMPLICATIONS OF MULTI-PARAMETER ACTIVATION DESIGN.

Smooth-Leaky and Randomized Smooth-Leaky expose several shape-controlling hyperparameters. This enlarges the tuning space—and thus the cost—but also grants useful control, as reflected in our benchmarks. A common response is to replace sweeps with adaptive or learnable parameters. However, as shown in Section 3, adaptive, granular learnable slopes (e.g., PReLU) underperformed in all our settings, while bounded, stochastic constraints (e.g., RReLU) worked better. In continual learning, where short-term objectives dominate, unconstrained learned parameters can drift to task-local optima that hurt performance across tasks. Attempts to automate shape learning (Bingham & Miikkulainen, 2023; Bingham et al., 2020; Manessi & Rozza, 2018) with highly expressive forms (e.g., Rational activations (Delfosse et al., 2020)) also lagged behind our first-principles designs here. This suggests that we have not yet reconciled expressivity with robust automation. A promising direction is to rethink where and on what timescale hyperparameters are adapted—potentially decoupling their updates from the main training loop to better handle non-stationarity.

**Budget-matched robustness and computational fairness.** Our proposed activation family admits a larger hyperparameter grid than some baselines (because it has more tunable parameters), potentially inflating performance by enabling a broader search. To address this, we evaluate *budget-matched robustness* rather than enforcing an arbitrary equality in the raw number of tested settings across methods. For each activation $i$ and dataset, we define a tuning budget $N_i$ as the number of evaluated hyperparameter configurations for $i$ (Table B2). We then simulate a practitioner with budget $N_i$ who can try $N_i$ randomly chosen configurations and keep the best result: we bootstrap a *best-of-$N_i$* distribution by repeatedly sampling $N_i$ configurations (with replacement) and taking the maximum accuracy. We perform a one-sided Mann–Whitney U test with alternative

$$H_1 : \text{BestOf}_{N_i}(\text{Rand Smooth-Leaky}) > \text{BestOf}_{N_i}(i),$$

and report the resulting p-values in Table B1. Under this formulation, p-values close to 0 (reported as $< 10^{-4}$ if values are smaller than such for simplicity) indicate that Rand. Smooth-Leaky yields consistently higher best-of-$N_i$ outcomes than activation $i$ when both are given the *same* number of random hyperparameter trials, while p-values closer to 1 indicate the opposite direction.

The results show that Rand Smooth-Leaky remains competitive under matched budgets across a wide range of baselines and datasets. In particular, for PERMUTED-MNIST, RANDOM LABEL MNIST, RANDOM LABEL CIFAR, and CONTINUAL IMAGENET, Rand Smooth-Leaky significantly dominates most baselines under their own budgets (e.g., ReLU, Sigmoid, Tanh, PReLU, Swish/SiLU, GeLU, eLU, SeLU, CReLU, Rational, and SwiGLU yield $p < 10^{-4}$ in nearly all cases). This

| Activations | Permuted MNIST | Random Label MNIST | Random Label CIFAR | CIFAR 5+1 | Continual ImageNet |
|---|---|---|---|---|---|
| ReLU | $< 10^{-4}$ | $< 10^{-4}$ | $< 10^{-4}$ | $< 10^{-4}$ | $< 10^{-4}$ |
| Leaky-ReLU | 1.00 | 0.0005 | 0.5820 | 0.0004 | $< 10^{-4}$ |
| Sigmoid | $< 10^{-4}$ | $< 10^{-4}$ | $< 10^{-4}$ | $< 10^{-4}$ | $< 10^{-4}$ |
| Tanh | $< 10^{-4}$ | $< 10^{-4}$ | $< 10^{-4}$ | $< 10^{-4}$ | $< 10^{-4}$ |
| RReLU | 0.0310 | 0.471 | 0.0118 | 1.00 | $< 10^{-4}$ |
| PReLU | $< 10^{-4}$ | 0.071 | $< 10^{-4}$ | $< 10^{-4}$ | $< 10^{-4}$ |
| Swish (SiLU) | $< 10^{-4}$ | $< 10^{-4}$ | $< 10^{-4}$ | $< 10^{-4}$ | $< 10^{-4}$ |
| GeLU | $< 10^{-4}$ | $< 10^{-4}$ | $< 10^{-4}$ | $< 10^{-4}$ | $< 10^{-4}$ |
| CeLU | $< 10^{-4}$ | $< 10^{-4}$ | $< 10^{-4}$ | 1.00 | $< 10^{-4}$ |
| eLU | $< 10^{-4}$ | $< 10^{-4}$ | $< 10^{-4}$ | 0.62 | $< 10^{-4}$ |
| SeLU | $< 10^{-4}$ | $< 10^{-4}$ | $< 10^{-4}$ | 0.37 | $< 10^{-4}$ |
| CReLU | 0.0008 | 0.035 | $< 10^{-4}$ | $< 10^{-4}$ | $< 10^{-4}$ |
| Rational | 0.28 | $< 10^{-4}$ | $< 10^{-4}$ | $< 10^{-4}$ | $< 10^{-4}$ |
| SwiGLU | $< 10^{-4}$ | $< 10^{-4}$ | $< 10^{-4}$ | $< 10^{-4}$ | $< 10^{-4}$ |
| DeepFourier | 1.00 | $< 10^{-4}$ | 0.77 | $< 10^{-4}$ | $< 10^{-4}$ |
| Smooth-Leaky | 1.00 | $< 10^{-4}$ | 1.00 | $< 10^{-4}$ | 1.00 |

Table B1: **Budget-fairness comparison significance test.** For each baseline activation $i$, combined learning rates, and dataset, let $N_i$ be the number of evaluated hyperparameter configurations for $i$ (Table B2). We bootstrap best-of-$N_i$ performance by repeatedly sampling $N_i$ configurations (with replacement) and taking the maximum accuracy. We then perform a one-sided Mann–Whitney U test with alternative $H_1 : \text{BestOf}_{N_i}(\text{Rand Smooth-Leaky}) > \text{BestOf}_{N_i}(i)$. Reported values are p-values. We perform 50 bootstrap repetitions.

indicates that the strong performance of Rand Smooth-Leaky is not contingent on conducting an unusually large, constrained hyperparameter search: even when restricted to the same tuning budget as each baseline (typically $N_i \in \{2, 36, 40, 62, 64\}$ depending on the activation), random budgeted sampling is sufficient to recover high-performing configurations with high probability.

Table B1 also makes the failure modes explicit via the directionality of the one-sided test. DeepFourier yields p-values of 1.00 or close to 1.00 on several datasets (e.g., PERMUTED-MNIST, and CIFAR 5+1), indicating that under its small budget ($N_i = 2$) its best-of-$N_i$ can exceed Rand Smooth-Leaky's best-of-$N_i$ in those settings; conversely, Rand Smooth-Leaky significantly exceeds DeepFourier on RANDOM LABEL MNIST, RANDOM LABEL CIFAR, and CONTINUAL IMAGENET ($p < 10^{-4}$). Similarly, Smooth-Leaky exhibits a mixed pattern: Rand Smooth-Leaky is not higher on PERMUTED-MNIST, RANDOM LABEL CIFAR, and CONTINUAL IMAGENET (p-values 1.00), but it is significantly higher on RANDOM LABEL MNIST and CIFAR 5+1 ($p < 10^{-4}$). However, this suggests the strength of the other activation function introduced: Smooth-Leaky. Thus, making such comparison between them independently good to whichever is better. Finally, CeLU shows a single notable exception on CIFAR 5+1 (p-value 1.00) while being dominated elsewhere, illustrating that dataset-specific interactions can invert the relative best-of-budget ordering. While other activation functions are close but not statistically significant in some datasets.

Overall, these budget-matched tests for computational fairness directly answer the concern: our conclusions do not rely on matching the raw number of hyperparameter combinations across methods (which is not meaningful once baselines plateau within their own hyperparameter ranges). Instead, by evaluating best-of-$N_i$ under matched budgets derived from each baseline's sweep size, we show that Rand Smooth-Leaky is typically robust to hyperparameter choice and achieves high performance even under limited random search budgets, while transparently identifying the few dataset/activation pairs where this dominance does not hold.

## B.3 CReLU INTEGRATION.

To insert CReLU without changing the hidden width or the rest of the network, we follow a capacity–neutral design: for any hidden layer with target width $H$, the preceding linear layer produces $H/2$ pre–activations $z \in \mathbb{R}^{H/2}$, and we apply $\text{CReLU}(z) = \big[\max(z, 0),\, \max(-z, 0)\big] \in \mathbb{R}^H$, which

| Activations | Permuted MNIST | Random Label MNIST | Random Label CIFAR | CIFAR 5+1 | Continual ImageNet |
|---|---|---|---|---|---|
| CeLU | 64 | 64 | 64 | 64 | 64 |
| CReLU | 2 | 2 | 2 | 2 | 2 |
| DeepFourier | 2 | 2 | 2 | 2 | 2 |
| eLU | 64 | 64 | 64 | 64 | 64 |
| GeLU | 64 | 64 | 64 | 64 | 64 |
| Leaky-ReLU | 64 | 64 | 64 | 64 | 64 |
| PReLU | 36 | 36 | 36 | 36 | 36 |
| Rational | 20 | 26 | 12 | 41 | 14 |
| ReLU | 2 | 2 | 2 | 2 | 2 |
| RReLU | 40 | 40 | 40 | 40 | 40 |
| SeLU | 62 | 62 | 62 | 62 | 62 |
| Sigmoid | 2 | 2 | 2 | 2 | 2 |
| Smooth-Leaky | 175 | 175 | 175 | 175 | 175 |
| SwiGLU | 2 | 2 | 2 | 2 | 2 |
| Swish (SiLU) | 64 | 64 | 64 | 64 | 64 |
| Tanh | 2 | 2 | 2 | 2 | 2 |

Table B2: **Configuration counts for budget fairness comparison.** $N_i$ denotes the number of successfully evaluated hyperparameter configurations for activation $i$ on each dataset. These values define the tuning budget used in Table B1.

restores the width to $H$ by concatenation. This mirrors the "CReLU (+half)" configuration: the parameter count of the producer linear is halved (input$\times H/2$ instead of input$\times H$), the consumer linear still receives $H$ features, and the forward shape everywhere else is identical to the ReLU baseline. Thus CReLU's representational benefit (explicit positive/negative phase features) is preserved while keeping model size and interface unchanged, enabling plug–in replacement of the activation without branching logic in the forward pass.

For convolutional blocks with target channel width $C$ and fully connected blocks with target width $H$, we apply CReLU in a capacity–neutral manner: the producer layer outputs $C/2$ (or $H/2$) pre–activations $z$, and we apply $\mathrm{CReLU}(z) = [\max(z, 0),\ \max(-z, 0)]$ along the channel/feature dimension, restoring the width to $C$ (or $H$) by concatenation. This mirrors the "CReLU (+half)" configuration, preserving parameter count relative to a ReLU baseline while exposing explicit positive/negative phase features. Subsequent layers therefore see the same interface shapes as in the baseline, allowing CReLU to be swapped in without modifying the forward pass.

## B.4 Hyper-parameter Sweeps for Activation Functions

| Activation | Symbol(s) | Values explored |
|---|---|---|
| ReLU / CReLU / Deep Fourier | $\alpha$ | 0.0 |
| Leaky-ReLU \| Swish / GeLU | $\alpha \mid \beta$ | 0.01, 0.05, 0.1, 0.2, 0.3, 0.4, 0.5, 0.6, 0.7, 0.8, 0.9, 1.0, 1.05, 1.1, 1.2, 1.3, 1.4, 1.5, 1.6, 1.7, 1.8, 1.9, 2.0, 2.1, 2.2, 2.3, 2.4, 2.5, 2.6, 2.7, 2.8, 2.9, 3.0 |
| ELU / CELU | $\alpha$ | 0.1, 0.5, 1.0, 1.5, 2.0, 2.2, 2.4, 2.6, 2.8, 3.0, 3.3, 3.5, 3.6, 3.7, 3.8, 3.9, 4.0, 4.1, 4.2, 4.3, 4.4, 4.5, 4.6, 4.7, 4.8, 4.9, 5.0, 5.1, 5.2, 5.3, 5.4, 5.5 |
| SELU | $\alpha$ | 1.0, 1.3, 1.673, 2.0, 2.3, 2.4, 2.6, 2.8, 3.0, 3.1, 3.3, 3.5, 3.7, 3.8, 3.9, 4.0, 4.1, 4.2, 4.3, 4.4, 4.5, 4.6, 4.7, 4.8, 4.9, 5.0, 5.1, 5.2, 5.3, 5.4, 5.5 |
| RReLU | $U = [l, u]$ | [0.01, 0.05], [0.05, 0.10], [0.1, 0.3], [0.125, 0.333], [0.3, 1.0], [0.4, 1.0], [0.5, 1.0], [0.6, 1.0], [0.6, 0.8], [0.7, 1.0], [0.8, 1.0], [0.9, 1.0], [1.0, 1.5], [1.6732, 1.6732], [1.4232, 1.9232], [1.168, 2.178], [0.9232, 2.4232], [1.548, 1.798], [0.673, 2.673], [0.423, 2.923] |
| PReLU | scope | global, layer, neuron |

*Continued on next page...*

| Activation | Symbol(s) | Values explored (Cont.) |
|---|---|---|
| Sigmoid / Tanh / SwiGLU / DFF | – | – |
| Rational | P(x), Q(x), V, $A_f$ | Grid over: $(P(x), Q(x)) \in \{(7,6), (5,4), (3,2)\}$ $V \in \{A, B, C, D\}$ $A_f \in \{\text{ReLU, Leaky-ReLU, Swish, Tanh, Sigmoid}\}$ |
| Smooth-Leaky$^\triangle$ | $(c, p, \alpha)$ | Grid over: $c, p \in \{1, 2, 3, 4, 5\}$ $\alpha \in \{0.1, 0.3, 0.5, 0.65, 0.7, 0.8, 0.9\}$ (total 175 combos). |
| Rand-Smooth-Leaky$^\triangle$ | $(c, p, l, u)$ | Grid over: $c, p \in \{1, 2, 3, 4, 5\}$ lower bound $l \in \{0.01, 0.3, 0.4, 0.5, 0.6, 0.7\}$ upper bound $u \in \{0.05, 0.8, 1.0\}$ (total 450 combos); $r \sim \mathcal{U}(l, u)$ sampled per element during training. |

Table B3: A summary of the hyper-parameter sweeps performed for each activation function. The table details the specific values, ranges, and distributions tested for the corresponding symbols during our experiments. Best-performing hyperparameters are used to compute the results in Sections 3 and 4. $^\triangle$ Proposed in this work. See Section 5 and App. G.

The hyperparameter sweep of the activation function for continual supervised problems and continual RL is similar to those presented in Table B3 with the exception of Smooth-Leaky and Ran. Smooth-Leaky where we will add new values to the shape-controlling variables $c$ and $p$, as well as new upper ($u$) and lower $l$ bounds. Therefore, the full list of variables will be: $c, p \in \{0.1, 0.3, 0.5, 0.8, 1, 2, 3, 4, 5\}$, and the new bounds are $[l, u] = [0.673, 2.673], [0.55, 1.0], [0.05, 0.70], [0.01, 0.02], [0.01, 0.10]$, which were used to test a wider range of possibilities outside of intuitive bounds. We also introduce two activations functions presented in previous works as mentioned earlier: CReLU and Rational Activations. CReLU, as a concatenated ReLU, only has the original negative value $\alpha$ of 0. On the other hand, Rational Activations are defined by the polynomial of the numerator P(x), denominator Q(x), the rational version (V) used (e.g., A, B, C, or D), and the function ($A_f$) that is trying to approximate (e.g., ReLU, Swish, etc).

## B.5 I.I.D. VS CLASS-INCREMENTAL CONTINUAL LEARNING COMPARISON HYPERPARAMETERS

| Activation | i.i.d. | | CL | |
|---|---|---|---|---|
| | $HP$ | $LR$ | $HP$ | $LR$ |
| ReLU | – | 0.001 | – | 0.0001 |
| Leaky-ReLU | 0.01 | 0.001 | 0.7 | 0.0003 |
| RReLU | (0.125, 0.333) | 0.001 | (0.673, 2.673) | 0.0003 |
| PReLU | layer | 0.001 | neuron | 0.001 |
| Swish | 1.0 | 0.001 | 0.05 | 0.0001 |
| GeLU | 1.0 | 0.001 | 0.05 | 0.001 |
| CeLU | 1.0 | 0.001 | 2.4 | 0.001 |
| eLU | 1.5 | 0.001 | 3.9 | 0.0001 |
| SeLU | 1.673 | 0.001 | 3.7 | 0.0001 |
| Tanh | – | 0.001 | – | 0.0001 |
| Sigmoid | – | 0.001 | – | 0.001 |

Table B4: Optimal Hyperparameters ($HP$) and Learning Rate ($LR$) on Split-CIFAR100. A dash ($-$) indicates that such activation function uses the unique or baseline parameter (e.g., ReLU does not have any shape-controlling parameter since is linear on the positive right side and 0 on the negative left side).

Following the grid search over the hyper-parameters listed in Table B3, we chose the best-performing configuration for each activation function presented in Table B4. Each continual-learning run comprised 100 epochs across 20 tasks (100 classes in total, five classes per task). We adopted a standard Experience Replay (ER) framework

with a random-sampling buffer of size $|M| = 10{,}000$, capped at 500 examples per task. The larger-than-usual buffer was intended to mitigate catastrophic forgetting and ensure consistent performance across activation functions, allowing us to isolate the effects of reduced plasticity (following the intuition presented in (Dohare et al., 2024)).

### B.6 UNDERSTANDING CLASS-INCREMENTAL CONTINUAL LEARNING METRICS

To properly understand the choice of metrics throughout this paper, we first need to define and explain some of the classical metrics used in continual learning—*Final Average Accuracy (ACC)*, *Backward Transfer (BWT)*, and *Forward Transfer (FWT)*—and relate them to *loss of plasticity*. Consider a learning process unfolding over $N+1$ sequential *phases/tasks* $t = 0, \ldots, N$. Between consecutive phases $t-1$ and $t$ ($t \geq 1$), the data distribution, environment dynamics, reward signal, or task objective may change, inducing non-stationarity; we denote the distribution at phase $t$ by $E_t$. The model enters phase $t$ with parameters $\theta_{t-1}$ and is updated for $T_{\text{phase}}$ adaptation steps[4], producing intermediate iterates $\theta_{t,k}$ ($k = 1, \ldots, T_{\text{phase}}$) and terminating at $\theta_t = \theta_{t,T_{\text{phase}}}$.

Then: (i) $\text{ACC}_T = \frac{1}{T} \sum_{i=1}^{T} A_{T,i}$ summarizes performance after learning $T$ tasks but mixes current-task performance, retention of past tasks, and interference; (ii) $\text{BWT}_T = \frac{1}{T-1} \sum_{i=1}^{T-1} \left( A_{T,i} - A_{i,i} \right)$ quantifies forgetting/retention; and (iii) $\text{FWT}_T$ compares pre-training performance on future tasks to a suitable baseline, capturing positive/negative transfer. Although $\text{BWT}_T$ and $\text{FWT}_T$ are widely reported in continual learning, in our setting we focus on performance metrics related to $\text{ACC}_T$ following recent work (Kumar et al., 2023). We therefore briefly outline their differences and use-cases.

Because $\text{ACC}_T$ averages over tasks, it cannot isolate *plasticity*—fast adaptation within the *current* phase. In our class-incremental continual learning setup with experience replay (Sec. 2; (Van de Ven et al., 2022)), we report $\text{ACC}_T$ for overall performance. For the rest of our continual supervised learning benchmarks we shift to a more plasticity-focused evaluation and report lifetime aggregate of per-task *Average Online Task Accuracy* named *Total Average Online Accuracy* which track within-phase learning and, when task difficulty is comparable, reveal trends of *loss of plasticity*.

#### B.6.1 DIFFERENT ACCURACY METRICS ACROSS CONTINUAL LEARNING

**Average accuracy across seen tasks ($\text{ACC}_T$).** This is the standard average accuracy used in (offline or online) class-incremental CL to summarize performance after training up to task $T$. Let $\text{acc}_i^{(T)}$ be the test accuracy on task $i$ *measured after finishing training on task $T$*. Then

$$\text{ACC}_T = \frac{1}{T} \sum_{i=1}^{T} \text{acc}_i^{(T)}. \tag{3}$$

High $\text{ACC}_T$ implies some combination of plasticity (learning new tasks) and stability (retaining old ones). However, $\text{ACC}_T$ **mixes** current-task performance, retention on past tasks, and the interference caused by learning the current task, so it does not isolate plasticity on the most recent shift. In our Case Studies 3–4, which use an experience-replay buffer, we report this metric as final accuracy.

**Average online task accuracy (per task).** To quantify adaptation on a single task independently of past-task test performance, we use the mean online accuracy over the batches of task $T_i$. Let $M_i$ be the number of mini-batches in task $i$ and let $a_{i,j}$ denote the online accuracy on batch $j$ of task $i$. Define

$$\text{AOA}(T_i) = \frac{1}{M_i} \sum_{j=0}^{M_i-1} a_{i,j}. \tag{4}$$

This captures how quickly the agent learns the current task (plasticity). A downward trend of $\text{AOA}(T_i)$ across tasks of equal difficulty indicates *plasticity loss*.

**Total average online accuracy (lifetime).** For optimal hyper-parameter selection we also aggregate online accuracy over *all* batches seen so far (common in online CL (Cai et al., 2021; Ghunaim et al., 2023; Prabhu et al., 2023; Kumar et al., 2023)). Let $B_{\leq T} = \sum_{i=1}^{T} M_i$ be the total number of processed batches up to task $T$, and let $a_t$ be the online accuracy at global batch index $t$. Then

$$\text{TAOA}_{\leq T} = \frac{1}{B_{\leq T}} \sum_{t=0}^{B_{\leq T}-1} a_t = \frac{1}{\sum_{i=1}^{T} M_i} \sum_{i=1}^{T} \sum_{j=0}^{M_i-1} a_{i,j}. \tag{5}$$

---

[4]"Steps" may be epochs, mini-batches, or timesteps, depending on the domain.

If all tasks have equal length $M_i \equiv M$, this reduces to

$$\text{TAOA}_{\leq T} = \frac{1}{MT} \sum_{t=0}^{MT-1} a_t. \tag{6}$$

We distinguish this lifetime aggregate from the per-task quantity $\text{AOA}(T_i)$. This metric is reported in Section 6.

### B.7 CURVATURE METRICS

In order to study the properties of the curvature of a neural network and how it affects the loss of plasticity, we need to work with the Hessian matrix. For a loss function $L(\theta)$ (where $\theta$ represents all the parameters of the network), the Hessian matrix $H$ is defined as:

$$H = \nabla^2 L(\theta)$$

This is a symmetric matrix that captures the second-order derivatives (curvature) of the loss function with respect to the parameters.

#### B.7.1 PRINCIPAL CURVATURE

The principal curvature is defined as the maximum eigenvalue $\lambda_{\max}$ of the Hessian $H$. In other words, it's the largest value in the set of eigenvalues $\{\lambda_1, \lambda_2, \ldots, \lambda_d\}$ where $d$ is the number of parameters (or the dimension of $H$). The largest eigenvalue indicates the steepest direction of curvature. If you move in the direction of the corresponding eigenvector (the principal direction), the loss increases most rapidly. We employ the Power iteration algorithm to find the dominant eigenvector, and thereby the dominant eigenvalue using the Rayleigh Quotient:

$$\lambda_{\max} \approx v_k^\top H v_k$$

The principal eigenvector is the direction in parameter space corresponding to $\lambda_{\max}$. This is the steepest direction—any movement along that eigenvector direction leads to the largest second-order change in the loss.

#### B.7.2 EFFECTIVE RANK

To quantify the intrinsic dimensionality of the loss-landscape curvature in a given layer, we collect per-sample gradients $\{g_i\}_{i=1}^m \subset \mathbb{R}^d$ and stack them into a matrix $G = [g_1, \ldots, g_m] \in \mathbb{R}^{d \times m}$. We then form the Gram matrix

$$M = G^\top G \in \mathbb{R}^{m \times m},$$

compute its singular values $\sigma_1 \geq \sigma_2 \geq \cdots \geq \sigma_m \geq 0$, and define the *effective rank* at threshold $\tau \in (0, 1)$ by

$$\text{erank}_\tau(G) = \min\left\{ k : \frac{\sum_{j=1}^k \sigma_j}{\sum_{j=1}^m \sigma_j} \geq \tau \right\}.$$

Effective Rank represents how many independent directions of principal directions that account for at least a fraction $\tau$ of the total "energy" in the gradient subspace (Kumar et al., 2020). Therefore, effective rank tells you the number of eigenvectors (or directions) that are necessary to capture a specified fraction (for example, 99%) of the total curvature energy of the Hessian. In other words, it indicates how many independent directions contribute significantly to the curvature. If the effective rank is low, then most of the curvature is concentrated in just a few directions; if it's high, then the curvature is spread out over many directions.

## C EXPANDING ON CASE STUDY 1: NEGATIVE-SLOPE 'GOLDILOCKS ZONE'

### C.1 FAILURE MODES FOR SLOPE MAGNITUDE.

We continue investigating the primary drivers of functional plasticity loss. As provided in the main section results, the correlation between Final Average Accuracy (ACC) and dead unit fraction remains robust and significant (Pearson $r = -0.51$, $p = 8.2 \times 10^{-28}$), as seen in Fig. C1, right. In contrast, the correlation between ACC and final scaled gradient norms is negligible (Pearson $r = -0.11$, $p = 0.029$) (Fig. C1, left). Settings that maintain dead unit fraction below approximately 8% consistently achieve high ACC (27–33%), whereas dead unit rates above 25% lead to ACC collapsing to 20% or lower. This indicates that while gradient magnitude is a factor, the rise in dead units (gradient starvation) is a more direct and reliable indicator of plasticity loss than the raw magnitude of gradients, which can be small in well-adapted, flat minima or large in failing networks with high curvature. The fact that dead units do not account for all performance variance aligns with our finding in Section 3 that landscape curvature drives failure at high slopes, distinct from unit death. The results highlight that plasticity is not about maximizing any single statistic but about achieving a delicate balance, primarily tuned by the negative slope characteristics, to ensure unit survival while maintaining a benign loss landscape.

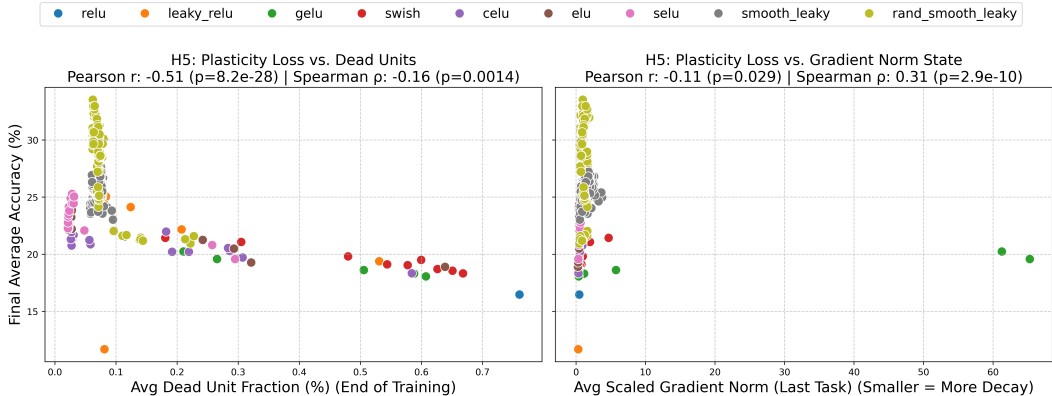

Figure C1: Person (r) correlation between Plasticity Loss and diverse metrics to evaluate the primary drivers of such phenomenon for some activation functions evaluated across Case Study 1 (see Section 3) **Left:** Dead Units; **Right:** Average Gradient Norm.

## C.2 ADAPTIVE, GRANULAR SLOPES ARE USEFUL—BUT NEED GUIDANCE TO STAY IN-BAND

As seen in Sec. 3, PReLU-N, with its per-neuron learnable $\alpha_i$, demonstrates this adaptability. Fig. C2 (left) shows the dynamic evolution over time of how PReLU-N learns heterogeneous $\alpha_i$ values, ideally we will observe many neurons dynamically adjusting their slopes into a beneficial range (e.g., 0.3-0.6, approaching the Goldilocks zone), while others can remain near zero if required by their local input statistics. However, our PReLU-N configuration (with default initialization $\alpha = 0.25$) undershot the empirically optimal $\alpha \approx 0.7$ for fixed Leaky-ReLU but its performance surpassed PReLU-G/L which tended towards slope decay and lower ACC ($\alpha$ evolution for PReLU-G/L shown in Figs. C3). All PReLU scopes (Neuron, Layer, Global) evolve towards values outside the Leaky-ReLU-family 'Goldilocks Zone'.

While not optimal, this might suggest that neuron-wise adaptability is a valuable trait, potentially enhanced further with optimized initialization or per-parameter learning rates if the learning can be guided towards the 'Goldilocks zone'. Which also opens up the question of properly finding such zone for different experimental settings.

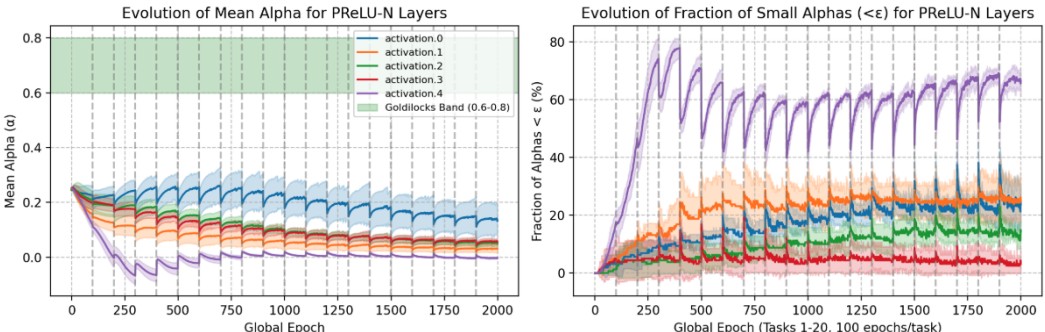

Figure C2: PReLU-N per-neuron learnable $\alpha$. **Left:** Distribution of all individual neurons' $\alpha$ per layer (activation.N) and the pre-defined 'Goldilocks zone' representing the best-performing $\alpha$ value in Leaky-ReLU. **Right:** Fraction of $\alpha$'s $< \epsilon$ indicating values for which the post-activation will be 'saturated' (too small to produce significant changes).

## C.3 SATURATION FRACTION AS METRIC FOR 'DEAD UNITS'.

Regarding Saturation Fraction used in Fig. 1 we acknowledge that recent works, such as (Sokar et al., 2023) or (Liu et al., 2025), highlighted the 'dormant neuron phenomenon', and 'GraMa' (Gradient Magnitude Neural Activity Metric) in deep reinforcement learning, proposing a valuable metric based on a neuron's average absolute activation (or gradient) relative to its layer-neuron connections to identify consistently underutilized units. While this provides a generalized approach to 'dead units', our investigation into a diverse array of activation functions—each with unique output ranges and saturation characteristics (e.g., ReLU's unbounded positive output versus Tanh's strict [-1, 1] bounds, or ELU's negative saturation plateau)—necessitated a more

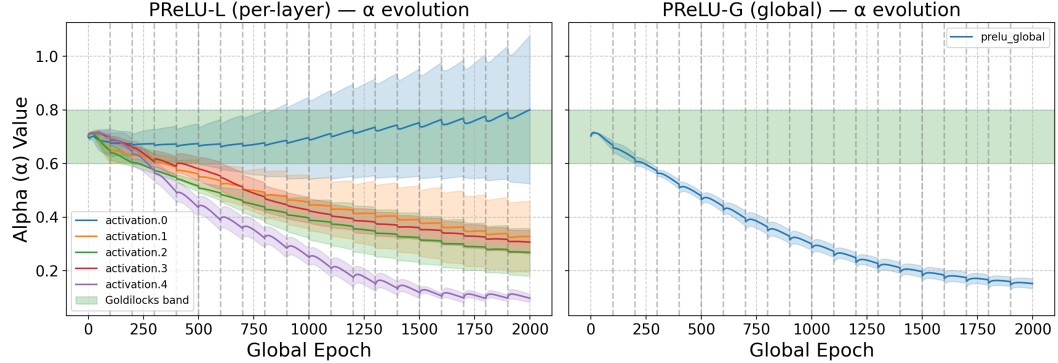

Figure C3: PReLU-GŁper-neuron learnable $\alpha$. **Left:** PReLU-G. Distribution of all neurons' $\alpha$ for the whole network (1 learnable parameter shared across all layers); **Right:** PReLU-L. PReLU-G. Distribution of all layers' parameters $\alpha$ for the whole network (1 learnable parameter per layer).

concrete definition of what constitutes a 'saturated' or 'effectively dead' state based on post-activation values. Therefore, while sharing the goal of identifying inactive units, our saturation fraction metrics employ criteria specific to each activation function family (such as proximity to +/-1 for Tanh, near-zero output for ReLU, or low output magnitude relative to batch statistics for functions like Swish or Leaky ReLU). This specific approach allows us to more precisely capture the distinct ways different activation architectures can lead to or avoid states of unresponsiveness under duress, rather than applying a single relative threshold across functions with fundamentally different output scales and properties. See code for the full criteria used for each activation function.

## D EXPANDING ON CASE STUDY 2: SATURATION-THRESHOLD STRESS TEST

### D.1 STREES PROTOCOL

Let $\Gamma = \{1.5, 0.5, 0.25, 2.0\}$ be the set of shock amplitudes and let $C_l$ be a user–defined *cycle length* (we use $C_l = 10$ for results in the main body Sec. 4). During training we step through the epochs of every task as

$$x^{\text{pre}} \;\longleftarrow\; \gamma_k(t)\, x^{\text{pre}}, \qquad \gamma_k(t) = \begin{cases} \Gamma_k, & t \bmod C_l = 0, \\ 1, & \text{otherwise}, \end{cases}$$

where the index $k$ is advanced cyclically $k \leftarrow (k+1) \bmod |\Gamma|$ *each time a shock epoch occurs*. Thus every $C_l$ epochs we devote exactly *one* epoch to a scale–shock whose value alternates $1 \to 1.5 \to 1 \to 0.5 \to 1 \to 0.25 \to 1 \to 2.00 \to 1 \to \dots$. The multiplicative factor is applied *after* all layers and *before* its non-linearity; all other epochs run with $\gamma = 1$.

### D.2 DERIVATIVE-FLOOR RULE.

The data presented in Figure 2 provides strong support for the idea which posits that activation functions with non-zero derivative floors offer superior desaturation and recovery dynamics.

Fig. 2 (left) reveals that, as expected, stronger "expanding" shocks ($\gamma$=1.5,2.0) generally induce a higher overall saturation impact (AUSC) than "shrinking" shocks ($\gamma$=0.25,0.5) across all floor types.

Critically, activations in the "Non-Zero Floor" category consistently demonstrate the most effective minimization of AUSC, maintaining the lowest values particularly under expanding shocks. Conversely, "Zero Floor" activations exhibit the highest AUSC, indicating a greater susceptibility to prolonged saturation. The "Effective Non-Zero Floor" group shows a mixed response, with AUSC values sometimes higher than "Zero Floor" for certain shrinking shocks (e.g., $\gamma$=0.5) but better than "Zero Floor" under strong expansion.

This pattern of robustness for "Non-Zero Floor" activations is further evidenced in their SF recovery capabilities, as shown in Fig. 2 (middle) and Fig. 2 (right). While successful recovery times tend to be quick (around 1 epoch or slightly more) for stronger shocks ($\gamma$=1.5,2.0) across most types that do recover, the "Non-Zero Floor" category stands out by also maintaining the lowest non-recovery rates, especially for expanding shocks where it approaches 0-5%. In contrast, "Zero Floor" activations not only take noticeably longer to recover SF during shrinking shocks but also suffer from extremely high non-recovery rates across all shock conditions. The

'Effective Non-Zero Floor' group (typically smooth-tailed functions) demonstrates rapid SF recovery Fig. 2 (middle) if recovery occurs. However, they suffer from high non-recovery rates, particularly akin to 'Zero Floor' types during shrinking shocks, and show less consistent AUSC improvements compared to 'Non-Zero Floor' activations. This indicates their negative tail mechanisms, despite providing an average non-zero gradient, may not consistently prevent saturation or ensure recovery across diverse shocks.

In conclusion, "Zero Floor" activations are clearly detrimental to saturation resilience. While "Effective Non-Zero Floor" functions offer fast recoveries when successful, it is the presence of a consistent and sufficiently large "Non-Zero Floor" that most effectively minimizes overall saturation impact and ensures reliable, rapid recovery from saturation-inducing shocks."

## D.3 TWO-SIDED PENALTY.

We hypothesized that activations that saturate on *both* sides (Sigmoid, Tanh, Swish/GELU at very low $\beta$) will accumulate $\geq 50\%$ more saturation and take $\geq 50\%$ longer to recover than one-sided families. The data provides nuanced support for this. As seen in Fig. 3, "Two-Sided" functions indeed accumulate more saturation, evidenced by having the highest average Peak SF Fig. 3 (left) and high AUSC values Fig. 3 (right). While their successful SF recoveries are very fast—around one epoch Fig. 3 (middle)—this is severely counterbalanced by a very high non-recovery rate of 49.83% (data not shown in figure). This high failure rate means that, in practice, they are far less reliable at desaturating than "One-Sided (Kink)" activations, which had a non-recovery rate of 13.30%. If non-recovery is considered an infinite recovery time, then "Two-Sided" functions effectively "take longer to recover" on average due to frequent outright failure. Interestingly, "One-Sided (Smooth)" functions also exhibit a high non-recovery rate (48.93%), barely surpassing that of "Two-Sided" functions, despite their successful recoveries also being very fast. This suggests that while their smooth negative tail can allow for quick desaturation if conditions are right, they are also prone to getting stuck in a saturated state.

Therefore, the penalty for two-sided saturation is evident in both the magnitude of saturation experienced (Peak SF, AUSC) and the overall reliability of recovery (high non-recovery rates). The "One-Sided (Kink)" group, while having a tail of longer successful recovery times, is actually the most reliable at achieving recovery.

## D.4 PERFORMANCE RECOVERY TIME ON CASE STUDY 2

To assess the functional recovery of the network after pre-activation shocks, we measured the performance recovery time $\tau_{95}$, defined as the number of epochs required to regain 95% of the pre-shock validation accuracy on the current task. This metric provides insight into the immediate resilience of the network's learning capability for the task at hand when subjected to sudden internal perturbations.

Overall, the rate of complete performance non-recovery (failure to reach 95% of pre-shock current-task accuracy within the observation window) was very low across all experiments (1.14%). This indicates that, in most instances, the model's ability to perform the current task bounces back effectively after the applied shocks. Figure D1 illustrates the mean $\tau_{95}$ for these successful recoveries, grouped by activation 'Floor Type' (left) and 'Sidedness' (right), across different gamma shock intensities.

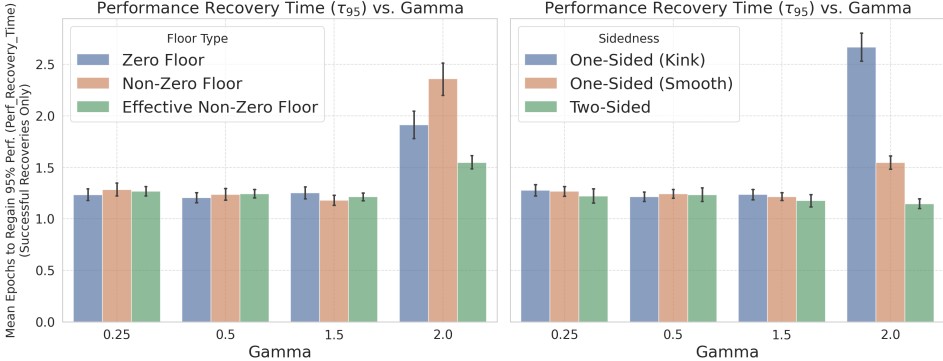

Figure D1: **Functional performance recovery time** $\tau_{95}$ **versus gamma shock intensity**. Grouped by Floor Type **(Left)** and Sidedness **(Right)**, showing mean epochs for successful recoveries. Most activations demonstrate rapid recovery from mild to moderate shocks; however, under strong $\gamma$=2.0 shocks, 'Two-Sided' functions are notably fastest and most reliable (0% non-recovery), while 'Non-Zero Floor' and 'One-Sided (Kink)' types show slower recovery. (Overall performance non-recovery rate: 1.14%)."

For mild to moderate expanding shocks ($\gamma$=1.5) and all shrinking shocks ($\gamma$=0.25,0.5), the mean $\tau_{95}$ is consistently low (around 1.2-1.3 epochs) and remarkably similar across all floor types and sidedness categories. This suggests that for less extreme shocks, most activation functions allow for rapid recovery of performance on the current task. However, differences emerge under strong expanding shocks ($\gamma$=2.0):

- **Impact of Floor Type (at $\gamma$=2.0)**: 'Non-Zero Floor' activations, despite their strong SF recovery characteristics, surprisingly exhibit the longest average $\tau_{95}$ ($\approx$3.4 epochs) for performance and also contribute most to the few performance non-recoveries observed (2.85% non-recovery rate for this group at $\gamma$=2.0). In contrast, 'Effective Non-Zero Floor' ($\approx$1.75 epochs $\tau_{95}$, 0.36% non-recovery) and 'Zero Floor' ($\approx$1.9 epochs $\tau_{95}$, 0.74% non-recovery) activations recover current-task performance more quickly under these strong shocks.
- **Impact of Sidedness (at $\gamma$=2.0)**: Most notably, 'Two-Sided' activations (e.g., Sigmoid, Tanh) demonstrate exceptional current-task functional recovery. They consistently recovered performance (0% non-recovery rate in your stats) and did so fastest on average ($\approx$1.1 epochs). 'One-Sided (Smooth)' activations also performed well. 'One-Sided (Kink)' activations showed the slowest average $\tau_{95}$ and the highest performance non-recovery rate among sidedness categories for these strong shocks.

It is crucial to note that this $\tau_{95}$ metric reflects the immediate recovery of performance on the task currently being trained. While it indicates a certain resilience to transient internal shocks, it does not directly measure the long-term impact on catastrophic forgetting or the overall ability to learn a sequence of tasks effectively.

The ultimate test of an activation function's suitability for continual learning lies in metrics like Final Average Accuracy (ACC) across all tasks and Average Forgetting (AF), evaluated at the end of the entire training sequence. It is in these end-to-end CL metrics that more significant and often different disparities between activation functions emerge. For instance, while 'Two-Sided' activations show rapid current-task performance recovery here, their known issues with vanishing gradients and saturation might still lead to poorer overall ACC or higher forgetting in the full CL scenario. The current $\tau_{95}$ results offer a valuable piece of the puzzle regarding short-term resilience, but the broader impact on plasticity and stability across the entire sequence of tasks remains best assessed by holistic CL evaluation metrics.

## E  EXPANDING ON CONTINUAL SUPERVISED LEARNING EXPERIMENTS

Following (Kumar et al., 2023), we evaluate five supervised continual image–classification benchmarks spanning two shift types: *input distribution shift* (Permuted MNIST, 5+1 CIFAR, Continual ImageNet) and *concept shift* (Random Label MNIST, Random Label CIFAR). Across all settings, training proceeds as a sequence of tasks *without task–identity signals*: the model is never told when a task switch occurs. Within each task, the learner receives mini–batches for a fixed duration (specified below) and is updated incrementally with cross–entropy on the arriving batches. Summary of experimental settings is shown in Table E1.

**Permuted MNIST** (Goodfellow et al., 2013) (input shift). We first sample a fixed subset of 10,000 images from the MNIST training set. Each task is defined by drawing a new *fixed* random permutation over pixel indices and applying it to every image in the subset. The permutation is constant within a task and independent across tasks. Each task presents exactly one pass (1 epoch) over its 10,000 permuted images in mini–batches of size 16; the next task then begins with a new permutation. We train for 500 tasks total. This setting induces strong input–space remapping while preserving label semantics within tasks, isolating rapid adaptation to input shift.

**Random Label MNIST** (Lyle et al., 2023) (concept shift). We fix a subset of 1,200 MNIST images once. For each task, we generate a fresh random label for each image in this subset, leaving the inputs unchanged but altering the input to label mapping. To encourage memorization under an arbitrary target function, the model is trained for 400 epochs per task with batch size 16. After 400 epochs, a new task arrives with an independent random labeling; we run 50 tasks in sequence. Inputs are identical across tasks; only concepts change, directly probing plasticity versus interference.

**Random Label CIFAR** (concept shift). Identical protocol to Random Label MNIST, but using images drawn from *CIFAR–10*. We again use a fixed subset of 1,200 images and re-assign random labels independently per task. Training uses 400 epochs per task, batch size 16, for 50 tasks. This mirrors the concept–shift regime on a higher–variability image domain than MNIST.

**5+1 CIFAR** (input shift with alternating difficulty). Tasks are constructed from *CIFAR–100* and alternate in difficulty: even–indexed tasks are *hard* and odd–indexed tasks are *easy*. A hard task contains data from 5 distinct classes with 2,500 examples total (500 per class); an easy task contains data from a *single* class with 500 examples. Classes do not repeat across the sequence, ensuring non–overlapping exposure. Each task lasts 780 timesteps, which corresponds to approximately 10 epochs on the hard task data set with batch size 32; easy tasks use the same 780 time step budget for consistency. We evaluated performance on the *hard* tasks only (single-class tasks are near the ceiling for all activation functions). Alternating input diversity stresses both rapid

adaptation (when diversity spikes) and retention across shifts, serving as a targeted stress test for plasticity–loss mitigation.

**Continual ImageNet** (Dohare et al., 2024) (input shift). Each task is a binary classification problem between two distinct ImageNet classes. For every task we draw 1,200 images total (600 per class) and *down-sample to* $32 \times 32$, following (Dohare et al., 2024), to reduce compute while maintaining semantic variability. Classes do not repeat across tasks, yielding clear, non–overlapping episodes of input shift. We train for 10 epochs per task with batch size 100 and report task accuracy. Despite down-sampling, the setting retains ImageNet–level variability while enabling precise measurement of adaptation and retention under non–reused classes.

| Benchmark | Per-Task Data Size | Batch | Epochs | Timesteps | # Tasks |
|---|---|---|---|---|---|
| Permuted MNIST | 10,000 images | 16 | 1 | 625 | 500 |
| Random Label MNIST | 1,200 images | 16 | 400 | 30,000 | 50 |
| Random Label CIFAR | 1,200 images | 16 | 400 | 30,000 | 50 |
| 5+1 CIFAR | Hard: 2,500 images (5 classes, 500/class) | 32 | Hard: $\approx 10$ | 780 | 15 |
| | Easy: 500 images (1 class) | | Easy: $\approx 50$ | | 15 |
| Continual ImageNet | 1,200 images/task (600/class) | 100 | 10 | 120 | 500 |

Table E1: Hyperparameters and schedule per benchmark. *Timesteps* denote parameter-update steps (i.e., mini-batches) within a task. For 5+1 CIFAR, a fixed timestep budget per task implies approximate epochs depending on data size.

**Notes.** (i) In 5+1 CIFAR, classes do not repeat across tasks; tasks alternate easy/hard. 780 timesteps $\approx 10$ epochs on the hard set (since $2,500/32 \approx 78.125$ batches/epoch) and $\approx 50$ epochs on the easy set (since $500/32 \approx 15.625$). (ii) In Continual ImageNet, images are downsampled to $32 \times 32$ to reduce compute; classes do not repeat across tasks. (iii) Timesteps are computed as the number of mini-batches per task.

## E.1 EXPERIMENTAL RESULTS EXPANSION

Here we expand the empirical comparison of activation functions across five continual supervised benchmarks. Table 2 reports Total Average Online Task Accuracy. Two broad patterns emerge. First, rectifiers with a learnable or randomized negative branch dominate: `Leaky-ReLU`, `RReLU`, `PReLU`, `Smooth-Leaky`, and its randomized variant consistently outperform `ReLU`—often by large margins (e.g., CIFAR 5+1: `ReLU` 4.76 vs. `Rand. Smooth-Leaky` 57.01). Second, smooth rectifiers (`Swish/SiLU`) also fare well, but tend to trail the best "leaky–family" members on the harder settings. On the memorization-friendly random-label tasks, several of these parametric/leaky activations saturate near 100%, whereas saturating `sigmoid` and `tanh` struggle on CIFAR with random labels—consistent with their known optimization brittleness under distributional churn. Overall, the strongest single performer is `Rand. Smooth-Leaky`, with `Smooth-Leaky`, `RReLU`, and `Leaky-ReLU` close behind.

Table E2 clarifies why these families succeed by showing the optimal shape parameters and learning rates. A striking regularity is a 'Goldilocks zone' for the negative linear sides: When an activation exposes a slope (or an effective initial slope) on the negative branch, the best values typically land between their respective best-performing ranges. For some it tends to be between 0.6 and 0.9, which was initially reported in Section 3. However, this is not a rule, and optimal 'Goldilocks zone' might vary between activations and settings. Nonetheless, this tends holds across `Leaky-ReLU`, `PReLU`, `Smooth-Leaky`, and `Rand. Smooth-Leaky`, and even for `RReLU` when considering the average of its bounds (important because that average initializes the effective leak). For these activations, we also observe that those that are within that range tend to also have higher accuracy values than the ones that are outside the preferred range. Intuitively, this regime prevents dead units and preserves gradient flow without collapsing the asymmetric gating that helps continual adaptation. We also observe a gentle LR shift: simpler datasets (Permuted/Random-Label MNIST) favor $10^{-3}$, while CIFAR 5+1 prefers $10^{-4}$, matching the increased optimization stiffness there.

## E.2 ANALYSIS OF GENERALIZATION GAPS

Table E3 reports the Generalization Gap (GAP), calculated as the difference between the final test accuracy on a specific task (after training on it) and the average online training accuracy during that same task, averaged across all tasks in the sequence. In this online continual learning setting, a positive gap is the expected, healthy outcome. Because the "online training accuracy" averages performance from the very first batch (where the model is ignorant) to the last, it naturally underestimates the model's final capability. A positive gap (Test > Online Train) therefore quantifies the "learning gain": it confirms that the final, converged model performs better on held-out data than its average performance while learning. On the other hand, a negative gap (Online Train > Test) is a critical diagnostic for overfitting or generalization failure. This occurs when the model fits the incoming stream of training batches but fails to retain that performance when evaluated on the static test set immediately after. This distinction highlights a key trade-off. For example, while Deep Fourier Features (DFF) demonstrate high plasticity (high online accuracy) in our main benchmarks, they exhibit large negative

| Activation | Permuted MNIST | Random Label MNIST | Random Label CIFAR | CIFAR 5+1 | Continual ImageNet |
|---|---|---|---|---|---|
| ReLU | − ‖ 0.001 | − ‖ 0.001 | − ‖ 0.001 | − ‖ 0.001 | − ‖ 0.0001 |
| Leaky-ReLU | 0.6 ‖ 0.001 | 0.8 ‖ 0.001 | 0.6 ‖ 0.001 | 0.4 ‖ 0.001 | 0.6 ‖ 0.001 |
| Sigmoid | − ‖ 0.001 | − ‖ 0.001 | − ‖ 0.001 | − ‖ 0.0001 | − ‖ 0.001 |
| Tanh | − ‖ 0.001 | − ‖ 0.001 | − ‖ 0.001 | − ‖ 0.0001 | − ‖ 0.0001 |
| RReLU | [0.6, 0.8] ‖ 0.001 | [0.125, 0.333] ‖ 0.001 | [0.6, 0.8] ‖ 0.001 | [0.673, 2.673] ‖ 0.001 | [0.6, 0.8] ‖ 0.001 |
| PReLU | neuron, $\alpha = 1.2$ ‖ 0.001 | neuron, $\alpha = 0.1$ ‖ 0.001 | global, $\alpha = 0.65$ ‖ 0.0001 | global, $\alpha = 0.9$ ‖ 0.0001 | neuron, $\alpha = 0.65$ ‖ 0.001 |
| Swish (SiLU) | 0.05 ‖ 0.001 | 0.01 ‖ 0.001 | 0.01 ‖ 0.001 | 0.1 ‖ 0.001 | 0.05 ‖ 0.001 |
| GeLU | 0.5 ‖ 0.001 | 0.8 ‖ 0.001 | 0.05 ‖ 0.001 | 1.0 ‖ 0.0001 | 1.0 ‖ 0.001 |
| CeLU | 3.6 ‖ 0.001 | 3.6 ‖ 0.0001 | 2.0 ‖ 0.0001 | 3.3 ‖ 0.001 | 3.3 ‖ 0.001 |
| eLU | 1.0 ‖ 0.001 | 3.6 ‖ 0.0001 | 3.6 ‖ 0.0001 | 3.6 ‖ 0.001 | 1.0 ‖ 0.001 |
| SeLU | 1.0 ‖ 0.001 | 3.0 ‖ 0.0001 | 3.7 ‖ 0.0001 | 3.7 ‖ 0.001 | 3.7 ‖ 0.001 |
| CReLU | − ‖ 0.001 | − ‖ 0.001 | − ‖ 0.001 | − ‖ 0.001 | − ‖ 0.001 |
| Rational | A, (5, 4), Leaky-ReLU ‖ 0.001 | D, (5, 4), Tanh ‖ 0.0001 | A, (5, 4), Tanh ‖ 0.001 | B, (5, 4), Swish ‖ 0.001 | C, (5, 4), ReLU ‖ 0.001 |
| SwiGLU | − ‖ 0.0001 | − ‖ 0.001 | − ‖ 0.0001 | − ‖ 0.001 | − ‖ 0.001 |
| Deep Fourier | − ‖ 0.001 | − ‖ 0.001 | − ‖ 0.001 | − ‖ 0.001 | − ‖ 0.001 |
| Smooth-Leaky | 0.1, 0.3, 0.3 ‖ 0.001 | 0.3, 0.1, 0.3 ‖ 0.001 | 0.3, 0.5, 0.65 ‖ 0.001 | 0.1, 3.0, 0.9 ‖ 0.001 | 3.0, 2.0, 0.65 ‖ 0.001 |
| Rand. Smooth-Leaky | 0.8, 1.0, 0.3, 0.6 ‖ 0.001 | 2.0, 0.8, 0.3, 0.6 ‖ 0.001 | 0.8, 3.0, 0.5, 0.5 ‖ 0.001 | 0.5, 0.5, 0.673, 2.673 ‖ 0.001 | 0.5, 0.5, 0.3, 0.3 ‖ 0.001 |

Table E2: Optimal Hyperparameters for each activation function in each Continual Supervised Benchmark Problem. Represented as activation function shape parameter on the left side of the ‖ symbol and the learning rate to the right. A dash (−) indicates that such activation function uses the unique or baseline parameter (e.g., ReLU does not have any shape-controlling parameter since is linear on the positive right side and 0 on the negative left side). The Total Average Online Accuracy reported for these optimal hyperparameters can be seen in Table 2. PReLU's $\alpha$ indicates initial parameter value. Smooth-Leaky triplets indicate $c, p, \alpha$, while Rand. Smooth-Leaky indicates $c, p$, and bounds $[l, u]$. The tuple of values from Rational indicates Version, ((P), (Q)), Function Approx. where (P) and (Q) are the numerator and denominator degrees respectively of the polynomial.

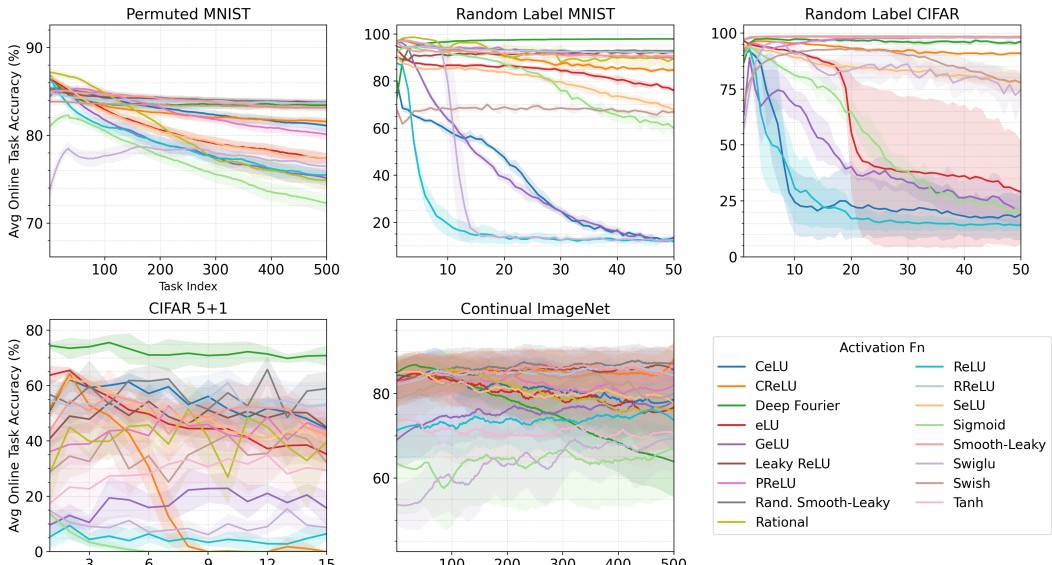

Figure E1: Comparison using Total Average Online Task Accuracy across all five Continual Supervised Learning Benchmarks for all activation functions. See Tab. 2 for the best total average online accuracies. Optimal hyperparameters can be seen in Tab. E2.

gaps on complex input-shift tasks (e.g., $-51.24\%$ on CIFAR 5+1, $-21.01\%$ on Continual ImageNet). This indicates that DFF's "adaptive linearity" comes at the cost of significant overfitting to the immediate training stream. In contrast, Rand. Smooth-Leaky achieves a far superior balance. On Continual ImageNet, it effectively closes the gap ($-1.47\%$), and on CIFAR 5+1, it reduces the gap by nearly half compared to DFF ($-29.67\%$ vs. $-51.24\%$). However, if we look directly at the performance values in Table 2 we see that DFF achieves $72.29\%$ vs Rand. Smooth-Leaky, $57.01$, which points to the same idea of conflation between trainability vs generalizability suggested in Section 7.1 with regard to loss of plasticity.

We believe that in this case, this metric captures a different phenomenon, overfitting the test set. This is a distinct concept from the specific challenge we are diagnosing in our RL analysis. That is the reasoning why for our supervised benchmarks, we instead adopted the standard metrics for generalization in the continual learning

community. As detailed in Appendix B.6.1, we use Total Average Online Accuracy (TAOA) (which follow the setup of (Kumar et al., 2023)) for our main online benchmarks 6, which is a common standard ((Ghunaim et al., 2023; Prabhu et al., 2023; Cai et al., 2021), and Average Accuracy ($ACC_T$) for our class-incremental case studies (Section 3 and Section 4). These metrics are the accepted measures of test-set performance in these CL settings.

| Activation | Permuted MNIST | Rand. Label MNIST | Rand. Label CIFAR | CIFAR 5+1 | Continual ImageNet |
|---|---|---|---|---|---|
| ReLU | $9.37 \pm 0.08$ | $3.98 \pm 1.52$ | $3.52 \pm 2.51$ | $23.42 \pm 3.85$ | $\mathbf{5.49 \pm 0.17}$ |
| Leaky-ReLU | $5.33 \pm 0.03$ | $8.20 \pm 0.29$ | $1.67 \pm 0.01$ | $-20.81 \pm 1.42$ | $-0.24 \pm 0.08$ |
| RReLU | $5.19 \pm 0.02$ | $6.83 \pm 0.11$ | $2.69 \pm 0.01$ | $-25.59 \pm 2.89$ | $-0.34 \pm 0.17$ |
| PReLU | $7.39 \pm 0.02$ | $6.48 \pm 0.90$ | $3.66 \pm 0.31$ | $23.11 \pm 0.48$ | $0.87 \pm 0.33$ |
| Sigmoid | $10.57 \pm 0.05$ | $17.56 \pm 0.39$ | $15.06 \pm 2.27$ | $18.18 \pm 0.66$ | $4.83 \pm 0.35$ |
| Tanh | $\mathbf{12.49 \pm 0.10}$ | $15.25 \pm 0.21$ | $11.57 \pm 0.05$ | $25.97 \pm 0.55$ | $4.77 \pm 0.32$ |
| Swish | $6.00 \pm 0.04$ | $9.86 \pm 0.39$ | $10.11 \pm 2.64$ | $3.38 \pm 2.73$ | $1.72 \pm 0.18$ |
| GeLU | $9.26 \pm 0.06$ | $9.13 \pm 0.56$ | $15.21 \pm 7.59$ | $\mathbf{29.36 \pm 0.42}$ | $0.74 \pm 2.50$ |
| eLU | $8.76 \pm 0.05$ | $16.48 \pm 0.96$ | $8.98 \pm 2.93$ | $-21.19 \pm 2.11$ | $2.45 \pm 0.23$ |
| CeLU | $7.41 \pm 0.05$ | $\mathbf{22.69 \pm 0.88}$ | $3.45 \pm 0.40$ | $-28.31 \pm 2.00$ | $1.67 \pm 0.24$ |
| SeLU | $8.32 \pm 0.15$ | $20.38 \pm 0.64$ | $16.47 \pm 0.77$ | $-20.17 \pm 2.14$ | $1.15 \pm 0.30$ |
| CReLU | $7.42 \pm 0.02$ | $10.33 \pm 0.39$ | $6.74 \pm 0.43$ | $-9.68 \pm 1.47$ | $0.85 \pm 0.10$ |
| Rational | $6.22 \pm 0.06$ | $8.09 \pm 0.95$ | $8.32 \pm 0.23$ | $-20.80 \pm 6.14$ | $2.01 \pm 0.32$ |
| SwiGLU | $10.22 \pm 0.02$ | $2.38 \pm 0.44$ | $\mathbf{17.25 \pm 1.32}$ | $25.01 \pm 2.99$ | $5.47 \pm 0.90$ |
| Deep Fourier | $7.03 \pm 0.04$ | $4.41 \pm 0.15$ | $1.63 \pm 1.40$ | $-51.24 \pm 2.09$ | $-21.01 \pm 0.71$ |
| Smooth-Leaky | $6.42 \pm 0.01$ | $8.27 \pm 0.39$ | $1.66 \pm 0.01$ | $-16.97 \pm 2.05$ | $-0.44 \pm 0.31$ |
| Rand. Smooth-Leaky | $5.98 \pm 0.02$ | $6.47 \pm 0.09$ | $1.59 \pm 0.00$ | $-29.67 \pm 2.42$ | $-1.47 \pm 0.20$ |

Table E3: Generalization Gap (GAP) in percentage (%) across five datasets. Values are reported as Mean $\pm$ Std. The metric is calculated per-task as $GAP_t = TestAcc_{t,final} - TrainAcc_{t,avg}$, then averaged across all tasks. Positive values indicate healthy learning (the final model generalizes better than its average performance during training). Negative values indicate overfitting (the model performed better on the training stream than on the held-out test set). Higher is better. Best performance per dataset is bolded.

## E.3 Continual Learning Interventions Targeting Loss of Plasticity

We evaluate the activation functions used throughout this paper to study the effects of being augmented with three standard continual-learning (CL) algorithms that explicitly target loss of plasticity: $L_2$-Init regularization (Kumar et al., 2023), Self-Normalized Resets (SNR) (Farias & Jozefiak, 2024), and Elastic Weight Consolidation (EWC) (Kirkpatrick et al., 2017). These three methods are applied orthogonally to the network architecture and activation function choices. They constrain or refresh parameters, allowing us to quantify the extent to which activation design still contributes to plasticity when strong CL mechanisms are present.

$L_2$-**Init:** $L_2$ **Regularization to the Initial Parameters.** $L_2$-Init modifies conventional weight decay by introducing a quadratic penalty that pulls the network parameters ($\theta$) back toward their initial values ($\theta_0$), rather than toward zero. The total loss ($\mathcal{L}$) is augmented by the $L_2$-Init regularization term ($\mathcal{L}_{L2Init}$):

$$\mathcal{L}_{L2Init} = \lambda_{L2Init}\|\theta - \theta_0\|_2^2$$

The rationale is that the randomly initialized network resides in a relatively "plastic" region of the parameter space, characterized by well-behaved gradients and non-saturated units. As training progresses across tasks, $\theta$ may drift into "stiff" regions where gradients are less effective or highly anisotropic. $L_2$-Init counteracts this drift by softly anchoring $\theta$ back toward the original, high-plasticity configuration. In our experiments, $\lambda_{L2Init}$ is a tunable hyperparameter, and the penalty is applied throughout training.

**Self-Normalized Resets (SNR).** SNR is a reset-based method specifically designed to mitigate neuron-level loss of plasticity. Instead of continuous regularization, SNR monitors the firing statistics of individual units over time. It tracks an estimate of the expected activation frequency and computes an inter-activation time distribution for each neuron. If a neuron remains inactive for a duration that is statistically unusual relative to its established baseline, SNR determines the unit is effectively dormant and triggers a reset. This reset re-initializes the neuron's incoming weights and its corresponding optimizer state. The core hyperparameter is a percentile threshold ($\eta$). A reset is triggered when the probability, under the tracked firing-rate model, of observing an inactivity stretch at least as long as the current one falls below $\eta$. Higher $\eta$ implies more aggressive resets (quicker refreshment of neurons). Lower $\eta$ corresponds to more conservative application of the reset mechanism. SNR intuitively injects fresh, plastic units into the network whenever existing ones become statistically improbable "dead" units.

**Elastic Weight Consolidation (EWC)**   EWC is a classical regularization-based CL method that addresses catastrophic forgetting by selectively restricting modifications to parameters deemed important for previously encountered tasks. Following the completion of each task, an approximate Fisher information matrix is computed to estimate the importance of each parameter. During subsequent task learning, the overall loss ($\mathcal{L}$) is augmented with a quadratic penalty term ($\mathcal{L}_{\text{EWC}}$):

$$\mathcal{L}_{\text{EWC}} = \lambda_{\text{EWC}} \sum_i F_i (\theta_i - \theta_i^\star)^2$$

Where: $F_i$ is the $i$-th diagonal element of the Fisher information matrix, quantifying the importance of parameter $\theta_i . \theta_i^\star$ are the reference parameters (e.g., those learned after the previous task). $\lambda_{\text{EWC}}$ is the hyperparameter controlling the regularization strength. This penalty increases the cost of updates along directions crucial for past performance, thus preserving stability while permitting movement in less constrained directions essential for new task plasticity. EWC serves as a strong CL baseline that explicitly manages the stability-plasticity trade-off.

**Ablations.**   By applying this complementary suite of CL mechanisms, we can rigorously assess whether the effects arising from activation function geometry and design persist, and how strongly they interact with parameter-constraining or refreshing strategies. For all methods we report Total Average Online Task Accuracy (%) averaged over 5 independent runs. Values are reported as mean $\pm$ standard deviation (SD). For each activation, we reuse its best hyperparameters from the main experiments (see Table 2 and sweep a small log-spaced grid over the CL-method hyperparameters, reporting the best value in each column. **Vanilla** indicates original results. For each CL intervention (L2-Init, SNR, EWC) we swept a small grid over its main hyperparameter and report the best values per activation and dataset in Table E11, Table E10, and Table E9.

### E.3.1   DISCUSSION OF RESULTS

**Activation Choice Defines the Plasticity Ceiling.**   A consistent hierarchy emerges across all benchmarks: while CL interventions—most notably L2-Init—can mitigate plasticity loss in fragile activations like `ReLU`, they cannot compensate for the geometric limitations of the activation function itself. For instance, in the challenging Random Label regimes (Tables E5 and E6), `Vanilla ReLU` suffers total collapse ($\approx 20\%$ accuracy), effectively requiring L2-Init to function at all. In contrast, our proposed `Rand. Smooth-Leaky` demonstrates inherent robustness, achieving high performance even without interventions, and reaching the highest overall performance ceiling when augmented with L2-Init (e.g., 95.20% on Continual ImageNet). This suggests that optimal plasticity, when coupled with CL interventions, requires a synergy: an activation function that maintains gradient flow (Smooth-Leaky) combined with a regularization method that maintains a favorable operating point (L2-Init).

**Specific Activation Profiles.**   We observe distinct specialization among prior approaches. `Deep Fourier` excels in high-frequency permutation and noise tasks (Table E6), effectively solving the task without intervention (96.24%), yet struggles to generalize to natural image distributions (Table E8) where it lags behind smooth non-monotonic functions. `CReLU` acts as a reliable "stabilized ReLU," consistently outperforming the baseline but often hitting a lower asymptotic limit than the smooth variants. `Rand. Smooth-Leaky` effectively bridges this gap, matching `Deep Fourier`'s robustness on noise tasks while dominating on natural data, indicating it captures the necessary curvature for diverse task boundaries.

**The "Constraint-Plasticity" Paradox.**   A critical anomaly is observed in the CIFAR 5+1 benchmark (Table E7), where CL interventions actively *degrade* the performance of strong activations. While L2-Init aids `ReLU`, it catastrophic reduces the accuracy of `Deep Fourier` (from 72.29% to 20.40%) and `Rand. Smooth-Leaky` (from 57.01% to 34.56%). This suggests a regime where the optimal solution lies far from the initialization ($\theta_0$). By anchoring parameters to $\theta_0$, L2-Init prevents the significant semantic drift required for this specific task shift. Notably, `Vanilla Deep Fourier` achieves the highest performance in this setting, proving that in scenarios requiring extreme adaptation, the native plasticity of the activation function is superior to external constraints. Conversely, methods like EWC consistently underperform across most benchmarks, confirming that stability-focused regularization is often antithetical to the requirements of plasticity-heavy regimes.

## F   EXPANDING ON CONTINUAL REINFORCEMENT LEARNING EXPERIMENTS

**Robust Metrics for Non-Stationary RL.**   Previously, we defined the generalization gap as $\text{GAP}_{c,e} = R_{c,e}^{\text{train}} - R_{c,e}^{\text{test}}$. For each activation and environment, we summarize the change across cycles as $\Delta(\text{GAP}_e) = \text{GAP}_{3,e} - \text{GAP}_{1,e}$. A value of $\Delta < 0$ implies the gap *shrinks* (improved transfer), while $\Delta > 0$ implies it *widens* (worse transfer).

To provide a rigorous summary across environments with vastly different reward scales (e.g., `Humanoid` vs. `Hopper`), we depart from simple means or medians, which can be sensitive to outliers or mask the magnitude of failures. Instead, we adopt the **Interquartile Mean (IQM)** (Agarwal et al., 2021) for all cross-environment

| Activation | Vanilla | +L2-Init | +SNR | +EWC |
|---|---|---|---|---|
| ReLU | $78.85 \pm 0.06$ | $\mathbf{93.36 \pm 0.38}$ | $85.70 \pm 1.12$ | $53.85 \pm 2.11$ |
| Leaky-ReLU | $84.14 \pm 0.01$ | $\mathbf{91.60 \pm 0.47}$ | $89.35 \pm 0.38$ | $58.42 \pm 1.38$ |
| Sigmoid | $76.96 \pm 0.07$ | $\mathbf{89.88 \pm 0.30}$ | $85.02 \pm 0.35$ | $58.94 \pm 3.33$ |
| Tanh | $70.32 \pm 0.54$ | $\mathbf{86.12 \pm 0.32}$ | $85.32 \pm 0.19$ | $59.35 \pm 2.09$ |
| RReLU | $83.95 \pm 0.02$ | $\mathbf{90.31 \pm 0.44}$ | $88.91 \pm 0.33$ | $57.95 \pm 5.19$ |
| PReLU | $82.62 \pm 0.05$ | $\mathbf{92.49 \pm 0.59}$ | $88.72 \pm 0.38$ | $58.87 \pm 1.47$ |
| Swish | $83.41 \pm 0.03$ | $\mathbf{90.40 \pm 0.63}$ | $89.24 \pm 0.27$ | $60.50 \pm 2.57$ |
| GeLU | $78.97 \pm 0.09$ | $\mathbf{93.24 \pm 0.36}$ | $85.72 \pm 0.40$ | $56.43 \pm 3.18$ |
| CeLU | $82.93 \pm 0.04$ | $\mathbf{93.06 \pm 0.24}$ | $89.51 \pm 0.35$ | $62.10 \pm 5.64$ |
| eLU | $80.50 \pm 0.09$ | $\mathbf{93.56 \pm 0.40}$ | $87.56 \pm 0.42$ | $57.94 \pm 1.65$ |
| SeLU | $80.43 \pm 0.16$ | $\mathbf{93.67 \pm 0.58}$ | $87.22 \pm 0.34$ | $54.88 \pm 3.20$ |
| CReLU | $82.66 \pm 0.04$ | $\mathbf{93.22 \pm 0.58}$ | $89.58 \pm 0.46$ | $45.39 \pm 4.07$ |
| Rational | $80.08 \pm 0.05$ | $\mathbf{87.96 \pm 0.56}$ | $80.17 \pm 0.07$ | $50.10 \pm 2.30$ |
| SwiGLU | $77.69 \pm 0.26$ | $\mathbf{88.93 \pm 0.23}$ | $87.68 \pm 0.30$ | $74.95 \pm 1.72$ |
| Deep Fourier | $83.69 \pm 0.04$ | $\mathbf{93.76 \pm 0.18}$ | $90.68 \pm 0.50$ | $38.98 \pm 3.20$ |
| Smooth-Leaky | $84.03 \pm 0.02$ | $\mathbf{92.19 \pm 0.43}$ | $89.91 \pm 0.42$ | $60.16 \pm 3.30$ |
| Rand. Smooth-Leaky | $84.26 \pm 0.02$ | $\mathbf{93.92 \pm 0.37}$ | $90.10 \pm 0.48$ | $61.88 \pm 2.20$ |

Table E4: Effect of activation function with a series of Continual Learning algorithms on Permuted MNIST.

| Activation | Vanilla | +L2-Init | +SNR | +EWC |
|---|---|---|---|---|
| ReLU | $20.03 \pm 2.46$ | $\mathbf{100.00 \pm 0.00}$ | $14.65 \pm 1.94$ | $12.13 \pm 0.74$ |
| Leaky-ReLU | $91.53 \pm 0.18$ | $99.90 \pm 0.18$ | $\mathbf{100.00 \pm 0.00}$ | $22.77 \pm 1.80$ |
| Sigmoid | $79.59 \pm 0.75$ | $\mathbf{99.42 \pm 0.48}$ | $91.43 \pm 1.53$ | $82.10 \pm 12.65$ |
| Tanh | $63.40 \pm 0.12$ | $\mathbf{98.67 \pm 0.33}$ | $92.63 \pm 1.45$ | $13.07 \pm 1.11$ |
| RReLU | $93.10 \pm 0.02$ | $\mathbf{100.00 \pm 0.00}$ | $99.95 \pm 0.07$ | $77.85 \pm 25.94$ |
| PReLU | $92.67 \pm 0.23$ | $\mathbf{100.00 \pm 0.00}$ | $\mathbf{100.00 \pm 0.00}$ | $85.10 \pm 33.32$ |
| Swish | $67.73 \pm 0.46$ | $89.00 \pm 5.14$ | $\mathbf{100.00 \pm 0.00}$ | $24.72 \pm 1.49$ |
| GeLU | $38.79 \pm 0.95$ | $\mathbf{99.90 \pm 0.14}$ | $15.40 \pm 1.55$ | $36.57 \pm 3.11$ |
| CeLU | $37.16 \pm 0.90$ | $\mathbf{100.00 \pm 0.00}$ | $20.98 \pm 1.96$ | $27.27 \pm 1.03$ |
| eLU | $84.23 \pm 0.70$ | $\mathbf{100.00 \pm 0.00}$ | $99.98 \pm 0.04$ | $54.37 \pm 41.67$ |
| SeLU | $79.95 \pm 0.91$ | $\mathbf{100.00 \pm 0.00}$ | $98.45 \pm 1.09$ | $56.03 \pm 37.14$ |
| CReLU | $89.47 \pm 0.28$ | $\mathbf{100.00 \pm 0.00}$ | $\mathbf{100.00 \pm 0.00}$ | $16.75 \pm 2.22$ |
| Rational | $92.35 \pm 1.97$ | $\mathbf{100.00 \pm 0.00}$ | $\mathbf{100.00 \pm 0.00}$ | $36.27 \pm 19.21$ |
| SwiGLU | $31.20 \pm 2.10$ | $\mathbf{99.85 \pm 0.34}$ | $12.75 \pm 0.65$ | $12.13 \pm 1.19$ |
| Deep Fourier | $92.61 \pm 0.04$ | $\mathbf{100.00 \pm 0.00}$ | $\mathbf{100.00 \pm 0.00}$ | $47.45 \pm 47.97$ |
| Smooth-Leaky | $91.69 \pm 0.12$ | $99.40 \pm 1.34$ | $\mathbf{100.00 \pm 0.00}$ | $82.75 \pm 38.57$ |
| Rand. Smooth-Leaky | $93.33 \pm 0.05$ | $\mathbf{100.00 \pm 0.00}$ | $99.93 \pm 0.15$ | $33.52 \pm 2.83$ |

Table E5: Effect of activation function with a series of Continual Learning algorithms on Random Label MNIST.

| Activation | Vanilla | +L2-Init | +SNR | +EWC |
|---|---|---|---|---|
| ReLU | 25.79 ± 6.18 | **93.68 ± 2.81** | 18.23 ± 8.29 | 12.77 ± 1.47 |
| Leaky-ReLU | 98.34 ± 0.01 | **100.00 ± 0.00** | **100.00 ± 0.00** | **100.00 ± 0.00** |
| Sigmoid | 52.24 ± 2.99 | 46.83 ± 48.50 | 31.53 ± 4.99 | 12.37 ± 2.78 |
| Tanh | 58.56 ± 1.05 | **100.00 ± 0.00** | 87.55 ± 12.24 | 11.88 ± 1.72 |
| RReLU | 98.02 ± 0.03 | **100.00 ± 0.00** | **100.00 ± 0.00** | 87.25 ± 20.40 |
| PReLU | 96.86 ± 0.32 | **100.00 ± 0.00** | **100.00 ± 0.00** | 64.73 ± 48.29 |
| Swish | 87.40 ± 2.42 | **100.00 ± 0.00** | **100.00 ± 0.00** | 70.25 ± 40.74 |
| GeLU | 42.85 ± 2.12 | 11.45 ± 1.45 | 29.10 ± 11.14 | 13.83 ± 3.82 |
| CeLU | 29.64 ± 10.44 | **99.57 ± 0.57** | 17.72 ± 8.77 | 11.20 ± 0.25 |
| eLU | 57.45 ± 20.16 | 100.00 ± 0.00 | 44.92 ± 46.84 | 29.50 ± 39.43 |
| SeLU | 84.61 ± 2.07 | **100.00 ± 0.00** | 99.97 ± 0.07 | 43.95 ± 46.06 |
| CReLU | 92.90 ± 0.13 | **100.00 ± 0.00** | **100.00 ± 0.00** | 83.53 ± 36.82 |
| Rational | 94.82 ± 0.75 | **96.9 ± 1.24** | 95.56 ± 0.56 | 30.92 ± 38.66 |
| SwiGLU | 83.06 ± 3.51 | 84.57 ± 34.5 | **98.63 ± 1.87** | 19.88 ± 12.59 |
| Deep Fourier | 96.24 ± 0.51 | **100.00 ± 0.00** | **100.00 ± 0.00** | **100.00 ± 0.00** |
| Smooth-Leaky | 98.36 ± 0.00 | **100.00 ± 0.00** | **100.00 ± 0.00** | **100.00 ± 0.00** |
| Rand. Smooth-Leaky | 98.42 ± 0.01 | **100.00 ± 0.00** | **100.00 ± 0.00** | **100.00 ± 0.00** |

Table E6: Effect of activation function with a series of Continual Learning algorithms on Random Label CIFAR.

| Activation | Vanilla | +L2-Init | +SNR | +EWC |
|---|---|---|---|---|
| ReLU | 4.76 ± 1.01 | **47.52 ± 11.87** | 31.76 ± 7.13 | 21.20 ± 14.37 |
| Leaky-ReLU | **48.86 ± 0.70** | 33.60 ± 12.91 | 33.20 ± 13.82 | 23.44 ± 1.61 |
| Sigmoid | 1.79 ± 0.19 | **20.00 ± 0.00** | **20.00 ± 0.00** | 18.40 ± 2.31 |
| Tanh | 28.59 ± 2.34 | **52.00 ± 9.84** | 45.36 ± 12.45 | 42.56 ± 5.26 |
| RReLU | **53.60 ± 1.06** | 34.56 ± 7.30 | 34.00 ± 8.40 | 29.60 ± 10.48 |
| PReLU | 43.30 ± 0.61 | **56.94 ± 7.98** | 55.84 ± 7.69 | 56.92 ± 10.13 |
| Swish | 35.31 ± 1.87 | 39.44 ± 8.09 | **41.68 ± 8.20** | 29.36 ± 9.37 |
| GeLU | 17.60 ± 1.71 | **54.48 ± 5.56** | 49.36 ± 5.96 | 53.28 ± 16.69 |
| CeLU | **54.23 ± 1.44** | 27.04 ± 5.92 | 31.44 ± 11.17 | 25.84 ± 8.56 |
| eLU | **47.64 ± 1.44** | 36.16 ± 7.65 | 34.08 ± 2.69 | 30.64 ± 8.15 |
| SeLU | **49.07 ± 1.25** | 32.56 ± 4.81 | 36.48 ± 7.24 | 29.60 ± 7.14 |
| CReLU | **20.56 ± 2.28** | 20.00 ± 0.00 | 4.00 ± 8.94 | 16.72 ± 10.79 |
| Rational | **40.41 ± 4.21** | 20.00 ± 0.00 | 20.00 ± 0.00 | 25.20 ± 6.01 |
| SwiGLU | 9.57 ± 1.81 | **38.56 ± 4.48** | 35.28 ± 8.37 | 37.60 ± 11.30 |
| Deep Fourier | **72.29 ± 2.11** | 20.40 ± 1.52 | 20.08 ± 0.18 | 27.12 ± 5.10 |
| Smooth-Leaky | **49.87 ± 1.67** | 35.20 ± 19.55 | 32.96 ± 10.46 | 31.12 ± 11.40 |
| Rand. Smooth-Leaky | **57.01 ± 1.59** | 34.56 ± 6.01 | 29.44 ± 7.38 | 34.72 ± 10.91 |

Table E7: Effect of activation function with a series of Continual Learning algorithms on CIFAR 5+1.

| Activation | Vanilla | +L2-Init | +SNR | +EWC |
|---|---|---|---|---|
| ReLU | 73.71 ± 0.43 | **85.60 ± 8.29** | 81.20 ± 10.55 | 64.80 ± 11.45 |
| Leaky-ReLU | 85.28 ± 0.20 | 80.00 ± 8.25 | **92.00 ± 3.74** | 80.00 ± 6.63 |
| Sigmoid | 63.89 ± 7.38 | 71.29 ± 0.56 | 69.60 ± 14.10 | **72.00 ± 13.78** |
| Tanh | 70.97 ± 0.44 | 71.11 ± 1.45 | **80.40 ± 9.53** | 66.00 ± 10.86 |
| RReLU | 84.97 ± 0.17 | 84.40 ± 0.65 | **85.20 ± 8.56** | 80.40 ± 10.14 |
| PReLU | **82.37 ± 0.11** | 74.80 ± 14.39 | 80.80 ± 11.28 | 71.60 ± 15.58 |
| Swish | 82.64 ± 0.99 | 82.01 ± 0.56 | **83.60 ± 15.32** | 78.40 ± 20.85 |
| GeLU | 75.49 ± 0.11 | **80.40 ± 5.90** | 53.60 ± 8.05 | 72.40 ± 10.53 |
| CeLU | 81.15 ± 0.68 | 84.00 ± 3.12 | **85.20 ± 3.63** | 80.40 ± 4.56 |
| eLU | 80.10 ± 0.34 | 86.80 ± 2.28 | **88.40 ± 4.34** | 53.20 ± 10.83 |
| SeLU | 80.98 ± 0.49 | **84.36 ± 1.23** | 83.60 ± 7.40 | 75.60 ± 6.07 |
| CReLU | 84.85 ± 0.25 | **88.40 ± 4.56** | 86.40 ± 10.24 | 79.60 ± 7.40 |
| Rational | 80.65 ± 0.38 | **86.16 ± 1.13** | 81.60 ± 5.73 | 52.40 ± 8.99 |
| SwiGLU | 63.57 ± 2.04 | 72.00 ± 20.59 | 79.60 ± 10.71 | **82.80 ± 10.26** |
| Deep Fourier | 76.03 ± 0.75 | **78.40 ± 8.17** | 51.60 ± 9.32 | 74.80 ± 12.38 |
| Smooth-Leaky | 85.38 ± 0.25 | **91.20 ± 2.28** | 88.80 ± 5.40 | 90.40 ± 8.41 |
| Rand. Smooth-Leaky | 86.23 ± 0.13 | **95.20 ± 1.20** | 86.40 ± 10.81 | 81.60 ± 12.44 |

Table E8: Effect of activation function with a series of Continual Learning algorithms on Continual ImageNet.

| | Permuted MNIST | | Rand-label MNIST | | Rand-label CIFAR | | 5+1 CIFAR | | Cont. ImageNet | |
|---|---|---|---|---|---|---|---|---|---|---|
| Activation | LR | $\lambda$ | LR | $\lambda$ | LR | $\lambda$ | LR | $\lambda$ | LR | $\lambda$ |
| ReLU | 1e-3 | 3e-4 | 1e-4 | 1e-3 | 1e-4 | 1e-3 | 1e-4 | 3e-4 | 1e-4 | 1e-3 |
| Leaky-ReLU | 1e-3 | 1e-3 | 1e-3 | 1e-4 | 1e-3 | 1e-4 | 1e-3 | 3e-4 | 1e-3 | 1e-4 |
| Sigmoid | 1e-3 | 1e-3 | 1e-3 | 3e-4 | 3e-4 | 1e-3 | 1e-3 | 1e-3 | 1e-3 | 3e-4 |
| Tanh | 1e-4 | 1e-3 | 1e-4 | 1e-3 | 1e-4 | 1e-4 | 1e-4 | 1e-3 | 1e-4 | 1e-3 |
| PReLU | 1e-3 | 1e-4 | 1e-3 | 3e-4 | 1e-4 | 1e-3 | 1e-4 | 3e-4 | 1e-4 | 3e-4 |
| RReLU | 1e-3 | 3e-4 | 1e-3 | 1e-4 | 1e-4 | 1e-3 | 1e-3 | 3e-4 | 1e-4 | 1e-3 |
| Swish | 1e-3 | 1e-3 | 1e-3 | 3e-4 | 1e-3 | 1e-3 | 1e-3 | 3e-4 | 1e-4 | 1e-3 |
| CeLU | 1e-3 | 1e-3 | 1e-4 | 1e-4 | 1e-4 | 1e-3 | 1e-3 | 3e-4 | 1e-3 | 1e-3 |
| eLU | 1e-3 | 1e-3 | 1e-4 | 3e-4 | 1e-4 | 3e-4 | 1e-3 | 1e-4 | 1e-3 | 1e-3 |
| GeLU | 1e-3 | 3e-4 | 1e-3 | 3e-4 | 1e-4 | 1e-3 | 1e-4 | 1e-4 | 1e-3 | 3e-4 |
| SeLU | 1e-3 | 1e-3 | 1e-4 | 1e-4 | 1e-4 | 3e-4 | 1e-3 | 1e-4 | 1e-4 | 1e-3 |
| SwiGLU | 1e-4 | 1e-4 | 1e-3 | 1e-3 | 1e-4 | 1e-4 | 1e-4 | 1e-4 | 1e-3 | 1e-4 |
| Rational | 1e-3 | 3e-4 | 1e-4 | 3e-4 | 3e-4 | 1e-4 | 1e-3 | 1e-4 | 1e-3 | 1e-4 |
| CReLU | 1e-3 | 1e-3 | 1e-3 | 1e-4 | 3e-4 | 1e-3 | 1e-3 | 1e-3 | 1e-3 | 1e-4 |
| Deep Fourier | 1e-3 | 1e-3 | 1e-3 | 1e-4 | 3e-4 | 1e-4 | 1e-3 | 3e-4 | 1e-3 | 1e-3 |
| Smooth-Leaky | 1e-3 | 1e-3 | 1e-3 | 1e-4 | 1e-3 | 1e-3 | 1e-3 | 1e-4 | 1e-3 | 3e-4 |
| Rand. Smooth-Leaky | 1e-3 | 1e-3 | 1e-3 | 1e-4 | 1e-3 | 1e-3 | 1e-3 | 1e-3 | 1e-3 | 1e-3 |

Table E9: Best L2-Init hyperparameters per activation and dataset. We report the learning rate and the L2-Init coefficient $\lambda$. We sweep over $\lambda \in [1e-4, \ 3e-4, \ 1e-3]$ values.

| | Permuted MNIST | | Rand-label MNIST | | Rand-label CIFAR | | 5+1 CIFAR | | Cont. ImageNet | |
|---|---|---|---|---|---|---|---|---|---|---|
| **Activation** | LR | $\lambda$ | LR | $\lambda$ | LR | $\lambda$ | LR | $\lambda$ | LR | $\lambda$ |
| ReLU | 1e-3 | 3e-3 | 1e-4 | 3e-3 | 1e-4 | 1e-2 | 1e-4 | 3e-3 | 1e-4 | 1e-3 |
| Leaky-ReLU | 3e-3 | 3e-3 | 1e-3 | 1e-2 | 1e-3 | 1e-2 | 1e-3 | 3e-3 | 1e-3 | 3e-3 |
| Sigmoid | 1e-3 | 3e-3 | 1e-3 | 3e-3 | 1e-3 | 1e-2 | 1e-3 | 3e-3 | 1e-3 | 3e-3 |
| Tanh | 1e-3 | 1e-3 | 1e-3 | 1e-3 | 1e-4 | 1e-2 | 1e-4 | 1e-3 | 1e-4 | 1e-3 |
| PReLU | 1e-3 | 3e-3 | 1e-3 | 1e-3 | 1e-4 | 3e-3 | 1e-4 | 3e-3 | 1e-4 | 1e-3 |
| RReLU | 1e-3 | 1e-3 | 1e-3 | 3e-3 | 1e-3 | 1e-2 | 1e-3 | 1e-2 | 1e-3 | 3e-3 |
| CELU | 1e-2 | 1e-3 | 1e-4 | 3e-3 | 1e-4 | 1e-2 | 1e-3 | 1e-2 | 1e-3 | 1e-2 |
| ELU | 1e-3 | 3e-3 | 1e-4 | 1e-3 | 1e-4 | 3e-3 | 1e-3 | 3e-3 | 1e-3 | 3e-3 |
| GELU | 1e-2 | 1e-3 | 1e-3 | 1e-3 | 1e-3 | 1e-3 | 1e-4 | 1e-3 | 1e-3 | 3e-3 |
| SELU | 1e-3 | 1e-3 | 1e-4 | 3e-3 | 1e-4 | 1e-3 | 1e-3 | 3e-3 | 1e-3 | 1e-3 |
| Swish | 1e-3 | 1e-3 | 1e-3 | 1e-2 | 1e-4 | 1e-3 | 1e-3 | 1e-3 | 1e-3 | 1e-2 |
| CReLU | 1e-3 | 3e-3 | 1e-3 | 1e-2 | 1e-3 | 1e-2 | 1e-3 | 3e-3 | 1e-3 | 3e-3 |
| Rational | 1e-3 | 1e-2 | 1e-4 | 1e-3 | 1e-4 | 3e-3 | 1e-3 | 3e-3 | 1e-3 | 1e-3 |
| SwiGLU | 1e-4 | 1e-3 | 1e-4 | 3e-3 | 1e-4 | 1e-3 | 1e-4 | 3e-3 | 1e-4 | 3e-3 |
| Deep Fourier | 1e-3 | 3e-3 | 1e-3 | 1e-2 | 1e-4 | 3e-3 | 1e-3 | 3e-3 | 1e-3 | 1e-2 |
| Smooth-Leaky | 1e-3 | 1e-2 | 1e-3 | 3e-3 | 1e-3 | 3e-3 | 1e-3 | 3e-3 | 1e-3 | 1e-3 |
| Rand.  Smooth-Leaky | 1e-3 | 1e-2 | 1e-3 | 1e-2 | 1e-3 | 1e-2 | 1e-3 | 1e-2 | 1e-3 | 1e-3 |

Table E10: Best SNR hyperparameters per activation and dataset. We report the learning rate and the SNR $\eta$ controlling how rare an inactivity event must be (under the estimated firing-rate model) before resetting a neuron (tail probability $P(A >= a) <= \eta$). For all runs we used an activation magnitude threshold to count as "fired" of $\epsilon = 1\times10^{-3}$ and a max effective window size for mean estimate of 1000. We sweep over $\eta \in [1e3,\ 3e-3,\ 1e-2]$ values.

| | Permuted MNIST | | Rand-label MNIST | | Rand-label CIFAR | | 5+1 CIFAR | | Cont. ImageNet | |
|---|---|---|---|---|---|---|---|---|---|---|
| **Activation** | LR | $\lambda$ | LR | $\lambda$ | LR | $\lambda$ | LR | $\lambda$ | LR | $\lambda$ |
| ReLU | 1e-3 | 1e1 | 1e-4 | 5e1 | 1e-4 | 1e1 | 1e-4 | 5e1 | 1e-4 | 1e1 |
| Leaky-ReLU | 1e-3 | 1e3 | 1e-3 | 1e1 | 1e-3 | 5e1 | 1e-3 | 2e2 | 1e-3 | 1e1 |
| Sigmoid | 1e-3 | 1e1 | 1e-3 | 1e1 | 1e-3 | 1e1 | 1e-3 | 1e1 | 1e-3 | 1e1 |
| Tanh | 1e-3 | 1e1 | 1e-4 | 2e2 | 1e-4 | 1e1 | 1e-4 | 5e1 | 1e-4 | 2e2 |
| PReLU | 1e-3 | 1e1 | 1e-3 | 1e1 | 1e-4 | 5e1 | 1e-4 | 1e1 | 1e-4 | 1e1 |
| RReLU | 1e-3 | 1e1 | 1e-3 | 1e1 | 1e-3 | 1e1 | 1e-3 | 5e1 | 1e-3 | 1e1 |
| CELU | 1e-3 | 1e1 | 1e-4 | 1e1 | 1e-4 | 1e1 | 1e-3 | 2e2 | 1e-3 | 1e1 |
| ELU | 1e-3 | 1e1 | 1e-4 | 1e1 | 1e-4 | 5e1 | 1e-3 | 2e2 | 1e-3 | 2e2 |
| GELU | 1e-3 | 1e1 | 1e-3 | 1e1 | 1e-3 | 1e1 | 1e-4 | 1e1 | 1e-3 | 1e1 |
| SELU | 1e-3 | 1e1 | 1e-4 | 1e1 | 1e-4 | 2e2 | 1e-3 | 2e2 | 1e-3 | 1e1 |
| Swish | 1e-3 | 1e1 | 1e-3 | 1e1 | 1e-4 | 1e1 | 1e-3 | 1e1 | 1e-3 | 1e1 |
| CReLU | 1e-3 | 1e1 | 1e-3 | 1e1 | 1e-3 | 5e1 | 1e-3 | 1e1 | 1e-3 | 1e1 |
| Rational | 1e-3 | 1e1 | 1e-4 | 1e1 | 1e-3 | 5e1 | 1e-3 | 1e1 | 1e-4 | 1e1 |
| SwiGLU | 1e-4 | 1e1 | 1e-3 | 1e1 | 1e-4 | 1e1 | 1e-4 | 1e1 | 1e-4 | 1e1 |
| Deep Fourier | 1e-3 | 1e1 | 1e-3 | 2e2 | 1e-4 | 2e2 | 1e-3 | 2e2 | 1e-3 | 1e1 |
| Smooth-Leaky | 1e-3 | 1e1 | 1e-3 | 5e1 | 1e-3 | 1e1 | 1e-3 | 2e2 | 1e-3 | 1e1 |
| Rand.  Smooth-Leaky | 1e-3 | 1e1 | 1e-3 | 1e1 | 1e-3 | 5e1 | 1e-3 | 2e2 | 1e-3 | 1e1 |

Table E11: Best EWC hyperparameters per activation and dataset. We report the learning rate and the EWC strength of quadratic penalty $\lambda$. For all runs we used a decay for older tasks of $\gamma = 1$. We sweep over $\lambda \in [10,\ 50,\ 200]$ values.

| Activation | IQM $\pm$ 95% CI | | |
| --- | --- | --- | --- |
| | Plasticity Score | LR | Optimal HP |
| ReLU | $0.1593 \pm 0.043$ | 0.0001 | – |
| Leaky–ReLU | $0.1580 \pm 0.115$ | 1e-05 | 0.8 |
| Sigmoid | $0.3301 \pm 0.059$ | 0.0001 | – |
| Tanh | $0.2121 \pm 0.028$ | 1e-05 | – |
| RReLU | $0.2255 \pm 0.059$ | 0.0001 | Bounds: $[0.125, 0.333]$ |
| PReLU | $0.2695 \pm 0.038$ | 0.0001 | Layer. $\alpha$=0.65 |
| Swish (SiLU) | $0.3130 \pm 0.071$ | 0.0001 | 0.01 |
| GeLU | $0.1658 \pm 0.035$ | 0.0001 | 1.0 |
| eLU | $0.1476 \pm 0.107$ | 1e-05 | 1.0 |
| CeLU | $0.1431 \pm 0.070$ | 1e-05 | 1.0 |
| SeLU | $0.2182 \pm 0.032$ | 1e-05 | 1.673 |
| CReLU | $0.1200 \pm 0.010$ | 1e-05 | – |
| Rational | $0.2217 \pm 0.018$ | 1e-05 | C, 5, 4, Leaky-ReLU |
| SwiGLU | $0.0781 \pm 0.008$ | 0.0001 | – |
| Deep Fourier | $0.1766 \pm 0.008$ | 1e-05 | – |
| Smooth-Leaky | $0.3283 \pm 0.037$ | 0.0001 | C:0.5, P:2.0, $\alpha$=0.1 |
| Rand. Smooth-Leaky | $\mathbf{0.3853 \pm 0.038}$ | 0.0001 | C:0.1, P:1.0. Bounds: $[0.01, 0.02]$ |

Table F1: IQM plasticity score across 5 seeds with 95% bootstrap confidence intervals (higher is better). Best IQM per column is bolded. We also report optimal learning rate (LR) and optimal hyperparameters (HP) per activation function. In the case of bounded activations we provide the lower and upper bounds. PReLU provides granularity level at the number of parameters per activation layer. A dash $(-)$, in the Optimal HP column, indicates that such activation function uses the unique or baseline parameter (e.g., ReLU only has slope $\alpha = 0$).

aggregations. Crucially, we address the issue of physics simulation instabilities (where plasticity loss leads to rewards $\ll -10^6$). We apply a *stability filter*: any run resulting in a physics explosion is treated as a functional failure. For the Generalization Gap, this means assigning a gap of $0.0$ (as no meaningful performance difference exists between two failures), preventing numerical artifacts from skewing the aggregate $\Delta$. We report the average reward per environment for all activation functions in Table F2.

| Activation | HalfCheetah-v5 | Humanoid-v5 | Ant-v5 | Hopper-v5 |
| --- | --- | --- | --- | --- |
| ReLU | $730.7 \pm 787.7$ | $-1.0 \times 10^6 \pm 1.7 \times 10^6$ | $1041.7 \pm 313.6$ | $-6004.2 \pm 1.2 \times 10^4$ |
| Leaky-ReLU | $-2.5 \times 10^5 \pm 4.3 \times 10^5$ | $-2.0 \times 10^5 \pm 1.3 \times 10^5$ | $-4.5 \times 10^4 \pm 9.1 \times 10^4$ | $107.0 \pm 62.8$ |
| Sigmoid | $2113.2 \pm 811.5$ | $\mathbf{342.2 \pm 46.4}$ | $1769.3 \pm 243.6$ | $\mathbf{336.2 \pm 97.9}$ |
| Tanh | $1896.2 \pm 578.8$ | $194.9 \pm 36.0$ | $234.9 \pm 126.7$ | $109.7 \pm 47.5$ |
| RReLU | $1260.8 \pm 1259.2$ | $-9.3 \times 10^4 \pm 6.5 \times 10^4$ | $1151.3 \pm 200.7$ | $159.9 \pm 125.1$ |
| PReLU | $2123.3 \pm 1019.1$ | $-5.5 \times 10^5 \pm 4.5 \times 10^5$ | $1525.3 \pm 425.5$ | $146.2 \pm 13.8$ |
| Swish (SiLU) | $2161.9 \pm 806.8$ | $-7.6 \times 10^5 \pm 4.2 \times 10^5$ | $2135.2 \pm 614.1$ | $106.8 \pm 87.1$ |
| GeLU | $850.8 \pm 457.7$ | $-1.5 \times 10^7 \pm 2.6 \times 10^7$ | $766.1 \pm 144.8$ | $31.9 \pm 41.3$ |
| eLU | $274.9 \pm 2997.3$ | $-8.3 \times 10^5 \pm 8.6 \times 10^5$ | $-4.1 \times 10^4 \pm 7.4 \times 10^4$ | $116.5 \pm 102.7$ |
| CeLU | $521.3 \pm 897.6$ | $194.6 \pm 96.4$ | $42.2 \pm 248.5$ | $49.1 \pm 75.4$ |
| SeLU | $1437.2 \pm 618.1$ | $270.0 \pm 12.7$ | $342.3 \pm 17.4$ | $174.0 \pm 38.4$ |
| CReLU | $-472.6 \pm 684.6$ | $-1.0 \times 10^5 \pm 8.2 \times 10^4$ | $823.7 \pm 165.7$ | $12.1 \pm 4.6$ |
| Rational | $1526.4 \pm 317.9$ | $-6.1 \times 10^4 \pm 3.2 \times 10^4$ | $1083.3 \pm 368.6$ | $174.9 \pm 102.3$ |
| SwiGLU | $-4.6 \times 10^{10} \pm 6.0 \times 10^{10}$ | $-2.1 \times 10^{30} \pm 3.3 \times 10^{30}$ | $105.8 \pm 106.8$ | $118.0 \pm 27.4$ |
| Deep Fourier | $592.5 \pm 125.7$ | $176.3 \pm 41.5$ | $327.7 \pm 15.2$ | $185.9 \pm 47.0$ |
| Smooth-Leaky | $2622.4 \pm 536.5$ | $-2.2 \times 10^6 \pm 2.2 \times 10^6$ | $2202.9 \pm 533.6$ | $173.8 \pm 39.6$ |
| Rand. Smooth-Leaky | $\mathbf{3221.5 \pm 922.6}$ | $-6.0 \times 10^5 \pm 2.6 \times 10^5$ | $\mathbf{2791.2 \pm 320.2}$ | $187.2 \pm 19.4$ |

Table F2: Average reward per environment (Mean $\pm$ Std Dev) over 5 seeds. Higher is better. Best performance per environment is bolded. Large negative values (e.g., for SwiGLU or Humanoid) indicate potential issues.

**Plasticity Score Analysis.** Separately, we report a **Plasticity Score** that captures late-cycle *functional* performance on the training environments. This is distinct from the generalization gap; it answers: "can the agent still perform well after repeated shifts on the data it now collects?" To make this metric comparable across tasks, we normalize returns to $[0, 1]$ using robust global bounds derived from the entire experimental suite. We

apply a stability floor to clipped rewards (treating physics failures as 0.0) and report the IQM across seeds and environments.

Under this rigorous metric, the highest Plasticity Scores are obtained by *Rand. Smooth-Leaky* (0.388) and *Sigmoid* (0.340), followed by *Smooth-Leaky* (0.330) and *Swish* (0.315) (see Tab. F1 for full details). This ranking highlights a critical trade-off between *peak plasticity* and *safety*:

- **Rand. Smooth-Leaky** achieves the highest aggregate score by dominating in solvable locomotion tasks (`Ant`, `HalfCheetah`), demonstrating that smooth, randomized non-linearities facilitate superior gradient flow and rapid adaptation. However, it lacks an upper bound, which leads to divergence in the volatile `Humanoid` environment.

- **Sigmoid** achieves the second-best score via stability. Its bounded nature $(0, 1)$ prevents physics explosions in `Humanoid`, securing a baseline of performance where others fail. However, this saturation limits its peak learning capacity in simpler environments, resulting in a lower total score than the randomized variant.

Regarding transfer, high plasticity often correlates with a widening gap. *Sigmoid* and *Swish* show positive IQM $\Delta$ (gaps widen), suggesting that their adaptation is somewhat specific to the current stationary distribution. *Rand. Smooth-Leaky*, while failing in Humanoid, actually exhibits the lowest IQM $\Delta$ among high-performers in the environments where it remains stable, suggesting its randomized landscape encourages more generalizable solutions. Conversely, traditional activations like *ReLU* and *Leaky-ReLU* produce low Plasticity Scores, consistent with their known instability under repeated shifts.

Table F1 provides the raw absolute rewards for all activations. This data is essential for contextualizing the Generalization Gap (Tab. 4); a low gap should only be considered a "success" if the corresponding absolute reward in Tab. F1 indicates the agent has actually solved the task.

We therefore present both viewpoints: (i) **Plasticity Score** (IQM) for functional performance under non-stationarity, and (ii) **Generalization Gap** (IQM $\Delta$) for evaluation of the adaptation carried to perturbed tests. Full per-activation, per-environment results are in Tab. F3, and the cycle-by-cycle evolution is visualized in Fig. F1.

| Activation | HalfCheetah-v5 | Humanoid-v5 | Ant-v5 | Hopper-v5 | IQM $\Delta$ | Mean $\Delta$ | Std $\Delta$ |
|---|---|---|---|---|---|---|---|
| ReLU | $124.81 \pm 511.73$ | $203.06 \pm 158.47$ | $183.00 \pm 473.82$ | $\mathbf{-287.15 \pm 547.78}$ | 153.91 | 55.93 | 231.12 |
| Leaky-ReLU | $546.91 \pm 1425.72$ | $0.00 \pm 0.00$ | $-168.86 \pm 711.59$ | $-1.91 \pm 30.97$ | -0.95 | 94.04 | 312.12 |
| Sigmoid | $-288.53 \pm 2576.85$ | $18.92 \pm 111.38$ | $276.48 \pm 1018.48$ | $152.14 \pm 129.02$ | 85.53 | 39.75 | 242.81 |
| Tanh | $-467.58 \pm 1506.19$ | $-2.87 \pm 52.27$ | $21.61 \pm 213.56$ | $-49.31 \pm 117.34$ | -26.09 | -124.54 | 230.58 |
| RReLU | $527.09 \pm 1341.00$ | $38.56 \pm 1398.73$ | $-26.83 \pm 530.91$ | $21.89 \pm 69.33$ | 30.22 | 140.18 | 259.43 |
| PReLU | $839.60 \pm 596.03$ | $\mathbf{-316.28 \pm 2211.72}$ | $94.17 \pm 660.28$ | $-22.35 \pm 74.09$ | 35.91 | 148.78 | 491.86 |
| Swish (SiLU) | $533.55 \pm 1314.41$ | $0.00 \pm 0.00$ | $780.79 \pm 730.11$ | $13.39 \pm 37.27$ | 273.47 | 331.93 | 388.92 |
| GeLU | $317.71 \pm 730.89$ | $-9.92 \pm 245.15$ | $258.21 \pm 395.62$ | $-32.29 \pm 40.17$ | 124.15 | 133.43 | 180.32 |
| eLU | $237.41 \pm 740.39$ | $617.50 \pm 989.90$ | $-81.21 \pm 162.09$ | $-10.55 \pm 90.69$ | 113.43 | 190.79 | 315.58 |
| CeLU | $-341.13 \pm 459.03$ | $-49.31 \pm 74.86$ | $-281.51 \pm 277.68$ | $3.56 \pm 44.91$ | -165.41 | -167.10 | 169.68 |
| SeLU | $837.97 \pm 1628.38$ | $15.91 \pm 80.82$ | $-339.88 \pm 185.25$ | $51.90 \pm 52.95$ | 33.90 | 141.47 | 496.86 |
| CReLU | $-238.91 \pm 1039.91$ | $640.56 \pm 1255.50$ | $329.34 \pm 299.75$ | $1.61 \pm 3.40$ | 165.48 | 183.15 | 383.71 |
| Rational | $550.82 \pm 919.91$ | $862.09 \pm 2004.43$ | $270.39 \pm 526.94$ | $54.72 \pm 107.53$ | 410.60 | 434.51 | 350.01 |
| SwiGLU | $0.00 \pm 0.00$ | $0.00 \pm 0.00$ | $-326.59 \pm 186.38$ | $-16.59 \pm 44.16$ | -8.30 | -85.80 | 160.72 |
| Deep Fourier | $\mathbf{-568.95 \pm 551.62}$ | $0.34 \pm 58.99$ | $\mathbf{-477.68 \pm 208.27}$ | $41.53 \pm 50.34$ | **-238.67** | **-251.19** | 316.87 |
| Smooth-Leaky | $847.49 \pm 1611.57$ | $0.00 \pm 0.00$ | $44.09 \pm 1089.34$ | $27.01 \pm 72.58$ | 35.55 | 229.65 | 412.30 |
| Rand. Smooth-Leaky | $-49.38 \pm 883.36$ | $0.00 \pm 0.00$ | $-336.13 \pm 971.75$ | $-68.68 \pm 96.31$ | -59.03 | -113.55 | **151.18** |

Table F3: First-to-last cycle $\Delta$ of the Generalization Gap per environment and activation function (rounded to 2 decimals). The $\pm$ values indicate the 95% confidence interval. The IQM, mean, and std $\Delta$ are calculated across the 4 environments. Lower is better (negative values indicate the generalization gap decreased, improving plasticity). Best (lowest/most negative) per column is bolded.

## F.1 INTERPRETING NEGATIVE GENERALIZATION GAPS IN CONTINUAL RL

A strongly negative generalization gap at the end of training ($\text{GAP}_{c,e} < 0$) is not paradoxical under non-stationary streams (e.g., randomized MuJoCo friction (Dohare et al., 2024)). As the training environment keeps shifting, the agent can lose *plasticity*—it struggles to re-fit the *current* regime—so $R_{c,e}^{\text{train}}$ is depressed. Yet the policy may retain robust, regime-invariant skills that carry to perturbed test conditions, where evaluation is noise-free and does not suffer on-policy update instability. Consequently $R_{c,e}^{\text{test}}$ can exceed $R_{c,e}^{\text{train}}$, yielding a negative $\text{GAP}_{c,e}$. Our summary $\Delta(\text{GAP}_e)$ becomes highly negative when transferability improves over the training cycle even as within-cycle adaptation degrades—an acceptable and informative outcome in this setting.

Figure F1: The heatmap reports end-of-cycle $\mathrm{GAP}_{c,e}$ per activation (rows) and cycle (columns). Colors are centered at 0 (green = negative values, test > train; red = positive values, train > test). Values are computed on a held-out friction variant of the training environment. See Tab. F3 for the across-cycle summary $\Delta(\mathrm{GAP}_e)$. Together, these views reveal when apparent trainability gains translate (or fail to translate) into generalization.

## G  NOVEL ACTIVATION FUNCTION FORMULATIONS

Our characterization study enables the principled design of many novel activation functions; representative examples include:

| Activation | HDZ | NZG | Sat± | Sat− | $C^1$ | NonM | SelfN | $L/R_{slp}$ | $f''$ |
|---|---|---|---|---|---|---|---|---|---|
| RSeLU$^{\triangle}$ | – | ✓ | – | ✓ | – | – | ✓ | ✓ | ✓ |
| Bo-PReLU$^{\triangle}$ | – | ✓ | – | – | – | – | – | ✓ | – |

Table G1: Binary property grid (✓ = present, − = absent). **Abbreviations.** HDZ: hard dead zone; NZG: non-zero gradient for $x < 0$; Sat±: two-sided saturation; Sat−: negative-side saturation; $C^1$: first derivative continuous; NonM: non-monotonic segment; SelfN: self-normalizing output; L/R$_{\mathrm{slp}}$: learnable or randomized slope; $f''$: non-zero second derivative.
$^{\triangle}$ Proposed in this work.

### G.1  BOUNDED PRELU (BO-PRELU)

The **Bounded Parametric Rectified Linear Unit (Bo-PReLU)** is designed to combine the adaptability of PReLU with enhanced stability by constraining its learnable negative slope. Our case studies found that extreme slope values can be detrimental to performance. Bo-PReLU addresses this by forcing the slope to remain within a predefined "Goldilocks" range, preventing it from becoming excessively large or small.

The function follows the standard PReLU formulation:

$$f(x) = \begin{cases} x & \text{if } x \geq 0 \\ \alpha x & \text{if } x < 0 \end{cases} \tag{7}$$

The key innovation lies in how $\alpha$ is learned. It is constrained to the range $[\alpha_{\min}, \alpha_{\max}]$. To ensure this constraint is met without interfering with gradient-based optimization, we employ the reparameterization trick. An unconstrained parameter, $\alpha_{\mathrm{raw}}$, is learned, and the final slope is derived during the forward pass as:

$$\alpha = \alpha_{\min} + (\alpha_{\max} - \alpha_{\min}) \cdot \sigma(\alpha_{\mathrm{raw}}) \tag{8}$$

where $\sigma$ is the sigmoid function. This makes Bo-PReLU a robust and stable learnable rectifier.

Rand. Smooth-Leaky is only statistically significant with respect to Bo-PReLU on Random Label MNIST, CIFAR 5+1 and Continual ImageNet making Bo-PReLU highly competitive and highlighting the strengths of creating activation functions following the principles studied on Section 3 and Section 4. Optimal Hyperparameters for Bo-PReLU are indicated in Table G4, while the rest are in Table E2

### G.2  RANDOMIZED-SLOPE SELU (RSELU)

The **Randomized-Slope Scaled Exponential Linear Unit (RSELU)** is a hybrid activation function designed to merge the stochastic regularization benefits of RReLU with the training stability of SELU's self-normalization property.

| Activation | Permuted MNIST | Random Label MNIST | Random Label CIFAR | CIFAR 5+1 | Continual ImageNet |
|---|---|---|---|---|---|
| PReLU | $82.62 \pm 0.05$ | $92.67 \pm 0.23$ | $96.86 \pm 0.32$ | $43.30 \pm 0.61$ | $82.37 \pm 0.11$ |
| Bo-PReLU | $84.23 \pm 0.02$ | $91.57 \pm 0.11$ | $98.41 \pm 0.01$ | $48.15 \pm 1.20$ | $85.72 \pm 0.11$ |
| Rand. Smooth-Leaky | $\mathbf{84.26 \pm 0.02}$ | $\mathbf{93.33 \pm 0.05}$ | $\mathbf{98.42 \pm 0.01}$ | $\mathbf{57.01 \pm 1.59}$ | $\mathbf{86.23 \pm 0.13}$ |

Table G2: Total Average Online Task Accuracy (%) on Continual Supervised Benchmarks, averaged over 5 independent runs. Values are reported as mean $\pm$ standard deviation (SD). Statistical significance was determined using an independent two-sample Welch's t-test (p $< 0.05$).

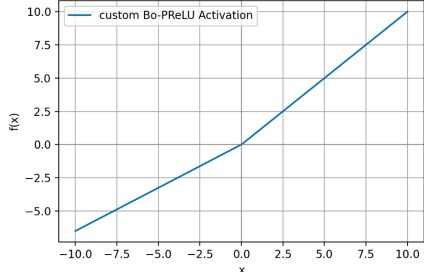

Figure G1: Bo-PReLU where $\alpha_{min} = 0.6$, $\alpha_{max} = 0.8$ and $\alpha_{init} = 0.65$.

The function has two modes of operation. During training, it introduces randomization to the negative slope:

$$f(x) = \begin{cases} \lambda x & \text{if } x \geq 0 \\ \lambda r(\exp(x) - 1) & \text{if } x < 0 \end{cases} \quad \text{where } r \sim \mathcal{U}(l, u) \tag{9}$$

During inference, the randomization is removed to ensure deterministic output, and the random variable $r$ is fixed to the mean of its distribution, $(l + u)/2$. A crucial feature of this design is that the bounds $l$ and $u$ are chosen to be symmetric around the original SELU alpha parameter ($\approx 1.6732$). This ensures that the self-normalizing property of SELU is preserved "in expectation" throughout training, providing a stable foundation while the slope randomization encourages robust learning and plasticity.

## H THE USE OF LARGE LANGUAGE MODELS (LLMS)

We used an LLM as a tool for early brainstorming, code debugging, and writing/editing of the earlier drafts of this paper. During ideation, we used it to enumerate experiment variants and sanity-check design choices; for code, we requested bug-finding hints and refactoring suggestions that we implemented only after manual review and testing; for text, we used it to improve clarity, organization, and grammar. The LLM did not generate novel research ideas, experiments, or results on our behalf; all methodological innovations, analyses, and conclusions are our own. We verified any technical claims, equations, and citations suggested during assisted drafting and we did not include uncited, model-generated text verbatim. No proprietary or personally identifiable data was provided to the LLM. The authors retain full responsibility for the content of this paper, and we affirm the novelty and originality of the work.

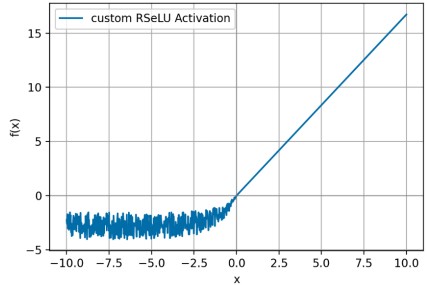

Figure G2: R-SeLU with bounds $r \sim \mathcal{U}(l, u)$ where $r = (0.9232, 2.4232)$.

| Activation | | Permuted MNIST | Random Label MNIST | Random Label CIFAR | CIFAR 5+1 | Continual ImageNet |
|---|---|---|---|---|---|---|
| SeLU | | $80.43 \pm 0.16$ | $79.95 \pm 0.91$ | $84.61 \pm 2.07$ | $49.07 \pm 1.25$ | $80.98 \pm 0.49$ |
| R-SeLU | | $81.72 \pm 0.04$ | $65.60 \pm 2.07$ | $47.23 \pm 0.65$ | $39.90 \pm 2.91$ | $79.87 \pm 0.13$ |
| Rand. | Smooth-Leaky | $\mathbf{84.26 \pm 0.02}$ | $\mathbf{93.33 \pm 0.05}$ | $\mathbf{98.42 \pm 0.01}$ | $\mathbf{57.01 \pm 1.59}$ | $\mathbf{86.23 \pm 0.13}$ |

Table G3: Total Average Online Task Accuracy (%) on Continual Supervised Benchmarks, averaged over 5 independent runs. Values are reported as mean $\pm$ standard deviation (SD). Statistical significance was determined using an independent two-sample Welch's t-test ($p < 0.05$).

| Activation | Permuted MNIST | Random Label MNIST | Random Label CIFAR | CIFAR 5+1 | Continual ImageNet |
|---|---|---|---|---|---|
| R-SeLU | [0.423, 2.923] ‖ 0.001 | [0.423, 2.923] ‖ 0.0001 | [0.423, 2.923] ‖ 0.001 | [1.6732, 1.6732] ‖ 0.001 | [1.6732, 1.6732] ‖ 0.001 |
| Bo-PReLU | layer, $\alpha = 0.75$, [0.5, 1.0] ‖ 0.001 | neuron, $\alpha = 0.65$, [0.3, 1.0] ‖ 0.001 | neuron, $\alpha = 0.75$, [0.5, 1.0] ‖ 0.001 | layer, $\alpha = 0.75$, [0.5, 1.0] ‖ 0.001 | neuron, $\alpha = 0.65$, [0.3, 1.0] ‖ 0.001 |

Table G4: Optimal Hyperparameters for extra custom activation function in each Continual Supervised Benchmark Problem. Represented as activation function shape parameter on the left side of the ‖ symbol and the learning rate to the right. Bo-PReLU's $\alpha$ indicates initial parameter value and $[l, u]$ bounds. R-SeLU only indicates bounds $[l, u]$.

