# OpenReview forum: "Activation Function Design Sustains Plasticity in Continual Learning"
_ICLR.cc/2026/Conference — ICLR 2026 Poster_

### Official Review · Reviewer_fK2M · 2025-10-25

**Soundness:** 3
**Presentation:** 2
**Contribution:** 3
**Rating:** 6
**Confidence:** 4

**Summary:**

This work presents two case studies to investigate the activation functions in the context of continual learning and identifies three principles for designing plasticity-friendly nonlinearities. Building upon these insights, the authors propose two new activation functions: **Smooth-Leaky** and **Randomized Smooth-Leaky**. Empirical studies on continual supervised learning and continual reinforcement learning demonstrate that the proposed activation functions achieve superior performance compared to existing alternatives.

**Strengths:**

- The case studies are interesting and thoughtfully executed, providing solid and well-supported conclusions.
- The large-scale empirical studies on continual supervised learning and continual reinforcement learning clearly demonstrate the superiority of the proposed activation function over prior alternatives.
- The proposed activation functions show promise for broader application in domains that rely on network plasticity.
- The authors also provide code to improve reproducibility.

**Weaknesses:**

- As the experiments already include multiple runs, I recommend that the authors report not only the mean results but also measures of statistical significance—such as standard deviation or 95% confidence intervals—for example, in Table 1 and Table 2.
- The continual reinforcement learning evaluation appears limited to a single experimental configuration. Expanding this analysis to include additional environments or settings would provide stronger support for the generality of the proposed activation functions.
- *Case Study 1* and *Case Study 2* are important parts of the paper, but too many related results and figures are placed in the appendix. To enhance readability, the authors might consider reorganizing the content despite the page limit.
- The term $C^1$ transition first appears in the introduction (line 60), but without explanation.

**Questions:**

- Do the proposed activation functions result in greater wall-clock time compared to existing activation functions?
- Could the authors provide the results of the **Smooth-Leaky** activation function in the continual reinforcement learning experiments?

---

> ### Author Response · Authors · 2025-11-20
> **Official Comment to Reviewer fk2m**
>
> We are grateful for the reviewer's encouraging comments regarding the thoughtful execution of our case studies, the strength of our empirical validation in CL and RL, and our commitment to reproducibility.
>
> ---
>
>  > I recommend that the authors report not only the mean results but also measures of statistical significance
>
> We have updated the caption of Table 1 and Table 2 to explicitly state that the +/- values represent the standard deviation (SD) over the 5 independent runs. We have also added to the caption that statistical significance was determined using an independent two-sample Welch's t-test with a significance level of $p < 0.05$.
> _Note: We also included this reply to Reviewer 5MGv_
>
> ---
>
> > The continual reinforcement learning evaluation appears limited to a single experimental configuration. Expanding this analysis to include additional environments or settings would provide stronger support.
>
> They are correct that the RL evaluation is focused on a single experimental configuration. We agree that expanding this analysis to additional settings would provide even stronger support for the generalizability of our findings.
> The primary reason for focusing on a single RL setting was Consistency with foundational literature, specifically by using the established non-stationary MuJoCo benchmark from papers like Dohare et al. (2024) [1] and Zhou et al. (2025) [2] to directly test our hypotheses where deep RL plasticity loss has already been studied. Secondary reasons include computational cost and variance (expanding to suites like Atari with many activations and a new algorithm sweep was resource-prohibitive) and isolating variables (fixing the algorithm and task allowed us to focus on the activation function as the central variable).
>
> * [1]. Dohare, S., Hernandez-Garcia, J. F., Lan, Q., Rahman, P., Mahmood, A. R., & Sutton, R. S. (2024). Loss of plasticity in deep continual learning. Nature, 632(8026), 768-774.
> * [2]. Zhou, H., Zhuang, Z. &; Wang, D.. (2025). Stay Hungry, Keep Learning: Sustainable Plasticity for Deep Reinforcement Learning. Proceedings of the 42nd International Conference on Machine Learning, in Proceedings of Machine Learning Research 267:79644-79672
>
> > 'Case Study 1 and Case Study 2 are important parts of the paper, ...'
>
> Since this has been requested by two independent Reviewers, we would like to propose a unified answer. *Please see comment titled: 'Case Study 1 and Case Study 2 enhancements'*
>
> ---
>
> > The term C^1 transition first appears in the introduction (line 60), but without explanation.
>
> We have added a definition at the term's first appearance (line 63).
>
> ---
>
> > Regarding wall-clock time
>
> We collected wall-clock time data for over 10 runs on **Continual ImageNet**, run on an HPC with a **CPU-only** configuration. Each run consisted of **5 independent Continual ImageNet experiments**.
>
> ### Continual ImageNet
> **Per-experiment wall-clock times (minutes; mean ± SD):**
>
> | Activation | Minutes / experiment |
> |---|---:|
> | ReLU | 47.0 ± 10.8 |
> | Leaky-ReLU | 47.0 ± 4.4 |
> | Swish | 45.2 ± 4.6 |
> | Smooth-Leaky | 44.7 ± 6.4 |
> | Rand. Smooth-Leaky | 52.8 ± 3.4 |
> | Rational | 90.8 ± 7.1 |
>
> **Per-5-experiment run totals (minutes; mean ± SD):**
>
> | Activation | Minutes / 5-experiment run |
> |---|---:|
> | ReLU | 235.2 ± 54.0 |
> | Leaky-ReLU | 235.2 ± 22.1 |
> | Swish | 226.0 ± 22.8 |
> | Smooth-Leaky | 223.6 ± 31.8 |
> | Rand. Smooth-Leaky | 264.2 ± 17.1 |
> | Rational | 453.9 ± 35.5 |
>
> **Takeaways.**
> - **Smooth-Leaky does not add wall-clock overhead.** It is:
>   - ~4.9% faster than ReLU and Leaky-ReLU,
>   - ~1.1% faster than Swish, and
>   - ~3.7% faster than the simple baseline average of {ReLU, Leaky-ReLU, Swish}.
> - **Randomized Smooth-Leaky adds overhead**, being:
>   - ~+12.4% vs. ReLU/Leaky-ReLU,
>   - ~+16.9% vs. Swish, and
>   - ~+13.8% vs. the baseline average.
>
> ---
>
> We also collected data on the **Continual RL benchmark**. We used **CPU-only**, with **10 experiments per activation**.
>
> ### Continual RL
>
> **Per-experiment wall-clock times (minutes; mean ± SD):**
>
> | Activation | Minutes / experiment |
> |---|---:|
> | ReLU | 418.9 ± 11.7 |
> | Leaky-ReLU | 438.3 ± 23.2 |
> | Swish | 382.1 ± 48.3 |
> | Smooth-Leaky | 379.1 ± 47.7 |
> | Rand. Smooth-Leaky | 424.1 ± 23.3 |
> | Rational | 491.7 ± 23.0 |
>
> **Takeaways.**
> - **Smooth-Leaky again shows no overhead.** It is:
>   - essentially the same as Swish (−0.8%),
>   - noticeably faster than ReLU (−9.5%), and
>   - faster than Leaky-ReLU (−13.5%).
> - **Randomized Smooth-Leaky** is:
>   - slower than the fastest baseline (+11.0% vs. Swish),
>   - about even with ReLU (+1.2%), and
>   - faster than Leaky-ReLU (−3.2%).
> - **Rational** remains the slowest by a large margin.
>
> ---
>
> > 'Could the authors provide the results of the Smooth-Leaky in the CRL experiments?'
>
> We are now including Smooth-Leaky in Tables 3 and 4.

---

> > ### Comment · Reviewer_fK2M · 2025-11-28
> >
> > Thank you for the clarifications and for addressing my concerns. I will keep my current score and recommend the paper for acceptance.

---

### Official Review · Reviewer_5MGv · 2025-10-31

**Soundness:** 2
**Presentation:** 1
**Contribution:** 3
**Rating:** 4
**Confidence:** 4

**Summary:**

The paper argues that activation choice is a primary lever for sustaining plasticity in continual learning. It presents a property-level analysis (negative-branch responsiveness, saturation, “dead-band” width) of activation functions and proposes two drop-in $C^1$ activations, Smooth-Leaky and Randomized Smooth-Leaky, that maintain a non-zero derivative floor. The paper evaluates the activation functions on standard Continual Learning tasks.

**Strengths:**

The paper provides a substantial empirical study on the properties of many activation functions under the lens of continual learning. The case studies are well thought out and well conducted, they isolate negative-branch slope, saturation, and “dead-band width” as key predictors under shift. The study of desaturation dynamics under shocks is an important diagnostic of activation functions and ties well activation shape to recovery behavior after controlled shifts.
The suggested drop-in replacements are intuitive and easy to implement.

**Weaknesses:**

In general, the paper is verbose and somewhat difficult to parse. Table 2 reports statistical significance but does not provide confidence intervals or which statistical tests were used. There also appears to be an imbalance in the hyper-parameter sweeps: the two proposed activation functions are explored across 175 and 450 combinations, respectively, whereas other hyper-parameters are examined over only a handful of settings. This imbalance could bias the experimental results.

**Questions:**

What distribution over pre-activations is used for the effective negative slope in Figure C-1(per-layer? per-epoch? across tasks?), and how sensitive are the conclusions to that choice? Please report variability across layers/epochs and any normalization applied before computing.

---

> ### Author Response · Authors · 2025-11-20
> **Official Comment to Reviewer 5MGv**
>
> We thank the reviewer for recognizing the value of our analysis on desaturation dynamics and for highlighting our success in isolating key predictors of plasticity, such as negative-branch slope and saturation.
>
> ---
> > Table 2 reports statistical significance but does not provide confidence intervals or which statistical tests were used.
>
> We have updated the caption of Table 1 and Table 2 to explicitly state that the +/- values represent the standard deviation (SD) over the 5 independent runs. We have also added to the caption that statistical significance was determined using an independent two-sample Welch's t-test with a significance level of $p < 0.05$.
> *Note: We also included this reply to Reviewer fk2M*
>
> ---
>
> > There also appears to be an imbalance in the hyper-parameter sweeps: the two proposed activation functions are explored across 175 and 450 combinations, respectively, whereas other hyper-parameters are examined over only a handful of settings. This imbalance could bias the experimental results.
>
> Since this has been commented by two independent Reviewers, we would like to propose a unified answer. Please see comment titled: *'Hyper-parameter sweeps and extra combinations'*
>
> ---
>
> > What distribution over pre-activations is used for the effective negative slope in Figure C-1(per-layer? per-epoch? across tasks?), and how sensitive are the conclusions to that choice?
>
> *Note: Figure C-1 is now Figure 1 in Section 3*
>
>
> The effective negative slope $\overline{s} = \mathbb{E}_{x<0}[\phi'(x)]$ in Figure 1-A is computed using a **static, synthetic pre-activation distribution**. It is **not** an empirical distribution sampled from network layers during training.
>
> Specifically, we generated 10,000 samples from a standard normal distribution, $\mathcal{N}(0, 1)$, and then filtered for all negative samples (approximately 5,000 points). The effective slope $\overline{s}$ for a given activation-function configuration (e.g., Leaky-ReLU with $\alpha = 0.7$) was then calculated as the mean of its first derivative over this fixed set of negative samples.
>
> This is a deliberate design choice. We used a static, synthetic distribution to isolate the activation’s **intrinsic shape** as the independent variable. This follows our main line of reasoning throughout the paper: to isolate and characterize activation-function properties so they can be studied in relation to loss of plasticity.
>
> **Note 1.** We considered using an empirical distribution (e.g., sampled per-layer, per-task). However, this approach would increase complexity and mix multiple variables. The empirical distribution of pre-activations is itself a result of training dynamics, which are influenced by the activation function, the current task, and the training epoch. Plotting final accuracy against an $\overline{s}$ derived from a changing empirical distribution would make it impossible to disentangle the cause (the activation’s shape) from the effect (the network’s final performance and internal state).
>
> **Note 2.** By using a fixed, common distribution ($\mathcal{N}(0, 1)$), we can project all activation functions onto a standardized axis. This allows for a shape-matched comparison. Figure 1-A therefore isolates the relationship:
> > >*_How does the intrinsic average negative-side gradient of a function (our independent variable) correlate with the total final accuracy of a network trained with that function (our dependent variable)?_*
>
> ---
>
> > Please report variability across layers/epochs and any normalization applied before computing.
>
> The distribution is intrinsically normalized, as it is derived from a standard normal distribution. No other normalization was applied.
> The X-axis ($\overline{s}$) in Figure 1-A has zero variability across layers or epochs. As previously mentioned, tt is computed analytically once for each activation's specific hyperparameter configuration (e.g., the point for Leaky-ReLU with $\alpha$=0.7 is fixed). The plot's Y-axis shows the empirical result (mean and standard deviation) of the full 20-task continual learning run using that specific activation.
>
> **Note 3.** The evolution of an empirical slope is analyzed separately, but only for PReLU in Figures C-2 and C-3, to show that its learnable parameters drift out of the optimal zone.

---

### Official Review · Reviewer_hFaa · 2025-11-01

**Soundness:** 3
**Presentation:** 3
**Contribution:** 2
**Rating:** 6
**Confidence:** 4

**Summary:**

The authors provide a thorough investigation on the role of activation functions in addressing plasticity loss in continual learning. They propose two activation functions, Smooth-Leaky and Randomized Smooth-Leaky, which involve maintaining a non-zero derivative floor, optimizing negative-side responsiveness and ensuring smooth transitions at the origin. They evaluate these activations across supervised continual learning benchmarks and RL environments, showing their effectiveness in maintaining plasticity.

**Strengths:**

- The main strength of the paper lies in its rigorous evaluation of activation functions across continual learning benchmarks (supervised and RL).
- Proposed activation functions, and the concept of a "Goldilocks zone" for negative-side responsiveness, offer a interesting perspective on mitigating plasticity loss in continual learning. ​The findings highlight the role of activation functions in maintaining plasticity.
- Metrics like saturation fraction and dead-band width provide valuable insights into activation function behavior under non-stationary settings.
- The paper is well-structured and clearly written, with visualizations, explanations.

**Weaknesses:**

- While the paper discussed impact unconstrained adaptive slopes (e.g., PReLU), it does not explore alternative methods to guide parameters into the "Goldilocks zone" which itself is primarily identified empirically, but lacks theoretical grounding or formal analysis. Therefore, explaining why this range is optimal would help the paper overall.
- The paper does not explore how activation functions interact with different optimizers or learning rate schedules. For example, adaptive optimizers like Adam or RMSProp may behave differently with the proposed activations. Are the proposed activations universally effective across all continual learning settings? Since the dynamics of plasticity loss may vary significantly between tasks, a more nuanced analysis of task-specific performance could provide deeper insights.
- The reinforcement learning experiments focus solely on MuJoCo tasks, Have authors tried running experiments on domains like sparse reward or high-dimensional tasks?

**Questions:**

Please refer to the comments in the weaknesses section.

---

> ### Author Response · Authors · 2025-11-20
> **Official Comment to Reviewer hFaa I**
>
> We thank the reviewer for explicitly appreciating our 'Goldilocks zone' perspective on negative-side responsiveness and for recognizing the value of our saturation fraction and dead-band width metrics.
>
> ---
>
> >While the paper discussed impact unconstrained adaptive slopes (e.g., PReLU), it does not explore alternative methods to guide parameters into the "Goldilocks zone" which itself is primarily identified empirically, but lacks theoretical grounding or formal analysis. Therefore, explaining why this range is optimal would help the paper overall.
>
> We thank the reviewer for this comment, which gives us the opportunity to clarify the intended role of the “Goldilocks zone” in our paper. The reviewer is correct that the “Goldilocks zone” is presented as an **empirical observation**, not the central theoretical claim of our work. Our paper’s primary hypothesis is that plasticity failure is driven by specific, measurable activation properties. The “Goldilocks zone” is the **consequence** of these properties, not the cause.
>
> Our core theoretical claims, which we investigate in Sections 3 and 4, are that plasticity is governed by two primary failure modes:
>
> 1. **Failure Mode 1 (Slopes are too low, $\overline{s} \to 0$).**
>    As the negative slope approaches zero, the network suffers from “gradient starvation.” Our results show a strong negative correlation (Pearson $r = -0.51$) between final accuracy and the “dead unit fraction” (see Figure 1-B).
>
> 2. **Failure Mode 2 (Slopes are too high, $\overline{s} \ge 0.9$).**
>    Conversely, as the negative slope becomes too large, performance also degrades. Our analysis in Figure 1-C and 1-D links this to a sharp increase in landscape stiffness, evidenced by a rising dominant Hessian eigenvalue ($\lambda_{\max}$).
>
> Therefore, the “Goldilocks zone” is the empirically observed range where **neither** of these two failure modes dominates. It represents a trade-off: maintaining a non-zero derivative floor to avoid unit death, while remaining moderate enough to avoid explosive landscape curvature.
>
> Regarding the reviewer’s second point on exploring methods to guide parameters:
>
> Our main analysis of adaptive slopes (PReLU) was intended to show that **unconstrained adaptation fails to find this optimal balance**, as the learnable slopes tend to drift toward zero and out of the high-performing zone (see Section C.2, Figures C2 and C3). While a full exploration of constrained-adaptive methods was outside our primary scope, we did design **Bounded PReLU (Bo-PReLU)** (Appendix G.1) as a proof-of-concept specifically to address this. Bo-PReLU is designed to combine the adaptability of PReLU with enhanced stability by using a reparameterization trick to **force the slope to remain within a bounded range** that could empirically be matched to the “Goldilocks zone.” We proposed it in the appendix to showcase that the characterization developed throughout the paper can be used to build new activation functions from first principles.
>
> We have updated results regarding Bo-PReLU and compared them to their inspiration (PReLU) and our other custom activation Rand. Smooth-Leaky (Section G). We show that Bo-PReLU is highly competitive and that it strengthens the conclusion that new activations can be developed directly from the ideas studied in the paper.
>
> ---
>
> > The paper does not explore how activation functions interact with different optimizers or learning rate schedules.
>
> Our primary methodological goal for this study was to isolate the effect of activation functions on plasticity. To achieve this, it was necessary to fix other key components of the training pipeline—such as the optimizer and learning rate—in order to create a controlled experimental setup.
>
> We chose the **Adam** optimizer for two specific and deliberate reasons:
>
> 1. **Consistency with prior work.**
>    Our continual supervised learning benchmarks are adapted from recent literature, such as Kumar et al. (2023) [1], which used Adam for their experiments.
>
> 2. **A “stress test” for plasticity.**
>    Adaptive optimizers like Adam have been implicated in the literature as a contributing factor to plasticity loss. Adam can struggle to update weights with large magnitudes (Dohare et al., 2021) [2], and lose its ability to learn on non-stationary tasks (Lyle et al., 2023) [3].
>
> We fully agree with the reviewer that the interaction between different optimizers (e.g., SGD, RMSProp) and activation shapes is an important area for future investigation. We do not claim that Adam is the optimal choice for all activations; rather, we show that under the same fixed and challenging optimization conditions, the choice of activation function is a fundamental and powerful tool for preserving a network’s ability to learn.
>
> We have also included an explanation along the lines of the one given here in the appendix, now under **Section B.1** of the revised manuscript, titled *Limitations and implications of choosing an optimizer*.

---

> ### Author Response · Authors · 2025-11-20
> **Official Comment to Reviewer hFaa II**
>
> > Are the proposed activations universally effective across all continual learning settings? Since the dynamics of plasticity loss may vary significantly between tasks, a more nuanced analysis of task-specific performance could provide deeper insights.
>
> To answer the first point: **no**, we do not claim the proposed activations are *universally* the single best performer in every setting. The design space is too vast, and performance can be sensitive to other components, as the reviewer notes. However, our results in **Table 2** show that our proposed **Rand. Smooth-Leaky** is the top-performing activation in **4 out of 5 supervised benchmarks** and also achieves the highest **Plasticity Score** in our RL experiments.
>
> This consistent, top-tier performance leads directly to the reviewer’s second point about a *more nuanced, task-specific analysis*. Specifically, we categorized our supervised benchmarks into two distinct regimes: **Concept Shift** and **Input Shift**.
>
> - **Concept shift** tasks, such as *Random Label MNIST* and *Random Label CIFAR*, stress the network’s ability to sever existing mappings and memorize arbitrary new ones. In these settings, which require extreme plasticity, **Rand. Smooth-Leaky** consistently achieved the highest performance (e.g., **93.33%** on Random Label MNIST and **98.42%** on Random Label CIFAR).
>
> - **Input shift** tasks like *CIFAR 5+1* and *Continual ImageNet* test the ability to adapt to novel feature distributions and class expansions. Here too, our method proved robust, achieving top performance (e.g., **57.01%** on CIFAR 5+1 *(second to DFFs)* and **86.23%** on Continual ImageNet).
>
> Together, this demonstrates that the properties we identified are fundamental to plasticity: they allow the network to sustain its learning capability whether the challenge is **pure memorization** (concept shift) or **adapting to novel domains** (input shift).
>
> ---
> > The reinforcement learning experiments focus solely on MuJoCo tasks. Have authors tried running experiments on domains like sparse reward or high-dimensional tasks?
>
> We thank the reviewer for this constructive feedback. They are absolutely correct that the current RL evaluation is focused on a single experimental configuration. We agree that expanding the analysis to additional settings—especially sparse reward and high-dimensional domains—would provide vital support for the generality of our findings.
>
> The primary reason for focusing on a single RL setting was **consistency with foundational literature**, specifically by using the established non-stationary MuJoCo benchmark from papers like Dohare et al. (2024) [4] and Zhou et al. (2025) [5] to directly test our hypotheses in a setting where deep RL plasticity loss has already been studied. Secondary reasons include:
>
> - **Computational cost and variance.** Expanding to suites like Atari (studied with respect to loss of plasticity in Lyle et al. (2022) [6] and Sokar et al. (2023) [7]) with many activations and a new algorithm sweep was resource-prohibitive.
> - **Isolating variables.** Fixing the algorithm and task allowed us to focus on the activation function as the central variable.
>
> We acknowledge the necessity of testing generalizability across sparse domains. We believe this is a critical and promising avenue for future work that will build directly on our findings. To this end, we are currently working on future research that includes sparse reward environments like **MiniGrid** to explicitly test whether our activation-design principles hold across this domain boundary and other loss-of-plasticity research directions.
>
> ---
>
> * [1].Saurabh Kumar, Henrik Marklund, and Benjamin Van Roy. Maintaining plasticity in continual learning via regenerative regularization. arXiv preprint arXiv:2308.11958, 2023.
> * [2]. Shibhansh Dohare, Richard S Sutton, and A Rupam Mahmood. Continual backprop: Stochastic gradient descent with persistent randomness. arXiv preprint arXiv:2108.06325, 2021.
> * [3]. Clare Lyle, Zeyu Zheng, Evgenii Nikishin, Bernardo Avila Pires, Razvan Pascanu, and Will Dabney. Understanding plasticity in neural networks. arXiv preprint arXiv:2303.01486, 2023.
> * [4]. Dohare, S., Hernandez-Garcia, J. F., Lan, Q., Rahman, P., Mahmood, A. R., & Sutton, R. S. (2024). Loss of plasticity in deep continual learning. Nature, 632(8026), 768-774.
> * [5]. Zhou, H., Zhuang, Z. &; Wang, D.. (2025). Stay Hungry, Keep Learning: Sustainable Plasticity for Deep Reinforcement Learning. Proceedings of the 42nd International Conference on Machine Learning, in Proceedings of Machine Learning Research 267:79644-79672
> * [6]. Lyle, C., Rowland, M., & Dabney, W. (2022). Understanding and preventing capacity loss in reinforcement learning. arXiv preprint arXiv:2204.09560.
> * [7]. Sokar, G., Agarwal, R., Castro, P. S., & Evci, U. (2023, July). The dormant neuron phenomenon in deep reinforcement learning. In International Conference on Machine Learning (pp. 32145-32168). PMLR.

---

### Official Review · Reviewer_kvHD · 2025-11-02

**Soundness:** 3
**Presentation:** 3
**Contribution:** 3
**Rating:** 6
**Confidence:** 3

**Summary:**

The authors explore the role of activation function design on plasticity in continual learning. The authors introduce the Smooth-Leaky and the Randomized Smooth-Leaky activation functions.The authors evaluate their proposed activation functions in the supervised class incremental continual learning benchmarks and in RL environments and show that the randomized smooth leaky activation function consistently outperforms existing activation functions.

**Strengths:**

- The empirical results clearly show that the proposed activation function randomized smooth-leaky consistently attains superior average online task accuracy.
- The experiments consider a comprehensive set of baseline or existing activation functions.
- The discussion and experiments regarding the generalization gap, specifically, that plasticity loss can manifest in reduced training loss and test loss in different ways, is a positive touch to the paper which is not usually evaluated in this space. As the authors point out, maximizing plasticity via maximizing training loss may result in over-fitting. While earlier papers have argued that solely analyzing test loss in continual settings can be confounded by issues of regularization and generalization. The focus that the authors put here is a strength of this paper.

**Weaknesses:**

- There is a paper by Lewandowski et al. "Plastic Learning with Deep Fourier Features" which also examines the role of activation function design on plasticity. It would be useful if the authors could comment on this line of work, and ideally, incorporate this paper's contribution as a baseline competitor.
- It would be worthwhile to apply a few existing SOTA methods for plasticity loss such as reset-based: SNR and regularization-based: L2 Init, and to generate a table with results for a reasonable set of METHOD X + ACTIVATION FUNCTION Y. When given an additional continual learning intervention how important is the choice of activation function?

**Questions:**

- Given the recent popularity of the SwiGLU, why isn't this activation function evaluated?

---

> ### Author Response · Authors · 2025-11-20
> **Official Comment to Reviewer kvHD I**
>
> We thank the reviewer for specifically validating our discussion on the generalization gap and for recognizing the importance of distinguishing how plasticity loss manifests differently in training versus test loss.
>
> ---
>
> > There is a paper by Lewandowski et al. "Plastic Learning with Deep Fourier Features" which also examines the role of activation function design on plasticity. It would be useful if the authors could comment on this line of work, and ideally, incorporate this paper's contribution as a baseline competitor.
>
> We thank the reviewer for pointing out this relevant line of work. We were not aware of it. We have now added DFF to the results in the updated draft.
>
> ---
>
> ## Comment on DFF and results
>
> We found the DFF proposal particularly interesting, and we would like to take this opportunity to connect it to our work, since it can be analyzed directly through the three design rules for plasticity-friendly nonlinearities that we propose in Section 5:
>
> 1. **(i) Maintain a non-zero derivative floor.**
>    DFF aligns perfectly with this rule. By using a concatenated block
>    $f(z) = [\sin(z), \cos(z)]$,
>    the gradient vector
>    $f'(z) = [\cos(z), -\sin(z)]$
>    is never zero. This mechanism, like ours, should avoid wide analytic dead bands ($|\varphi'|<10^{-3}$), which our paper identifies as a key predictor of non-recovery from shocks.
>
> 2. **(ii) Keep responsiveness in a moderate “Goldilocks zone.”**
>    This is the critical point of divergence. Our work (Case Study 1) provides empirical evidence for a stable, moderate, linear slope. We show that this “Goldilocks zone” is an optimal trade-off that avoids two distinct failure modes:
>    - **dead units** (as $\overline{s} \to 0$), and
>    - **landscape stiffness / curvature explosion** (as $\overline{s} \ge 0.9$).
>    DFF proposes a fundamentally different mechanism: a dynamic, oscillating gradient (“adaptive linearity”) via periodic functions.
>
> 3. **(iii) Prefer a $C^1$ (smooth) transition.**
>    DFF also aligns with this rule. The function is infinitely differentiable ($C^\infty$), avoiding the abrupt “kink” of $C^0$ rectifiers (like Leaky-ReLU) that we identify as less ideal.
>
> Regarding implementation, the DFF paper uses a **“doubling producer”** architecture, which matches the design used by **CReLU**. Our own experiments (Table 2) found that CReLU—despite satisfying Rule (i) via a non-zero floor—underperforms our proposed methods. This strongly suggests that **simply having a non-zero gradient is insufficient**.
>
> Adding DFF provides strong support to our analysis. In the Continual SL benchmarks, DFF proves to be a highly competitive baseline, confirming that its mechanism is effective. It is, however, outperformed by our **Rand. Smooth-Leaky** in 4 out of 5 tasks (e.g., **86.23% vs. 76.03%** on Continual ImageNet, and **93.33% vs. 92.61%** on Random Label MNIST). Interestingly, DFF shows exceptional performance on the **CIFAR 5+1** task (**72.29% vs. 57.01%**), indicating that its mechanism is clearly beneficial in that specific non-stationary setting.
>
> This exception aside, the overall trend is even clearer in the RL domain: **Rand. Smooth-Leaky** achieves a final Plasticity Score of **1310.94**, compared to **266.16** for DFF.
>
> Overall, this confirms our intuition. DFF, through a different line of reasoning, lands on a method that validates one of our core principles (**a non-zero gradient is critical**), but our proposed mechanism appears to be a more robust and generally effective solution for sustaining plasticity than an oscillatory one. The door remains open for further research to understand the differences more deeply.
>
> ---
>
> > Regarding application of a few existing SOTA methods
>
> We have added a targeted experiment (in a new **Appendix Section E.3: ‘Continual Learning Interventions Targeting Loss of Plasticity’**) showing the performance of **L2 Init, SNR and EWC** when combined with all activation functions studied throughout the paper to address how our work interacts with other CL interventions. Results are reported in **Tables E.4 - E.11**.
>
> To sum it up, the experimental results demonstrate that while L2-Init is the most universally effective method for rescuing weak activations (like ReLU), the choice of activation function ultimately dictates the maximum attainable performance. _Our custom Rand. Smooth-Leaky activation consistently yields the highest performance ceiling, particularly on complex natural image tasks (Continual ImageNet)_. Crucially, the CIFAR 5+1 benchmark reveals a "constraint paradox" where applying plasticity interventions to naturally plastic activations actually harms performance, proving that in cases requiring significant parameter drift, our custom activation functions, and other vanilla activations, perform better in isolation than standard CL algorithms do with baselines.

---

> ### Author Response · Authors · 2025-11-20
> **Official Comment to Reviewer kvHD II**
>
> # Questions
>
> > Given the recent popularity of the SwiGLU, why isn't this activation function evaluated?
>
> We thank the reviewer for bringing this activation back to our attention. We agree that incorporating it is a useful comparison. Therefore, we have now added **vanilla SwiGLU** to our Continual Supervised Benchmark and Continual RL experiments; results will appear in the updated rebuttal version. Nonetheless, we report here that **SwiGLU underperforms both Smooth-Leaky and Rand. Smooth-Leaky in all settings** (e.g., comparisons between SwiGLU and R.-SL are **63.572 vs 86.232** on Continual ImageNet, and **9.565 vs 57.01** on CIFAR 5+1, respectively).
>
> ### Why we did not originally evaluate SwiGLU
>
> - **SwiGLU is a gated FFN nonlinearity:** SiLU on a “gate” branch multiplies a linear “value” branch. It has been shown to improve Transformer feed-forward layers and is widely adopted in PaLM and LLaMA. [1,2,3]
>
> - **Our initial focus was different.**
>   Our study initially targeted **unary, shape-controlled activations** in CNN/MLP backbones to test hypotheses tied to plasticity—negative-branch slope, dead-band width, saturation, and their impact on loss of plasticity in continual settings.
>
> - **Not a plug-in unary activation.**
>   SwiGLU is **not** a drop-in unary nonlinearity: it requires a two-branch projection and width change, which complicates strict apples-to-apples parameter/FLOP parity in non-Transformer backbones and was outside our original scope.
>
> - **No intrinsic shape-control parameter.**
>   SwiGLU has no built-in shape-control hyperparameter. Thus, to test **shape–plasticity causality** (our core focus), one would need a **parametric variant** rather than vanilla SwiGLU.
>
> ### Theoretical connection to our principles
>
> From a theoretical standpoint, SwiGLU’s design conflicts with the core principles our paper identifies for sustaining plasticity. We argue for activation shapes that maintain a **strict, non-zero derivative floor** to prevent “dead units” and ensure rapid desaturation (Secs. 3–5). SwiGLU’s multiplicative gating
> \[
> \text{Swish}(z_{\text{gate}}) \otimes z_{\text{value}}
> \]
> does the opposite: it introduces a **dynamic, coupled dead-zone**. Specifically:
>
> - The gradient for the **gate** branch is nullified if the **value** branch output is zero.
> - The gradient for the **value** branch is nullified if the **gate** (via SiLU) closes.
>
> This creates high-risk situations for gradient flow, which is precisely the failure mode our proposed activations (Smooth-Leaky, Rand. Smooth-Leaky) are designed to mitigate.
>
> ### Scope note
>
> Finally, given the vast design-space of activation functions (e.g., the 400 surveyed by Kunc et al. [4]), we prioritized the most commonly used activations that were also best suited to be scrutinized under the mechanisms most directly tied to plasticity theory, as hypothesized in our paper.
>
> * [1] Shazeer, N. (2020). *Glu variants improve transformer.* arXiv preprint arXiv:2002.05202.
> * [2] Touvron, H., Lavril, T., Izacard, G., Martinet, X., Lachaux, M. A., Lacroix, T., ... & Lample, G. (2023). *Llama: Open and efficient foundation language models.* arXiv preprint arXiv:2302.13971.
> * [3] Chowdhery, A., Narang, S., Devlin, J., Bosma, M., Mishra, G., Roberts, A., ... & Fiedel, N. (2023). *Palm: Scaling language modeling with pathways.* Journal of Machine Learning Research, 24(240), 1–113.
> * [4] Kunc, V., & Kléma, J. (2024). *Three decades of activations: A comprehensive survey of 400 activation functions for neural networks.* arXiv preprint arXiv:2402.09092.

---

### Official Review · Reviewer_mY97 · 2025-11-05

**Soundness:** 2
**Presentation:** 1
**Contribution:** 2
**Rating:** 2
**Confidence:** 4

**Summary:**

This paper looks at the effect of various characteristics of activation functions on the plasticity of neural networks on supervised continual learning and RL benchmarks. These characteristics include things like the slope of the activation function for negative inputs, where they identify a “Goldilocks zone,” and dead-band width which is the portion of the activation function domain that has a small gradient. They propose two different activation functions, Smooth-Leaky and Randomized Smooth-Leaky which are modifications of the Leaky-ReLU activation function with a smooth transition region rather than a kinked one.

**Strengths:**

- The initial part of the paper with the case studies on the negative slope and dead band width are valuable experiments for the community.
- The diversity of metrics and analyses presented for the experiments is quite high.

**Weaknesses:**

- The presentation of the paper can use some work. Most of the results for the experiments in Sections 3 and 4 are not presented in the main paper, and are instead presented in an Appendix. This is fine for one or two auxiliary results, but here, most of the main results for a significant portion of the paper are presented elsewhere. I’d ask that either the authors figure out a way to present a coherent story with results in the main paper, either by moving some parts of the paper to the appendix (not just the graphs), or if that is not possible in the page limit, consider another venue.
- The paper introduces two activation functions Smooth-Leaky and Randomized Smooth-Leaky. These activation functions introduce 3 and 4 new hyperparameters respectively which are searched for in a grid, with 175 and 450 combinations respectively. Compare this to the baseline of Leaky-ReLU which had 12 configurations evaluated. Furthermore, these hyperparameters were individually tuned on each setting. The amount of extra tuning given to the new methods is an unfair advantage, and could definitely be the cause of any improvement in performance rather than something inherent about the methods themselves.
- The metrics used to measure plasticity in the RL section seem to be flawed. If we look at just the rewards, Sigmoid seems to significantly outperform the other activation functions on 2/4 environments. Yet the plasticity score seems designed in a way to remove that effect, taking the median across environments to result in a number where the proposed method does better. A better approach could be to do some kind of normalization of the return and presenting the mean. Taking the median across 4 environments specifically excludes the environment where none of the activation functions except for Sigmoid are able to learn.
- For generalizability, there’s another metric presented which is a non-standard metric. It looks at the gap between train performance and a test environment which is some perturbation of the train environments. The metric reported compares the difference between the gap after cycle 3 and cycle1. First, it’s unclear what the perturbations are. Second, this metric has no notion of what the actual reward obtained in each of the settings was. A small gap with low absolute rewards is less preferable to a large gap with high absolute rewards, but this metric doesn’t capture that. Finally, measuring generalizability is much easier in the supervised setting, and has been done in several prior works. The results on those settings should be reported before creating such a metric.
- Tables 1, 2, and 4 are missing standard deviations.

**Questions:**

- Could you further clarify the reasoning behind the criteria for dead units defined for non-ReLU activation functions? For ReLU it makes sense because there is a region where the value of the activation function is 0 with slope 0. That doesn’t seem to be the case with most of the other activation functions studied, and it seems a bit odd to talk about dead-unit fractions.

---

> ### Author Response · Authors · 2025-11-20
> **Official Comment to Reviewer mY97 I**
>
> We thank the reviewer for explicitly recognizing that our case studies on negative slope and dead-band width constitute valuable experiments for the community.
>
> ---
>
> > Regarding: 'The presentation of the paper can use some work. Most of the results for the experiments in Sections 3 and 4 are not presented in the main paper, and are instead presented in an Appendix.'
>
> We agree with the reviewer. We had many essential figures and results placed in the appendix. In the original submission, we tried our best to fit everything necessary to understand the paper within the strict 9-page limit; however, works that include multiple experiments and ideas that are all necessary to explain the reasoning behind tend to be difficult to fit within the limits. Now, we believe that the extra page has given us sufficient room to introduce new figures, rephrase explanations for better readability, and add valuable new baselines requested during the review process.
>
> Since this has been requested by two independent Reviewers, we would like to propose a unified answer. Please see comment titled: **'Case Study 1 and Case Study 2 enhancements'**.
>
> ---
>
> > "The paper introduces two activation functions Smooth-Leaky and Randomized Smooth-Leaky. These activation functions introduce 3 and 4 new hyperparameters respectively which are searched for in a grid, with 175 and 450 combinations respectively, ..."
>
> Since this has been commented by two independent Reviewers, we would like to propose a unified answer. Please see comment titled: **'Hyper-parameter sweeps and extra combinations'**.
>
> ---
>
> > Regarding the metrics used to measure plasticity in the RL section seem to be flawed.
>
> We fully agree that our previous use of the raw median inadvertently masked catastrophic failures in specific environments (specifically **Humanoid-v5**) and prevented a fair assessment of stability. Following the reviewer’s suggestion, we have revised our evaluation protocol to ensure it captures both peak performance and failure cases.
>
> ---
>
> ## 1. Methodological Improvements
>
> Instead of the raw median, we have implemented a rigorous **Min–Max Normalization** protocol following best practices in Agarwal et al. (2021) [16]:
>
> - **Normalization.**
>   We normalize rewards in every environment to a **$[0,1]$** scale using robust global bounds derived from the entire experimental suite. This resolves the scale disparity (e.g., Humanoid vs. Hopper) that the reviewer correctly identified.
>
> - **Handling physics failures.**
>   As the reviewer noted, Humanoid-v5 is prone to divergence. We observed that plasticity loss often leads to physics simulation instabilities (rewards $\ll -10^6$). To allow meaningful aggregation, we implemented a **stability floor**: runs that exhibit physics explosions are treated as functional failures (reward clipped to the minimum bound), resulting in a normalized score of **$0.0$** rather than a numerical artifact.
>
> - **Aggregation (IQM).**
>   We now report the **Interquartile Mean (IQM)** and the **Mean with bootstrap confidence intervals**. IQM remains sensitive to the magnitude of failures (unlike the median) while still being robust to statistical outliers.
>
> ---
>
> ## 2. New Insights
>
> Implementing the reviewer’s suggested normalization revealed a critical and nuanced **trade-off between stability and plasticity** that was previously obscured:
>
> - **Sigmoid (the stability specialist).**
>   The reviewer was correct that Sigmoid is unique in its ability to learn in Humanoid-v5. Its bounded nature acts as a safety mechanism, preventing the physics explosions that plague unbounded functions.
>
> - **Rand. Smooth-Leaky (the performance specialist).**
>   However, even under this rigorous normalized metric—which penalizes the Humanoid failure with a **$0.0$** score—our proposed **Rand. Smooth-Leaky** still achieves the highest aggregate Plasticity Score (**0.388 vs. Sigmoid’s 0.340**).
>
> ---
>
> ## 3. Conclusion
>
> The normalized metrics show that while **Sigmoid is safer**, **Rand. Smooth-Leaky** provides such superior adaptation in the solvable environments (achieving nearly **double** the normalized reward of Sigmoid in **Ant** and **HalfCheetah**) that it outweighs its instability in Humanoid.
>
> We have updated **Table 1**, **Figure 4**, and **Section 7** to reflect these new metrics. We believe this new analysis—prompted directly by the reviewer’s critique—significantly strengthens the paper by explicitly surfacing the **“Stability vs. Plasticity” tension** rather than masking it, and we thank the reviewer for suggesting it.
>
> Finally, we have updated **Table F1** and **Table F2** (now **Table F3**) and created a new **Table F2** in Appendix Section F reporting the average per-environment rewards for all activation functions, to provide a direct view of the raw rewards.

---

> ### Author Response · Authors · 2025-11-20
> **Official Comment to Reviewer mY97 II**
>
> Regarding generalizability.
>
> >> First, it’s unclear what the perturbations are.
>
> We noted in the Appendix F caption and followed protocol of Dohare et al. (2023) [1]. The perturbation is a held-out friction variant of the training environment. This is a standard method for testing generalization in robust RL and sim-to-real transfer (e.g., Pinto et al., ICML (2017) [13], Lyu et al., NeurIPS (2024) [14], Zhou et al., (2025) [15], etc).
>
> We have moved this explicit citation into the main text of Section 7.1 to make this setup clear.
>
> ---
>
> >> A small gap with low absolute rewards is less preferable to a large gap with high absolute rewards, but this metric doesn’t capture that.
>
> We completely agree with this assessment. A “perfect” generalization gap of $0.0$ is meaningless if the agent has failed to learn the task entirely (e.g., reward $\approx 0$). As the reviewer correctly points out, looking at the GAP metric in isolation can lead to false positives where broken agents appear to “generalize perfectly.”
>
> To address this, we have revised our presentation to ensure these metrics are never interpreted in isolation:
>
> - **Paired analysis.**
>   We frame the **Plasticity Score** (Table 1) and **Generalization Gap** (Table 3) as a coupled system. Plasticity Score filters for **competence** (absolute reward), while GAP filters for **overfitting** (relative transfer).
>
> - **Explicit breakdown.**
>   We have added a new **Reward Breakdown Table** (Table 2) that reports the raw absolute rewards per environment. This allows to immediately disambiguate the scenario described. In Humanoid-v5, Rand. SL shows negligible GAP (Table 3), but Table 2 reveals this is due to task failure (reward $\approx 0$), whereas Sigmoid shows larger GAP but meaningful rewards.
>
> We have updated **Section 7.1** to warn readers against interpreting low gaps as success without verifying absolute rewards.
>
> ---
>
> >> Regarding measuring generalizability in the supervised setting.
>
> We agree that before analyzing complex “trainability–generalizability tensions” in RL, we should demonstrate the fundamental generalization capabilities of activations in an SL setting.
>
> Following the suggestion, we added **Appendix Section E.2** reporting the standard **Generalization Gap** for Continual Supervised Learning benchmarks. The metric per task is:
> \[
> \mathrm{GAP}_t = \text{TestAcc}_{t,\text{final}} - \text{TrainAcc}_{t,\text{avg}} .
> \]
>
> The results suggest that:
>
> - We observe a **positive gap** (Test $>$ Online Train). The final model outperforms its average state during the learning process.
>
> - The metric exposed significant **overfitting** in the **Deep Fourier Features (DFF)** baseline, which showed large negative gaps (e.g., $-51.24\%$ on CIFAR 5+1). This provides an example of the conflation between trainability and generalizability discussed in Section 7.1. For instance, on CIFAR 5+1, DFF achieves a much higher online accuracy than Rand. Smooth-Leaky ($72.29\%$ vs. $57.01\%$ in Table 2), suggesting superior plasticity. However, the massive negative gap reveals that this “plasticity” was largely overfitting, as the method failed to retain that performance on the test set. This validates our argument.
>
> Nonetheless, we believe that reporting this in the **main body** of the paper would not follow the reasoning behind our GAP metric for RL. This is a distinct concept from the specific challenge we are diagnosing in our RL analysis. For our supervised benchmarks, we instead adopted the **standard generalization metrics used in the continual learning community**. As detailed in Appendix B.5.1, we use:
>
> - **Total Average Online Accuracy (TAOA)** (following setup of Kumar et al., (2023) [2]) for our benchmarks (Sec. 6) as common standard (Cai et al. (2021) [3]; Ghunaim et al. (2023) [4]; Prabhu et al. (2023) [5]), and
> - **Average Accuracy ($\mathrm{ACC}_T$)** for class-incremental case studies (Secs. 3–4).
>
> These are the accepted measures of test-set performance in CL settings.
>
> Our RL analysis in Section 7.1 was explicitly designed to tackle a **different and more specific challenge**: the conflation of **trainability** (Berariu et al., (2021) [6]; Dohare et al., (2021) [7]; Elsayed & Mahmood, (2024) [8]; Lyle et al., (2022) [9]) and **generalizability** (Ash & Adams, (2020) [10]; Zilly et al., (2021) [11]) under a **dynamics shift**. This is a known point of confusion in the field that we wanted to isolate.
>
> Therefore, the $\mathrm{GAP}$ metric was introduced as a **diagnostic tool for robustness to dynamics shift** (a change in friction), which is a core challenge of non-stationary RL. Because the fundamental “generalization” challenge being evaluated is different (**i.i.d. test performance** vs. **robustness to a dynamics shift**), we believe it is most appropriate and insightful to use the standard, specialized metrics for each respective field, rather than forcing a metric from one domain onto a problem it was not designed to measure.

---

> ### Author Response · Authors · 2025-11-20
> **Official Comment to Reviewer mY97 III**
>
> > Tables 1, 2, and 4 are missing standard deviations.
>
> Thank you for bringing this up. We have included standard deviations to Tables 1 and 2, and 95% confidence intervals to Table 4.
>
> ---
>
> > Could you further clarify the reasoning behind the criteria for dead units defined for non-ReLU activation functions?
>
> We agree with the reviewer that the “dead unit” concept is clearest for ReLU, where there is an explicit region with output 0 and slope 0. Extending that terminology to other activations is indeed nontrivial and is precisely part of the difficulty when comparing such diverse functions. We have added a clarification in line 134, and the threshold was also indicated in line 174.
>
> We are explicitly detecting units that are operating in regions where their gradient is effectively zero, which is the classical notion of “deadness” and is directly tied to our Dead-Band Width analysis.
>
> First, we would like to clarify our terminology. We deliberately avoided using the term **“dormant neurons”** in the paper to not suggest that we were implementing the specific mechanism and per-neuron metric introduced in *“The Dormant Neuron Phenomenon in Deep Reinforcement Learning”* [12]. Our statistic is different: it is a simple diagnostic measure of effective inactivity that we can apply uniformly across activation families.
>
> Concretely, our criteria (as implemented in the analysis code) are designed to capture distinct failure modes across activation families:
>
> ### Saturating activations
> For functions that have flat regions where the derivative is (approximately) zero, we identify units whose post-activations lie close to those plateaus:
>
> - **Tanh / Sigmoid:**
>   We check whether the unit’s output is in the saturated region, e.g.
>   - Tanh: `abs(a) > 0.98`
>   - Sigmoid: `a > 0.98` or `a < 0.02`
>
> - **ELU / CELU / SELU:**
>   These saturate on the negative side. We check whether output is near the negative-side limit, e.g.
>   - `a < -alpha + 0.02`
>
> ### Non-saturating activations
> For non-saturating functions such as **Leaky-ReLU, Swish, GELU**, and our proposed **Smooth-Leaky**, there is no region where the theoretical gradient is exactly zero. Therefore, we cannot use a saturation-based definition. Instead, we define a unit as *effectively inactive* if its absolute post-activation is negligible relative to the typical working scale of that layer.
>
> The logic is purely functional. A unit that persistently outputs values orders of magnitude smaller than its peers (e.g., \(10^{-4}\) when others are around \(10^{-1}\)) contributes negligibly to the next layer’s pre-activations. Because its forward contribution is tiny, the gradient signal flowing back through its outgoing weights is also tiny, further reducing the chance that this unit will ever “wake up.”
>
> ---
>
> **In summary:** for saturating activations we identify units stuck in flat-gradient plateaus, while for non-saturating activations we identify units that are effectively negligible given the working scale of their own layer. Our “dead unit fraction” is therefore a **hybrid metric** tracking *effective inactivity*, not exclusively ReLU-style hard deadness.
>
> Finally, we stress that while this intuition is broadly consistent with the idea that under-utilized units can emerge in over-parameterized networks (as also qualitatively discussed in [12]), our metric is **not** the same as the dormant-neuron score used there. We do not compute per-neuron layer-normalized averages nor apply any reinitialization mechanism; instead, we use a simple, activation-specific thresholding rule to obtain a comparable inactivity statistic across all activation architectures in our study.

---

> ### Author Response · Authors · 2025-11-20
> **Official Comment to Reviewer mY97 IV - References**
>
> # References
> * [1]. Dohare, S., Hernandez-Garcia, J. F., Lan, Q., Rahman, P., Mahmood, A. R., & Sutton, R. S. (2024). Loss of plasticity in deep continual learning. Nature, 632(8026), 768-774.
> * [2]. Saurabh Kumar, Henrik Marklund, and Benjamin Van Roy. Maintaining plasticity in continual learning via regenerative regularization. arXiv preprint arXiv:2308.11958, 2023.
> * [3]. Zhipeng Cai, Ozan Sener, and Vladlen Koltun. Online continual learning with natural distribution shifts: An empirical study with visual data. In Proceedings of the IEEE/CVF international conference on computer vision, pp. 8281–8290, 2021
> * [4]. Yasir Ghunaim, Adel Bibi, Kumail Alhamoud, Motasem Alfarra, Hasan Abed Al Kader Hammoud, Ameya Prabhu, Philip HS Torr, and Bernard Ghanem. Real-time evaluation in online continual learning: A new hope. In Proceedings of the IEEE/CVF conference on computer vision and pattern recognition, pp. 11888–11897, 2023.
> * [5]. Ameya Prabhu, Zhipeng Cai, Puneet Dokania, Philip Torr, Vladlen Koltun, and Ozan Sener. Online continual learning without the storage constraint. arXiv preprint arXiv:2305.09253, 2023.
> * [6] Tudor Berariu, Wojciech Czarnecki, Soham De, Jorg Bornschein, Samuel Smith, Razvan Pascanu, and Claudia Clopath. A study on the plasticity of neural networks. arXiv preprint arXiv:2106.00042, 2021.
> * [7]. Shibhansh Dohare, Richard S Sutton, and A Rupam Mahmood. Continual backprop: Stochastic gradient descent with persistent randomness. arXiv preprint arXiv:2108.06325, 2021.
> * [8]. Mohamed Elsayed and A Rupam Mahmood. Addressing loss of plasticity and catastrophic forgetting in continual learning. arXiv preprint arXiv:2404.00781, 2024.
> * [9]. Clare Lyle, Mark Rowland, and Will Dabney. Understanding and preventing capacity loss in reinforcement learning. arXiv preprint arXiv:2204.09560, 2022
> * [10]. Jordan Ash and Ryan P Adams. On warm-starting neural network training. Advances in neural information processing systems, 33:3884–3894, 2020
> * [11]. Julian Zilly, Alessandro Achille, Andrea Censi, and Emilio Frazzoli. On plasticity, invariance, and mutually frozen weights in sequential task learning. Advances in neural information processing systems, 34: 12386–12399, 2021.
> * [12]. Sokar, G., Agarwal, R., Castro, P. S., & Evci, U. (2023, July). The dormant neuron phenomenon in deep reinforcement learning. In International Conference on Machine Learning (pp. 32145-32168). PMLR.
> * [13]. Pinto, L., Davidson, J., Sukthankar, R., & Gupta, A. (2017, July). Robust adversarial reinforcement learning. In International conference on machine learning (pp. 2817-2826). PMLR.
> * [14]. Lyu, J., Xu, K., Xu, J., Yang, J. W., Zhang, Z., Bai, C., ... & Li, X. (2024). Odrl: A benchmark for off-dynamics reinforcement learning. Advances in Neural Information Processing Systems, 37, 59859-59911.
> * [15]. Zhou, H., Zhuang, Z. &; Wang, D.. (2025). Stay Hungry, Keep Learning: Sustainable Plasticity for Deep Reinforcement Learning. Proceedings of the 42nd International Conference on Machine Learning, in Proceedings of Machine Learning Research 267:79644-79672
> * [16]. Agarwal, R., Schwarzer, M., Castro, P. S., Courville, A. C., & Bellemare, M. (2021). Deep reinforcement learning at the edge of the statistical precipice. Advances in neural information processing systems, 34, 29304-29320.

---

### Author Response · Authors · 2025-11-20
**Official Comment by Authors**

We wish to thank all reviewers for the time and consideration they have devoted to reviewing our work, as well as for their constructive feedback. We took these concerns very seriously and feel the revised manuscript is significantly stronger as a result. We have made a compilation list with all changes made to the paper to speed up the reviewers' process and to help in identifying revisions directly.

## Main Paper Structural & Content Additions

1. **Figure C1** *(Performance vs. $\overline{s}$ and Dead Units vs. $\overline{s}$)* was moved from the appendix to **Section 3**. This provides direct visual evidence for the "Goldilocks zone" and its associated "dead unit" failure mode.

2. **Figure C2** *($\lambda_{\max}$ vs. $\overline{s}$ and Effective Rank vs. $\overline{s}$)* was moved from the appendix to **Section 3**. This provides the evidence for the second failure mode (high curvature) that defines the "Goldilocks zone".

3. The text in **Section 3** regarding *Failure Modes for Slope Magnitude* was revised, condensed, and updated to improve readability.

4. The text in **Section 4** was revised. **Figure D1** *(Correlations with Dead-Band Width)* was moved from the appendix to **Section 4**. This provides the key evidence that our analytical DBW metric strongly predicts empirical saturation and non-recovery rates.

6. We revised **Section 7** and updated **Table 3** and **Figure 6** with a **Min-Max Normalized IQM** score rather than a vanilla average, following literature standards (see answer to Reviewer mY97).

8. We updated **Table 4** to provide $\Delta_{\mathrm{IQM}}$ rather than median.

10. We added clarifications to **Section 7.1** with respect to **Plasticity Score** and $\Delta(\mathrm{GAP})$ are a two-metric system designed to expose the "trainability-transfer tension", and that the RL test environment perturbation is a **"held-out friction variant,"**

12. A clarification regarding what optimizer was used (**Adam**) was included in **Section 2**.

13. We updated **Tables 1, 2, and 4** to include standard deviations (Tables 1 and 2) and confidence intervals (Table 4) as requested by reviewers fk2M and mY97.

14. A clarification regarding dead-unit definition for non-ReLU activation functions was included in **line 143**. The threshold used to quantify those for Case Study 2 was indicated in **line 174**. The implementation resides in `utils/dead_units.py`.

## New Experimental Baselines

1. We have added **Deep Fourier Features (DFF)** as a new baseline in the Continual Supervised Learning benchmarks (**Table 2**) and Continual RL experiments. Since it was not competitive with the set of activations shown in C-RL experiments, the RL results only appear in the complete appendix tables (**Tables F1 and F2**).

2. We have added **SwiGLU** as a new baseline in the CSL benchmarks (**Table 2**) and Continual RL experiments; the same applies as for DFF (**Tables F1 and F2**).

3. The text in **Section 6** was updated to introduce these new baselines and provide context for their inclusion.

## New Additions to the Appendix

1. We added a new subsection to **Appendix B (Experimental Design)** providing a clear *Rationale for Optimizer Choice*. This section explains our deliberate choice of the Adam optimizer as a "stress test" for plasticity, citing relevant literature that implicates it as a contributor to plasticity loss.

2. We added a targeted experiment in a new **Appendix Section E.3: "Continual Learning Interventions Targeting Loss of Plasticity"** showing the performance of **L2 Init, SNR, and EWC** when combined with all activation functions studied throughout the paper. Results are reported in **Tables E.4–E.11**.

3. We added the Generalization **GAP** results for the CSL benchmarks in a new appendix section. While we maintain that **Total Average Online Accuracy** remains the standard and most appropriate metric for these benchmarks (as detailed in our response regarding the distinction between SL generalization and RL robustness to reviewer mY97), we agree that reporting GAP enhances completeness and transparency.

4. We updated **Section C1 and Figure C1** to reflect statistical changes due to the addition of new hyperparameter configurations. The conclusion remains the same.

5. We updated **Table B1** with more hyperparameter values for activations such as **Leaky-ReLU, SeLU, eLU, cELU, Swish, and GeLU**.

6. We updated **Section F** after introducing the metric IQM. We also introduced a new **Table F2** that reports raw rewards per environment for all activation functions.

7. We updated results regarding **Bo-PReLU** and **R-SeLU** as other custom activation suggestions and compared them to their inspirations (**PReLU and SeLU**), and our other custom activations as suggested by Reviewer hFaa (Section G). We showed how they are competitive and how we can develop activations based on ideas studied in the paper.

---

### Author Response · Authors · 2025-11-20
**Case Study 1 and Case Study 2 enhancements**

Since this has been requested by two independent reviewers, we would like to propose a unified answer. First of all, we thank both reviewers (fk2M and mY97) for this critical feedback. You are both correct that the original 9-page limit forced us to move essential figures for Case Studies 1 and 2 to the appendix, which weakened the main paper's narrative and readability. We apologize for this and, thanks to the newly allotted extra page, we have made significant structural changes to Case Study 1 and Case Study 2 to directly address this concern. Changes to other sections are reported in the main, general comment that serves as a summary of our revised draft.

We are confident that these changes make Sections 3 and 4 a self-contained, coherent, and empirically supported story, which we agree greatly improves the paper. Specifically, the following key changes were made:

---

## Section 3 (Case Study 1: “Goldilocks Zone”)

- We moved **Figure C1** from the appendix into the main text. This is the most critical figure for this section, as its subplots visually:
  1. define the “Goldilocks zone” by plotting *Performance vs.* $\overline{s}$, and
  2. show the first failure mode ($\overline{s} \to 0$) by plotting *Dead Units vs.* $\overline{s}$, directly supporting our claim about the “dead-unit regime.”

- We also moved **Figure C2** into the main text. This figure provides evidence for the second failure mode ($\overline{s} \ge 0.9$), by plotting the “curvature explosion” (*$\lambda_{\max}$ vs.* $\overline{s}$) and the resulting drop in representational diversity (*Effective Rank vs.* $\overline{s}$).

- Both figures have been condensed into a single figure (now **Figure 1**). As we also report, we collected results for a larger hyperparameter sweep (as pointed out by reviewers 5MGv and mY97), and we now include them. The updated **Figure 1** provides stronger visual confirmation of the two competing failure modes that define the “Goldilocks zone”:

  - **Left side ($\overline{s} \to 0$):** performance collapse correlates with a surge in dead units (gradient starvation).
  - **Right side ($\overline{s} \ge 1$):** performance collapse correlates with a spike in principal curvature and a change in effective rank, showing that despite high unit activity, the optimization landscape becomes unstable.

  Together, this visually confirms that plasticity requires balancing these two opposing constraints.

---

## Section 4 (Case Study 2: Desaturation Dynamics)

- We moved **Figure D1** from the appendix into the main text. This figure is the driving argument of the section: it provides explicit, strong correlations showing that our analytical **Dead-Band Width (DBW)** metric is a valid predictor of empirical failure rates (namely *Avg. AUSC* and *Avg. SF Non-Recovery Rate*).

- This section has also been rewritten and condensed. In particular, the surrounding text in Section 4 has been made clearer, less repetitive, and updated to integrate the newly added Figure D1.

---

### Author Response · Authors · 2025-11-20
**Hyper-parameter sweeps and extra combinations**

Since this has been requested by two independent reviewers, we would like to propose a unified answer. First of all, we thank both reviewers (5MGv and mY97).

We fully accept the reviewer’s critique regarding the discrepancy in the hyperparameter search space. The reviewer is correct that, in an ideal compute-normalized comparison, we would allocate each activation function the same overall computational budget (e.g., total number of training runs). Because our proposed functions have 3–4 degrees of freedom while baselines like Leaky-ReLU have only 1, we opted to provide the same computation budget to each free hyperparameter, such that each would be sufficiently fit. We acknowledge that this gave our method more total shots at the goal, and that a general weakness of our proposed activation function is the additional hyperparameters that must be tuned.

However, we respectfully submit that this discrepancy does not account for the performance gap, for two key reasons:

1. **Single-parameter baselines saturate quickly.**
   Leaky-ReLU has a single shape degree of freedom (negative slope $\alpha$). In 1-parameter spaces, performance typically saturates quickly. “Spending” 400 more runs on fine-grain hyperparameter tuning for Leaky-ReLU yields negligible gains compared to the coarse search alone. To verify this—and to directly address the reviewer’s concern—we conducted supplemental experiments (added to Appendix B, Table B1) where we significantly expanded the search grids for Leaky-ReLU, SeLU, ELU, and Swish. This suggests the baselines were already operating at their performance ceiling, and the “unfair advantage” of extra compute would not have helped them further.

2. **Our gains reflect a higher ceiling, not lucky tuning.**
   Smooth-Leaky (SL) and Randomized Smooth-Leaky (RSL) expose additional shape controls for transition width, scale factor, or the upper and lower bounds for the randomized $\alpha$. A significant portion of these configurations outperformed the best-tuned baseline. This indicates that our method’s superiority is not due to finding one “lucky needle in a haystack” (which would imply overfitting to hyperparameters), but rather due to the inherent geometric properties of the function (smoothness + variance) that unlock a higher performance ceiling than is mathematically possible with fixed or 1-parameter functions.

In the revised manuscript (Section B.2), we explicitly discuss this trade-off: while multi-parameter designs introduce a larger search space (a practical limitation), they break the performance ceiling that limits simpler activations.

---

### Meta-Review · Area_Chair_BZ2K · 2026-01-04

**Summary:**

I recommend accepting the paper, but I strongly suggest working further based on the critical review from Reviewer mY97, where a concern is also shared by another reviewer.

The main issue is the computational fairness of the hyperparameter search, which I discussed in the later sections. This is an important question: whether to always go for computational fairness or search for more hyperparameter configurations if an algorithm holds more promise? While the authors added additional search for the baselines, the best thing to do here is to show the trend of these algorithms as more search is added. The authors already alluded that baselines with fewer hyperparameters saturate quickly in their performance as more search is added. This is best illustrated by showing the performance trend with respect to the number of configurations searched. This way, we clearly get a sense which algorithm would have been preferred for each computational budget. It will also illustrate whether it is worth searching more configurations for a particular algorithm or whether performance has already plateaued.

Notes to authors:

Authors claim: “We noted in the Appendix F caption and followed protocol of Dohare et al. (2023) [1]. The perturbation is a held-out friction variant of the training environment.” But there was no held-out environment used by Dohare et al.

Moreover, Abbas et al. (2023) was cited three times for “randomized MuJoCo friction” while that paper does not use MuJoCo tasks at all.

**Reviewer Concerns:**

Most major concerns are addressed except the ones by Reviewer mY97, which are mentioned below.

Reviewer mY97 mentioned that the proposed activations have way more hyperparameters resulting in more extensive grid search than baselines (also raised by Reviewer 5MGv). They were also tuned on a per-setting basis. The reviewer also had a concern about the plasticity metric that didn’t match the superior performance of a baseline. The authors acknowledged this limitation. The right way to address this issue is to provide a comparison where all algorithms receive a similar amount of total compute for the hyperparameter search. That was not done. The weakness regarding metric is addressed by redoing the results with a new metric, which needs to be reviewed and discussed by all reviewers.

Reviewer kvHD sought the impact of the activations on some existing continual learning approaches. The authors responded by adding extra experiments.

Reviewer hFaa sought a theoretical explanation of the found Goldilocks zone and interaction of the activations with different optimizers. The authors clarified that the zone is a consequence of some underlying properties, not a cause.

Reviewer fK2M asked for confidence intervals for the results. The authors have provided them.

**Reviewer Scores:**

Given the responses, I would have increased most scores, including the most critical one. A bigger concern here is the computational fairness question raised by one of the reviewers. For that, I have added a strong recommendation for the authors.

---

### Decision · Program_Chairs · 2026-01-26

Accept (Poster)